# General Low-rank Matrix Optimization: Geometric Analysis and Sharper Bounds

**Haixiang Zhang**
Department of Mathematics
University of California, Berkeley
Berkeley, CA 94704
`haixiang_zhang@berkeley.edu`

**Yingjie Bi**
Department of IEOR
University of California, Berkeley
Berkeley, CA 94704
`yingjiebi@berkeley.edu`

**Javad Lavaei**
Department of IEOR
University of California, Berkeley
Berkeley, CA 94704
`lavaei@berkeley.edu`

## Abstract

This paper considers the global geometry of general low-rank minimization problems via the Burer-Monteiro factorization approach. For the rank-1 case, we prove that there is no spurious second-order critical point for both symmetric and asymmetric problems if the rank-2 RIP constant $\delta$ is less than $1/2$. Combining with a counterexample with $\delta = 1/2$, we show that the derived bound is the sharpest possible. For the arbitrary rank-$r$ case, the same property is established when the rank-$2r$ RIP constant $\delta$ is at most $1/3$. We design a counterexample to show that the non-existence of spurious second-order critical points may not hold if $\delta$ is at least $1/2$. In addition, for any problem with $\delta$ between $1/3$ and $1/2$, we prove that all second-order critical points have a positive correlation to the ground truth. Finally, the strict saddle property, which can lead to the polynomial-time global convergence of various algorithms, is established for both the symmetric and asymmetric problems when the rank-$2r$ RIP constant $\delta$ is less than $1/3$. The results of this paper significantly extend several existing bounds in the literature.

## 1 Introduction

Given the natural numbers $n$, $m$ and $r$, consider the low-rank matrix optimization problems

$$\min_{M \in \mathbb{R}^{n \times n}} \ f_s(M) \quad \text{s.t. } \text{rank}(M) \leq r, \quad M^T = M, \quad M \succeq 0 \tag{1}$$

and

$$\min_{M \in \mathbb{R}^{n \times m}} \ f_a(M) \quad \text{s.t. } \text{rank}(M) \leq r, \tag{2}$$

where the functions $f_s(\cdot)$ and $f_a(\cdot)$ are twice continuously differentiable. Problems (1)-(2) are referred to as the *symmetric* and the *asymmetric* problems, respectively. In addition, we call these problems *linear* if the objective function is induced by a linear measurement operator, i.e.,

$$f(M) = \tfrac{1}{2} \|\mathcal{A}(M) - b\|_F^2$$

for some vector $b \in \mathbb{R}^p$ and linear operator $\mathcal{A}$ mapping each matrix $M$ to a vector in $\mathbb{R}^p$, where $f(M)$ denotes either $f_s(M)$ or $f_a(M)$. Those problems not fitting into the above model are called *nonlinear*. One common example with non-linearity is the one-bit matrix sensing problem; please see Zhu et al. (2018); Li et al. (2019); Zhu et al. (2021) for more concrete discussions. Low-rank

35th Conference on Neural Information Processing Systems (NeurIPS 2021).

optimization problems arise in a wide range of applications, e.g., matrix completion (Candès & Recht, 2009; Recht et al., 2010), phase synchronization (Singer, 2011; Boumal, 2016), and phase retrieval (Shechtman et al., 2015); see Chen & Chi (2018); Chi et al. (2019) for an overview of the topic. To overcome the non-convex rank constraint, one may resort to convex relaxations. The approach of replacing the rank constraint with a nuclear norm regularization is proven to provide the optimal sample complexity (Candès & Recht, 2009; Recht et al., 2010; Candès & Tao, 2010). However, solving the convexified problems involves computing a Singular Value Decomposition (SVD) in each iteration and results in heavy computational burdens; see the numerical comparison in Zheng & Lafferty (2015). Along with the issue of large space complexities, the convexification approach is impractical for large-scale problems. Therefore, it is important to design efficient alternative methods with similar theoretical guarantees. Another line of works generalizes techniques from Orthogonal Matching Pursuit (OMP) to the low-rank matrix problem (Shalev-Shwartz et al., 2011; Axiotis & Sviridenko, 2020), which also require implementing SVD for $r$ times in their algorithms.

## 1.1 Burer-Monteiro factorization and basic properties

Instead of directly solving convex relaxations of problems (1)-(2), we consider a computationally efficient approach, namely the Burer-Monteiro factorization (Burer & Monteiro, 2003). The factorization approach is based on the observation that any matrix $M \in \mathbb{R}^{n \times m}$ with rank at most $r$ can be written in the form of $UV^T$, where $U \in \mathbb{R}^{n \times r}$ and $V \in \mathbb{R}^{m \times r}$. Then, the asymmetric problem (2) is equivalent to

$$\min_{U \in \mathbb{R}^{n \times r}, V \in \mathbb{R}^{m \times r}} h_a(U, V), \tag{3}$$

where $h_a(U, V) := f_a(UV^T)$. Similarly, the symmetric problem (1) is equivalent to

$$\min_{U \in \mathbb{R}^{n \times r}} h_s(U), \tag{4}$$

where $h_s(U) := f_s(UU^T)$. The Burer-Monteiro factorization provides a natural parameterization of the low-rank structure of the unknown solution, and reformulates problems (1)-(2) as unconstrained optimization problems. In addition, the number of variables reduces from $O(n^2)$ or $O(nm)$ to as low as $O(rn)$ or $O(r(n+m))$ when $r \ll \min\{n, m\}$. However, the reformulated problems are highly non-convex, and $\mathcal{NP}$-hard to solve in general. On the other hand, these problems share a specific non-convex structure, which makes it possible to utilize the structure and design efficient algorithms to find a global optimum under some conditions. In addition to the special structure, a regularity condition, named the Restricted Isometry Property, can be used to guarantee the convergence of common iterative algorithms. We state the following two definitions only in the context of the symmetric problem since the corresponding definitions for the asymmetric problem are similar.

**Definition 1** (Recht et al. (2010); Zhu et al. (2018)). Given natural numbers $r$ and $t$, the function $f_s(\cdot)$ is said to satisfy the **Restricted Isometry Property** (RIP) of rank $(2r, 2t)$ for a constant $\delta \in [0, 1)$, denoted as $\delta$-RIP$_{2r,2t}$, if

$$(1 - \delta)\|K\|_F^2 \leq \left[\nabla^2 f_s(M)\right](K, K) \leq (1 + \delta)\|K\|_F^2$$

holds for all matrices $M, K \in \mathbb{R}^{n \times n}$ such that $\text{rank}(M) \leq 2r, \text{rank}(K) \leq 2t$, where $\left[\nabla^2 f_s(M)\right](\cdot, \cdot)$ is the curvature of the Hessian at point $M$.

The RIP condition appears in a variety of applications of the low-rank matrix optimization problem. For instance, in the case of linear measurements with a Gaussian model, Candès & Recht (2009) showed that $O\left(nr/\delta^2\right)$ samples are enough to ensure the $\delta$-RIP$_{2r,2r}$ property with high probability. Please see the survey paper by Chi et al. (2019) for more examples. In certain applications, even the RIP condition cannot be established over the whole low-rank manifold, we are able to establish similar strongly convex and smooth conditions on part of the manifold. If the iteration points of algorithms are constrained to or regularized (either explicitly or implicitly) towards those benign regions, the proof techniques in this work may still be applicable. Examples include the phase retrieval problem (Ma et al., 2018) and the matrix completion problem (Chen et al., 2020). However, the analysis of the case when the strong convexity does not hold is usually application-specific and cannot be generalized to general low-rank problems. Moreover, the RIP assumption is standard in the literature of general low-rank matrix optimization problem. Furthermore, if we drop the strong convexity assumption, we are unable to achieve linear convergence in general (Bhojanapalli et al., 2016a). The

work by Zhang et al. (2019) shows that the existence of RIP is enough to obtain guarantees on the local landscape of the problem and the size of this local region depends on the RIP constant that can be anything between 0 and 1 (however, the provided bounds on the RIP constant are not sharp). Although we aimed to obtain sharp bounds on the RIP constant for global landscape of the problem in this paper, we believe that our analysis can be adopted to obtain sharp RIP bounds for local regions. We leave the precise derivation to a future work since it needs a number of lemma and we have space restrictions. We note that the RIP property is equivalent to the restricted strongly convex and smooth property defined in Wang et al. (2017); Park et al. (2018); Zhu et al. (2021) with the condition number $(1 + \delta)/(1 - \delta)$. Intuitively, the RIP property implies that the Hessian matrix is close to the identity tensor when the perturbation is restricted to be low-rank. This intuition naturally leads to the following definition.

**Definition 2** (Bi & Lavaei (2021)). Given a natural number $r$, the function $f_s(\cdot)$ is said to satisfy the **Bounded Difference Property** (BDP) of rank $2r$ for a constant $\kappa \geq 0$, denoted as $\kappa$-BDP$_{2r}$, if

$$\left| \left[ \nabla^2 f_s(M) - \nabla^2 f_s(M') \right] (K, L) \right| \leq \kappa \|K\|_F \|L\|_F$$

holds for all matrices $M, M', K, L \in \mathbb{R}^{n \times n}$ such that $\text{rank}(M), \text{rank}(M'), \text{rank}(K), \text{rank}(L) \leq 2r$.

It has been proven in Bi & Lavaei (2021, Theorem 1) that those functions satisfying the $\delta$-RIP$_{2r,2r}$ property also satisfy the $4\delta$-BDP$_{2r}$ property. With the RIP property, there are basically two categories of algorithms that can solve the factorized problem in polynomial time. Algorithms in the first category require a careful initialization so that the initial point is already in a small neighbourhood of a global optimum, and a certain local regularity condition in the neighbourhood ensures that local search algorithms will converge linearly to a global optimum; see Tu et al. (2016); Bhojanapalli et al. (2016a); Park et al. (2018) for a detailed discussion. The other class of algorithms is able to converge globally from a random initialization. The convergence of these algorithms is usually established via the geometric analysis of the landscape of the objective function. One of the important geometric properties is the strict saddle property (Sun et al., 2018), which combined with the smoothness properties can guarantee the global polynomial-time convergence for various saddle-escaping algorithms (Jin et al., 2017, 2018; Sun et al., 2018; Huang & Becker, 2019). For the linear case, Ge et al. (2016, 2017) proved the strict saddle property for both problems (3)-(4) when the RIP constant is sufficiently small. More recently, Zhu et al. (2021) extended the results to the nonlinear asymmetric case. Moreover, a weaker geometric property, namely the non-existence of *spurious (non-global) second-order critical points*, has been established for both problems when the RIP constant is small (Li et al., 2019; Ha et al., 2020). We note that second-order critical points are points that satisfy the first-order and the second-order necessary optimality conditions, and thus the result of the non-existence of second-order critical points implies the non-existence of spurious local minima. Under certain regularity conditions, this weaker property is also able to guarantee the global convergence from a random initialization without an explicit convergence rate (Lee et al., 2016; Panageas & Piliouras, 2016). Please refer to Table 1 for a summary of the state-of-the-art results.

Most of the aforementioned papers are based on the following assumption on the low-rank critical points of the functions $f_s(\cdot)$ and $f_a(\cdot)$:

**Assumption 1.** The function $f_a(\cdot)$ has a first-order critical point $M_a^*$ such that $\text{rank}(M_a^*) \leq r$. Similarly, the function $f_s(\cdot)$ has a first-order critical point $M_s^*$ that is symmetric, positive semi-definite and of rank at most $r$.

This assumption is inspired by the noiseless matrix sensing problem in the linear case for which the non-negative objective function becomes zero (the lowest value possible) at the true solution. This is a natural property of the matrix sensing problem for nonlinear measurement models as well. Under the above assumption and the RIP property, Zhu et al. (2018) proved that $M_s^*$ and $M_a^*$ are the unique global minima of problems (1)-(2).

**Theorem 1** (Zhu et al. (2018)). *If the functions $f_s(\cdot)$ and $f_a(\cdot)$ satisfy the $\delta$-RIP$_{2r,2r}$ property, then the critical points $M_s^*$ and $M_a^*$ are the* unique global minima *of problems* (1)-(2).

Given a solution $(U^*, V^*)$ to problem (3), we observe that $(U^*P, V^*P^{-T})$ is also a solution for any invertible $P \in \mathbb{R}^{r \times r}$. This redundancy may induce an extreme non-convexity on the landscape of the objective function. To reduce this redundancy, Tu et al. (2016) considered the regularized problem

$$\min_{U \in \mathbb{R}^{n \times r}, V \in \mathbb{R}^{m \times r}} \rho(U, V), \tag{5}$$

Table 1: Comparison of the state-of-the-art results and our results. Here $\delta_{2r,2t}$ and $\kappa$ are the $\text{RIP}_{2r,2t}$ and $\text{BDP}_{2r}$ constants of $f_s(\cdot)$ or $f_a(\cdot)$, respectively. Constant $\alpha(M_a^*) \in (0,1)$ only depends on $M_a^*$.

| Problem Setups | | No Spurious Second-order Critical Pts. | | Strict Saddle Property | |
|---|---|---|---|---|---|
| | | Existing | Ours | Existing | Ours |
| Rank-1 Sym. | Linear | $\delta_{2,2} < \frac{1}{2}$ (Zhang et al., 2019) | $\delta_{2,2} < \frac{1}{2}$ | - | - |
| | Nonlinear | $\delta_{2,2} < \frac{2-O(\kappa)}{4+O(\kappa)}$ (Bi & Lavaei, 2021) | $\delta_{2,2} < \frac{1}{2}$ | - | - |
| Rank-1 Asym. | Linear & Nonlinear | - | $\delta_{2,2} < \frac{1}{2}$ | - | - |
| Rank-$r$ Sym. | Linear | $\delta_{2r,2r} < \frac{1}{5}$ (Ge et al., 2016) | $\delta_{2r,2r} \le \frac{1}{3}$ | $\delta_{2r,2r} < \frac{1}{10}$ (Ge et al., 2017) | $\delta_{2r,2r} < \frac{1}{3}$ |
| | Nonlinear | $\delta_{2r,4r} < \frac{1}{5}$ (Li et al., 2019) | $\delta_{2r,2r} \le \frac{1}{3}$ | - | $\delta_{2r,2r} < \frac{1}{3}$ |
| Rank-$r$ Asym. | Linear | $\delta_{2r,2r} < \frac{1}{3}$ (Ha et al., 2020) | $\delta_{2r,2r} \le \frac{1}{3}$ | $\delta_{2r,2r} < \frac{1}{20}$ (Ge et al., 2017) | $\delta_{2r,2r} < \frac{1}{3}$ |
| | Nonlinear | $\delta_{2r,2r} < \frac{1}{3}$ (Ha et al., 2020) | $\delta_{2r,2r} \le \frac{1}{3}$ | $\delta_{2r,4r} < \frac{\alpha(M_a^*)}{100}$ (Zhu et al., 2021) | $\delta_{2r,2r} < \frac{1}{3}$ |

where

$$\rho(U,V) := h_a(U,V) + \frac{\mu}{4} \cdot g(U,V)$$

with a constant $\mu > 0$ and the regularization term

$$g(U,V) := \|U^T U - V^T V\|_F^2.$$

The regularization term is introduced to balance the magnitudes of $U^*$ and $V^*$. Zhu et al. (2018) showed that the regularization term does not introduce bias and thus problem (5) is equivalent to the original problem (2) in the sense that any first-order critical point $(U,V)$ of problem (2) corresponds to a first-order critical point of problem (5) with balanced energy, i.e. $U^T U = V^T V$.

**Theorem 2** (Zhu et al. (2018)). *Any first-order critical point $(U^*, V^*)$ of problem* (5) *satisfies* $(U^*)^T U^* = (V^*)^T V^*$. *Moreover, problems* (2) *and* (5) *are equivalent.*

Detailed optimality conditions for problems (1)-(5) are provided in the appendix.

## 1.2 Contributions

In this work, we analyze the geometric properties of problems (4)-(5). Novel analysis methods are developed to obtain less conservative conditions for guaranteeing benign landscapes for both problems. We note that, unlike the linear measurements case, the RIP constant of nonlinear problems may not concentrate to 0 as the number of samples increases. Therefore, a sharper RIP bound leads to theoretical guarantees that hold under less stringent statistical requirements. In addition, even if the RIP constant concentrates to 0 when more samples are included, there may only be a limited number of samples available, either due to the constraints of specific applications or to the great expense of taking more samples. Hence, obtaining a sharper RIP bound is essential for many applications. We summarize our results in Table 1. More concretely, the contributions of this paper are three-folds.

First, we derive necessary conditions and sufficient conditions for the existence of spurious second-order critical points for both symmetric and asymmetric problems. Using our necessary conditions, we show that the $\delta$-$\text{RIP}_{2r,2r}$ property with $\delta \le 1/3$ is enough to guarantee the non-existence of such points. This result provides a marginal improvement to the previous work (Ha et al., 2020), which developed the sufficient condition $\delta < 1/3$ for asymmetric problems, and is a major improvement over Ge et al. (2016) and Li et al. (2019), which requires $\delta < 1/5$ for symmetric problems. With this non-existence property and under some common regularity conditions, Lee et al. (2016); Panageas &

Piliouras (2016) showed that the vanilla gradient descent method with a small enough step size and a random initialization almost surely converges to a global minimum. We note that the convergence rate was not studied and could theoretically be exponential in the worst case. In addition, by studying our necessary conditions, we show that every second-order critical point has a positive correlation to the global minimum when $\delta \in (1/3, 1/2)$. When $\delta = 1/2$, a counterexample with spurious second-order critical points is given by utilizing the sufficient conditions. We note that the sufficient conditions can greatly simplify the construction of counterexamples.

Second, we separately study the rank-1 case to further strengthen the bounds. In particular, we utilize the necessary conditions to prove that the $\delta$-RIP$_{2,2}$ property with $\delta < 1/2$ is enough for the non-existence of spurious second-order critical points. Combining with a counterexample in the $\delta = 1/2$ case, we conclude that the bound $\delta < 1/2$ is the sharpest bound for the rank-1 case. Our results significantly extend the bounds in Zhang et al. (2019) derived for the linear symmetric case to the linear asymmetric case and the general nonlinear case. It also improves the bound in Bi & Lavaei (2021) by dropping the BDP constant.

Third, we prove that in the exact parametrization case, problems (4)-(5) both satisfy the strict saddle property (Sun et al., 2018) when the $\delta$-RIP$_{2r,2r}$ property is satisfied with $\delta < 1/3$. This result greatly improves the bounds in Ge et al. (2017); Zhu et al. (2021) and extends the result in Ha et al. (2020) to approximate second-order critical points. With the strict saddle property and certain smoothness properties, a wide range of algorithms guarantee a global polynomial-time convergence with a random initialization; see Jin et al. (2017, 2018); Sun et al. (2018); Huang & Becker (2019). Due to the special non-convex structure of our problems and the RIP property, it is possible to prove the boundedness of the trajectory of the perturbed gradient descent method using a similar method as in Jin et al. (2017). Since the smoothness properties are satisfied over a bounded region, combined with the strict saddle property, it follows that the perturbed gradient descent method (Jin et al., 2017) achieves a polynomial-time global convergence when $\delta < 1/3$.

### 1.3 Notation and organization

The operator 2-norm and the Frobenius norm of a matrix $M$ are denoted as $\|M\|_2$ and $\|M\|_F$, respectively. The trace of matrix $M$ is denoted as $\mathrm{tr}(M)$. The inner product between two matrices is defined as $\langle M, N \rangle := \mathrm{tr}(M^T N)$. For any matrix $M \in \mathbb{R}^{n \times m}$, we denote its singular values by $\sigma_1(M) \geq \cdots \geq \sigma_k(M)$, where $k := \min\{n, m\}$. For any symmetric matrix $M \in \mathbb{R}^{n \times n}$, we denote its eigenvalues by $\lambda_1(M) \geq \cdots \geq \lambda_n(M)$. The minimal eigenvalue is denoted as $\lambda_{min}(\cdot)$. For any matrix $U$, we use $\mathcal{P}_U$ to denote the orthogonal projection onto the column space of $U$. For any matrices $A, B \in \mathbb{R}^{n \times m}$, we use $A \otimes B$ to denote the fourth-order tensor whose $(i, j, k, \ell)$ element is $A_{i,j}B_{k,\ell}$. The identity tensor is denoted as $\mathcal{I}$. The notation $M \succeq 0$ means that the matrix $M$ is symmetric and positive semi-definite. The sub-matrix $R_{i:j,k:\ell}$ consists of the $i$-th to the $j$-th rows and the $k$-th to the $\ell$-th columns of matrix $R$. The action of the Hessian $\nabla^2 f(M)$ on any two matrices $K$ and $L$ is given by $[\nabla^2 f(M)](K, L) := \sum_{i,j,k,\ell} [\nabla^2 f(M)]_{i,j,k,\ell} K_{ij} L_{k,\ell}$.

In Section 2, the Singular Value Projection algorithm is analyzed as an enlightening example for our main results. Sections 3 and 4 are devoted to the non-existence of spurious second-order critical points and the strict saddle property of the low-rank optimization problem in both symmetric and asymmetric cases, respectively.

## 2 Motivating Example: Singular Value Projection Algorithm

Before providing theoretical results for problems (4)-(5), we first consider the Singular Value Projection Method (SVP) algorithm (Algorithm 1) as a motivating example, which is proposed in Jain et al. (2010). The SVP algorithm is basically the projected gradient method of the original low-rank problems (1)-(2) via the truncated SVD. For the asymmetric problem (2), the low-rank manifold is

$$\mathcal{M}_{asym} := \{M \in \mathbb{R}^{n \times m} \mid \mathrm{rank}(M) \leq r\}$$

and the projection is given by only keeping components corresponding to the $r$ largest singular values. For the symmetric problem (1), the low-rank manifold is

$$\mathcal{M}_{sym} := \{M \in \mathbb{R}^{n \times n} \mid \mathrm{rank}(M) \leq r, \quad M^T = M, \quad M \succeq 0\}.$$

We assume without loss of generality that the gradient $\nabla f(\cdot)$ is symmetric; see Appendix A for a discussion. The projection is given by only keeping components corresponding to the $r$ largest

---

**Algorithm 1** Singular Value Projection (SVP) Algorithm

---

**Input:** Low-rank manifold $\mathcal{M}$, initial point $M_0$, number of iterations $T$, step size $\eta$, objective
    function $f(\cdot)$.
**Output:** Low-rank solution $M_T$.
 1: **for** $t = 0, \ldots, T-1$ **do**
 2:    Update $\tilde{M}_{t+1} \leftarrow M_t - \eta \nabla f(M_t)$.
 3:    Set $M_{t+1}$ to be the projection of $\tilde{M}_{t+1}$ onto $\mathcal{M}$ via truncated SVD.
 4: **end for**
 5: **return** $M_T$.

---

eigenvalues and dropping all components with negative eigenvalues. Since both low-rank manifolds are non-convex, the projection solution may not be unique and we choose an arbitrary solution when it is not unique. We note that the above projections are orthogonal in the sense that

$$\|M_+ - M\|_F = \min_{K \in \mathcal{M}} \|K - M\|_F,$$

where $M_+$ is the projection of a matrix $M$. Henceforth, $\mathcal{M}$ stands for $\mathcal{M}_{sym}$ or $\mathcal{M}_{asym}$, which should be clear from the context. Although each truncated SVD operation can be computed within $O(nmr)$ operations, the constant hidden in the $O(\cdot)$ notation is considerably larger than 1. Thus, the truncated SVD operation is significantly slower than matrix multiplication, which makes the SVP algorithm impractical for large-scale problems. However, the analysis of the SVP algorithm, combining with the equivalence property given in Ha et al. (2020), provides some insights into how to develop proof techniques for problems (4)-(5).

We extend the proof in Jain et al. (2010) and show that Algorithm 1 converges linearly to the global minimum under the $\delta$-RIP$_{2r,2r}$ property with $\delta < 1/3$.

**Theorem 3.** *If function $f_s(\cdot)$ (resp. $f_a(\cdot)$) satisfies the $\delta$-RIP$_{2r,2r}$ property with $\delta < 1/3$ and the step size is chosen to be $\eta = (1+\delta)^{-1}$, then Algorithm 1 applied to problem* (1) *(resp.* (2)*) returns a solution $M_T$ such that $M_T \in \mathcal{M}$ and $f(M_T) - f(M^*) \leq \epsilon$ within*

$$T := \left\lceil \frac{1}{\log[(1-\delta)/(2\delta)]} \cdot \log \left[ \frac{f(M_0) - f(M^*)}{\epsilon} \right] \right\rceil$$

*iterations, where $f(\cdot) := f_s(\cdot)$ (resp. $f(\cdot) := f_a(\cdot)$), $M^*$ is the global minimum, $M_0$ is the initial point and $\lceil \cdot \rceil$ is the ceiling function.*

The proof is almost identical to that in Jain et al. (2010) except that we have replaced the quadratic function with the RIP bounds. However, the result of the proof provides a key inequality (13) for the subsequent proofs. We note that the above proof can be applied to other low-rank optimization problems with a suitable definition of the orthogonal projection. In Ha et al. (2020), it is proved that the unique global minimum is the only fixed point of the SVP algorithm if the RIP constant $\delta$ is less than $1/3$. However, the above paper has not proven the linear convergence (as done in Theorem 3). This difference leads to a strengthened inequality in the following analysis, which further serves as an essential step in proving the strict saddle property. The results in this section provide a hint that the landscape may be benign when the RIP constant is smaller than $1/3$ and we may be able to establish linear convergence under this condition, which is the main topic of the remainder of this paper.

## 3 No Spurious Second-order Critical Points

In this section, we develop necessary conditions and sufficient conditions for the existence of spurious second-order critical points of problems (4)-(5). Besides the non-existence of spurious local minima, the non-existence of spurious second-order critical points also guarantees the global convergence of many first-order algorithms with random initialization under certain regularity conditions (Lee et al., 2016; Panageas & Piliouras, 2016). More precisely, we require the iterates of the algorithm to converge to a single point and the objective function to have a Lipschitz-continuous gradient. The first condition is satisfied by the gradient descent method applied to a large class of functions known as the KŁ-functions (Attouch et al., 2013). For the second condition, many objective functions that appear in applications, e.g., the $\ell_2$-loss function, do not satisfy this condition. However, if the step

size is small enough, the special non-convex structure of the Burer-Monteiro decomposition and the RIP property ensure that the trajectory of the gradient descent method stays in a compact set, where the Lipschitz condition is satisfied due to the second-order continuity of the functions $f_s(\cdot)$ and $f_a(\cdot)$. The proof of this claim is similar to Theorem 8 in Jin et al. (2017) and is omitted here. Therefore, the non-existence of spurious second-order critical points can ensure the global convergence of the gradient descent method for many applications.

The non-existence of spurious second-order critical points has been proved in Ge et al. (2017); Zhu et al. (2018) for problems with linear and nonlinear measurements, respectively. Recently, Ha et al. (2020) proved a relation between the second-order critical points of problem (3) or (5) and the fixed points of the SVP algorithm on problem (2). Using this relation, they showed that problems (3) and (5) have no spurious second-order critical points when the $\delta$-RIP$_{2r,2r}$ property is satisfied with $\delta < 1/3$. In this work, we take a different approach to show that $\delta \leq 1/3$ is enough for the general case in both symmetric and asymmetric scenarios, and that $\delta < 1/2$ is enough for the rank-1 case. Moreover, we prove that there exists a positive correlation between every second-order critical point and the global minimum when $\delta \in (1/3, 1/2)$. We also show that there may exist spurious second-order critical points when $\delta = 1/2$ for both the symmetric and asymmetric problems, which extends the construction of such examples for the linear symmetric rank-1 problem in Zhang et al. (2018) to general cases. We first give necessary conditions and sufficient conditions for the existence of a function that satisfies the $\delta$-RIP$_{2r,2r}$ condition and spurious second-order critical points below.

**Theorem 4.** *Let $\ell := \min\{m, n, 2r\}$. For a given $\delta \in [0, 1)$, there exists a function $f_a(\cdot)$ with the $\delta$-RIP$_{2r,2r}$ property such that problems (3) and (5) have a spurious second-order critical point only if $1 - \delta < (1 + \delta)/2$ and there exists a constant $\alpha \in (1 - \delta, (1 + \delta)/2]$, a diagonal matrix $\Sigma \in \mathbb{R}^{r \times r}$, a diagonal matrix $\Lambda \in \mathbb{R}^{(\ell-r) \times (\ell-r)}$ and matrices $A \in \mathbb{R}^{r \times r}, B \in \mathbb{R}^{r \times r}, C \in \mathbb{R}^{(\ell-r) \times r}, D \in \mathbb{R}^{(\ell-r) \times r}$ such that*

*If $CB^T = 0$ and $AD^T = 0$, then there exists a function $f_a(\cdot)$ with the $\delta$-RIP$_{2r,2r}$ property such that problems (3) and (5) have a spurious second-order critical point.*

$$(1 + \delta) \min_{1 \leq i \leq r} \Sigma_{ii} \geq \max_{1 \leq i \leq \ell-r} \Lambda_{ii}, \quad \Sigma \succ 0, \ \Lambda \succeq 0,$$

$$\langle \Lambda, CD^T \rangle = \alpha \left[ \text{tr}(\Sigma^2) - 2\langle \Sigma, AB^T \rangle + \|AB^T\|_F^2 + \|AD^T\|_F^2 + \|CB^T\|_F^2 + \|CD^T\|_F^2 \right], \quad (6)$$

$$\text{tr}(\Lambda^2) \leq \alpha^{-1}(2\alpha - 1 + \delta^2) \cdot \langle \Lambda, CD^T \rangle, \quad \langle \Lambda, CD^T \rangle \neq 0.$$

*Remark* 1. We note that there may exist simpler forms of the above conditions. For instance, we may solve $\alpha$ via the condition in the second line of (6) and substitute into other conditions. In addition, the requirement that $\alpha \in (1 - \delta, (1 + \delta)/2]$ may also be dropped without affecting the conditions. More specifically, the conditions in (6) are equivalent to

$$(1 + \delta) \min_{1 \leq i \leq r} \Sigma_{ii} \geq \max_{1 \leq i \leq \ell-r} \Lambda_{ii}, \quad \Sigma \succ 0, \ \Lambda \succeq 0, \quad \langle \Lambda, CD^T \rangle \neq 0,$$

$$\text{tr}(\Lambda^2) \leq 2 \cdot \langle \Lambda, CD^T \rangle - (1 - \delta^2) \Big[ \text{tr}(\Sigma^2) - 2\langle \Sigma, AB^T \rangle$$
$$+ \|AB^T\|_F^2 + \|AD^T\|_F^2 + \|CB^T\|_F^2 + \|CD^T\|_F^2 \Big].$$

We state Theorem 4 in the current form since it helps with deriving corollaries more directly.

Intuitively, $\Lambda$ and $\Sigma$ correspond to the singular values of the second-order critical point and the gradient at the second-order critical point, respectively. Matrices $A, B, C, D$ correspond to the SVD of the global optimum. The original problem of the non-existence of spurious second-order critical points can be viewed as a property of the set of functions satisfying the RIP property, which is a convex set in an infinite-dimensional functional space. The conditions in (6) reduce the infinite-dimensional problem to a finite-dimensional problem by utilizing the optimality conditions and the RIP property, which provides a basis for solving these conditions numerically. We note that the conditions in the third line of (6) are novel and serve as an important step in developing strong theoretical guarantees. Although the conditions in (6) seem complicated, they lead to strong results on the non-existence of spurious second-order critical points. We provide two corollaries below to illustrate the power of the above theorem. The first corollary focuses on the rank-1 case. In this case, we can simplify condition

(6) through suitable relaxations to obtain a sharper bound on $\delta$ that ensures the non-existence of spurious second-order critical points.

**Corollary 1.** *Consider the case $r = 1$, and suppose that the function $f_a(\cdot)$ satisfies the $\delta$-RIP$_{2,2}$ property with $\delta < 1/2$. Then, problems* (3) *and* (5) *have no spurious second-order critical points.*

The following example shows that the counterexample in Zhang et al. (2019) designed for the symmetric case also works for the asymmetric rank-1 case.

**Example 1.** We note that Example 12 in Zhang et al. (2019) shows that problem (4) may have spurious second-order critical points when $\delta = 1/2$. In general, a second-order critical point for problem (4) is not a second-order critical point for problem (5), since the asymmetric manifold $\mathcal{M}_{asym}$ has a larger second-order critical cone than the symmetric manifold $\mathcal{M}_{sym}$. However, it can be verified that the same example also has a spurious second-order critical point in the asymmetric case. For completeness, we verify the claim in the appendix.

It follows from Corollary 1 and Example 1 that the bound $1/2$ is the *sharpest* bound for the rank-1 asymmetric case. The next corollary provides a marginal improvement to the state-of-the-art result for the general rank case, which derives the RIP bound $\delta < 1/3$ (Ha et al., 2020). In addition, we prove that there exists a positive correlation between every second-order critical point and the global minimum when $\delta < 1/2$.

**Corollary 2.** *Given an arbitrary $r$, suppose that the function $f_a(\cdot)$ satisfies the $\delta$-RIP$_{2r,2r}$ property. If $\delta \leq 1/3$, then both problems* (3) *and* (5) *have no spurious second-order critical points. In addition, if $\delta \in [0, 1/2)$, then every second-order critical point $\tilde{M}$ has a positive correlation with the ground truth $M_a^*$. Namely, there exists a universal function $C(\delta) : (0, 1/2) \mapsto (0, 1]$ such that*

$$\langle \tilde{M}, M_a^* \rangle \geq C(\delta) \cdot \|\tilde{M}\|_F \|M_a^*\|_F.$$

For the general rank-$r$ case, we construct a counterexample with spurious second-order critical points when $\delta = 1/2$.

**Example 2.** Let $n = m = 2r$. Now, we use the sufficiency part of Theorem 4 to construct a counterexample. We choose

$$\delta := \frac{1}{2}, \quad \alpha := \frac{3}{5}, \quad \Sigma := \frac{1}{2}I_r, \quad \Lambda := \frac{3}{4}I_r, \quad A = B := 0_r, \quad C = D := I_r.$$

It can be verified that the conditions in (6) are satisfied and $CB^T = AD^T = 0$, which means that there exists a function $f_a(\cdot)$ satisfying the $\delta$-RIP$_{2r,2r}$ property for which problems (3) and (5) have spurious second-order critical points. We also give a direct construction with linear measurements in the appendix. This example illustrates that Theorem 4 can be used to systematically design instances of the problem with spurious second-order critical points.

Before closing this section, we note that similar conditions can be obtained for the symmetric problem (4). Although there exists a natural transformation of symmetric problems to asymmetric problems (see the appendix), the approach requires the objective function $f_s(\cdot)$ to have the $\delta$-RIP$_{4r,2r}$ property, which provides sub-optimal RIP bounds compared to a direct analysis. We give the results of the direct analysis below and omit the proof due to the similarity to the asymmetric case.

**Theorem 5.** *Let $\ell := \min\{n, 2r\}$. For a given $\delta \in [0, 1)$, there exists a function $f_s(\cdot)$ with the $\delta$-RIP$_{2r,2r}$ property such that problem* (4) *has a spurious second-order critical point only if $1 - \delta < (1 + \delta)/2$ and there exists a constant $\alpha \in (1 - \delta, (1 + \delta)/2]$, a diagonal matrix $\Sigma \in \mathbb{R}^{r \times r}$, a diagonal matrix $\Lambda \in \mathbb{R}^{(\ell-r) \times (\ell-r)}$ and matrices $A \in \mathbb{R}^{r \times r}, C \in \mathbb{R}^{(\ell-r) \times r}$ such that*

$$(1 + \delta) \min_{1 \leq i \leq r} \Sigma_{ii} \geq \max_{1 \leq i \leq \ell-r} \Lambda_{ii}, \quad \Sigma \succ 0,$$

$$\langle \Lambda, CC^T \rangle = \alpha \left[ \text{tr}(\Sigma^2) - 2\langle \Sigma, AA^T \rangle + \|AA^T\|_F^2 + 2\|AC^T\|_F^2 + \|CC^T\|_F^2 \right], \quad (7)$$

$$\text{tr}(\Lambda^2) \leq \alpha^{-1}(2\alpha - 1 + \delta^2) \cdot \langle \Lambda, CC^T \rangle, \quad \langle \Lambda, CC^T \rangle \neq 0.$$

*If $AC^T = 0$, then there exists a function $f_s(\cdot)$ with the $\delta$-RIP$_{2r,2r}$ property for which problem* (4) *has a spurious second-order critical point.*

Compared to Theorem 4, the diagonal matrix $\Lambda$ is not enforced to be positive semi-definite. The reason is that the eigenvalue decomposition is used instead of the singular value decomposition in the symmetric case, and therefore some eigenvalues can be negative. Similarly, we can obtain the non-existence and the positive correlation results for the symmetric problem.

**Corollary 3.** *If function $f_s(\cdot)$ satisfies the $\delta$-RIP$_{2r,2r}$ property, then the following statements hold:*

- *If $\delta \leq 1/3$, then there are no spurious second-order critical points;*

- *If $\delta < 1/2$, then there exists a positive correlation between every second-order critical point and the ground truth;*

- *If $\delta = 1/2$, then there exists a counterexample with spurious second-order critical points;*

- *If $\delta < 1/2$ and $r = 1$, then there are no spurious second-order critical points.*

We note that the last statement serves as a generalization of the results in Zhang et al. (2019) to the nonlinear measurement case, and improves upon the bound in Bi & Lavaei (2021) by dropping the BDP constant.

## 4 Global Landscape: Strict Saddle Property

Although the non-existence of spurious second-order critical points can ensure the global convergence under certain regularity conditions, it cannot guarantee a fast convergence rate in general. Saddle-point escaping algorithms may become stuck at approximate second-order critical points for exponentially long time. To guarantee the global polynomial-time convergence, the following strict saddle property is commonly considered in the literature:

**Definition 3** (Sun et al. (2018))**.** Consider an arbitrary optimization problem $\min_{x \in \mathcal{X} \subset \mathbb{R}^d} F(x)$ and let $\mathcal{X}^*$ denote the set of its global minima. It is said that the problem satisfies the $(\alpha, \beta, \gamma)$-**strict saddle property** for $\alpha, \beta, \gamma > 0$ if at least one of the following conditions is satisfied for every $x \in \mathcal{X}$:
$$\text{dist}(x, \mathcal{X}^*) \leq \alpha; \quad \|\nabla F(x)\|_F \geq \beta; \quad \lambda_{min}[\nabla^2 F(x)] \leq -\gamma.$$

For the low-rank problems, we choose the distance to be the Frobenius norm in the factorization space. This distance is equivalent to the Frobenius norm in the matrix space in the sense that there exist constants $c_1(\mathcal{X}^*) > 0$ and $c_2(\mathcal{X}^*) > 0$ such that
$$c_1(\mathcal{X}^*) \cdot \|U - U^*\|_F \leq \|UU^T - U^*(U^*)^T\|_F \leq c_2(\mathcal{X}^*) \cdot \|U - U^*\|_F$$
holds for all $U \in \mathcal{X}$ as long as $\|U - U^*\|_F$ is small and $\mathcal{X}^*$ is bounded (Tu et al., 2016). A similar relation holds for the asymmetric case.

In Jin et al. (2017), it has been proved that the perturbed gradient descent method can find an $\epsilon$-approximate second-order critical point in $\tilde{O}(\epsilon^{-2})$ iterations with high probability if the Hessian of the objective function is Lipschitz. Namely, the algorithm returns a point $x \in \mathcal{X}$ such that
$$\|\nabla F(x)\|_F \leq O(\epsilon), \quad \lambda_{min}[\nabla^2 F(x)] \geq -O(\sqrt{\epsilon})$$
in $\tilde{O}(\epsilon^{-2})$ iterations with high probability. If we choose $\epsilon > 0$ to be small enough such that $O(\epsilon) < \beta$ and $-O(\sqrt{\epsilon}) > -\gamma$, then the strict saddle property ensures that the returned point satisfies $\text{dist}(x, \mathcal{X}^*) \leq \alpha$ with high probability. We note that the Lipschitz continuity of the Hessian can be similarly guaranteed by the boundedness of trajectories of the perturbed gradient method, which can be proved similarly as Theorem 8 in Jin et al. (2017). Since the smoothness properties are satisfied over a bounded region, we may apply the perturbed gradient descent method (Jin et al., 2017) to achieve the polynomial-time global convergence with random initialization.

In this section, we prove that problems (4) and (5) satisfy the strict saddle property with an arbitrary $\alpha > 0$ in the exact parameterization case, i.e., when the global optimum has rank $r$.

**Assumption 2.** The global optimum $M_a^*$ or $M_s^*$ has rank $r$.

It has been proved in Zhu et al. (2021) that the regularized problem (5) satisfies the strict saddle property if the function $f_a(\cdot)$ has the $\delta$-RIP$_{2r,4r}$ property with
$$\delta < \frac{\sigma_r(M_a^*)^{3/2}}{100\|M_a^*\|_F\|M_a^*\|_2^{1/2}}.$$

Our results improve upon their bounds by allowing a larger problem-free RIP constant and requiring only the RIP$_{2r,2r}$ property (note that there are problems with RIP$_{2r,2r}$ property for which the RIP$_{2r,4r}$ property does not hold (Bi & Lavaei, 2021)). Our result can also be viewed as a robust version of the results in Ha et al. (2020).

**Theorem 6.** *Suppose that the function $f_a(\cdot)$ satisfies the $\delta$-RIP$_{2r,2r}$ property with $\delta < 1/3$. Given an arbitrary constant $\alpha > 0$, if $\mu$ is selected to belong to the interval $[(1-\delta)/3, 1-\delta)$, then there exist positive constants*

$$\epsilon_1 := \epsilon_1(\delta, r, \mu, \sigma_r(M_a^*), \|M_a^*\|_F, \alpha), \quad \lambda_1 := \lambda_1(\delta, r, \mu, \sigma_r(M_a^*), \|M_a^*\|_F, \alpha)$$

*such that for every $\epsilon \in (0, \epsilon_1]$ and $\lambda \in (0, \lambda_1]$, problem (5) satisfies the $(\alpha, \beta, \gamma)$-strict saddle property with*

$$\beta := \min \left\{ \mu(\epsilon/r)^{3/2}, \lambda \right\}, \quad \gamma := \mu\epsilon.$$

We note that the constraint $\mu \in [(1-\delta)/3, 1-\delta)$ is not optimal and it can be similarly proved that $\mu \in (\delta, 1-\delta)$ also guarantees the strict saddle property. The key step in the proof is to show that for every point $(U, V)$ at which the gradient of $f_a(UV^T)$ is small, it holds that

$$\|\nabla f_a(UV^T)\|_2^2 \geq (1+\delta)^2 \sigma_r^2(UV^T) + C \cdot (1 - 3\delta)[f_a(UV^T) - f_a(M_a^*)],$$

where $C > 0$ is a constant independent of $(U, V)$. This inequality can be viewed as a major extension of the non-existence of spurious second-order critical points when $\delta < 1/3$ (Ha et al., 2020), which shows that every spurious second-order critical point $(U, V)$ satisfies

$$\|\nabla f_a(UV^T)\|_2^2 > (1+\delta)\sigma_r^2(UV^T).$$

We emphasize that our proof requires a new framework and is not a standard revision of the existing methods, which is the reason why sharper bounds can be established. By replacing $\|\nabla f_a(M)\|_2$ with $-\lambda_{min}(\nabla f_s(M))$, the analysis for the asymmetric case can be extended to the symmetric case with minor modifications and the same bound follows.

**Theorem 7.** *Suppose that the function $f_s(\cdot)$ satisfies the $\delta$-RIP$_{2r,2r}$ property with $\delta < 1/3$. Given an arbitrary constant $\alpha > 0$, there exists a positive constant $\lambda_1 := \lambda_1(\delta, r, \sigma_r(M_s^*), \|M_s^*\|_F, \alpha)$ such that for every $\lambda \in (0, \lambda_1]$, problem (4) satisfies the $(\alpha, \beta, \gamma)$-strict saddle property with*

$$\beta := \lambda, \quad \gamma := 2\lambda.$$

The above bound is the first theoretical guarantee of the strict saddle property for the nonlinear symmetric problem.

## 5   Conclusion

In this work, we analyze the geometric properties of low-rank optimization problems via the non-convex factorization approach. We prove novel necessary conditions and sufficient conditions for the non-existence of spurious second-order critical points in both symmetric and asymmetric cases. We show that these conditions lead to sharper bounds and greatly simplify the construction of counterexamples needed to study the sharpness of the bounds. The developed bounds significantly generalize several of the existing results. In the rank-1 case, the bound is proved to be the sharpest possible. In the general rank case, we show that there exists a positive correlation between second-order critical points and the global minimum for problems whose RIP constants are higher than the developed bound but lower than the fundamental limit obtained by the counterexamples. Finally, the strict saddle property is proved with a weaker requirement on the RIP constant for asymmetric problems. The paper develops the first strict saddle property in the literature for nonlinear symmetric problems.

### Acknowledgments and Disclosure of Funding

This work was supported by grants from AFOSR, ARO, ONR, NSF and C3.ai Digital Transformation Institute.

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
