# A  Optimality Conditions

In this section, we develop the optimality conditions for problems (1)-(5). We assume without loss of generality that $\nabla f_s(M)$ is symmetric for every $M \in \mathbb{R}^{n \times n}$. This is because we can always optimize the equivalent problem

$$\min_{M \in \mathbb{R}^{n \times n}} \frac{1}{2} \left[ f_s(M) + f_s(M^T) \right] \quad \text{s.t. } \text{rank}(M) \leq r, \quad M^T = M, \quad M \succeq 0.$$

We first consider problems (1) and (2).

**Theorem 8** (Li et al. (2019); Ha et al. (2020)). *The matrix $\tilde{M} = \tilde{U}\tilde{U}^T$ with $\tilde{U} \in \mathbb{R}^{n \times r}$ is a first-order critical point of the constrained problem* (1) *if and only if*

$$\begin{cases} \nabla f_s(\tilde{M})\tilde{U} = 0 & \text{if } \text{rank}(\tilde{M}) = r \\ \nabla f_s(\tilde{M}) \succeq 0 & \text{if } \text{rank}(\tilde{M}) < r. \end{cases}$$

*The matrix $\tilde{M} = \tilde{U}\tilde{V}^T$ with $\tilde{U} \in \mathbb{R}^{n \times r}$ and $\tilde{V} \in \mathbb{R}^{m \times r}$ is a first-order critical point of the constrained problem* (2) *if and only if*

$$\begin{cases} [\nabla f_a(\tilde{M})]^T \tilde{U} = 0, \ \nabla f_a(\tilde{M})\tilde{V} = 0 & \text{if } \text{rank}(\tilde{M}) = r \\ \nabla f_a(\tilde{M}) = 0 & \text{if } \text{rank}(\tilde{M}) < r. \end{cases}$$

In Ha et al. (2020), the authors proved that each second-order critical point of problem (3) or (5) is a fixed point of the SVP algorithm run on problem (2). We note that this relation can be extended to the symmetric and positive semi-definite case. This relation plays an important role in the analysis of Section 3.

**Theorem 9** (Ha et al. (2020)). *The matrix $\tilde{M} = \tilde{U}\tilde{U}^T$ with $\tilde{U} \in \mathbb{R}^{n \times r}$ is a fixed point of the SVP algorithm run on problem* (1) *with the step size $1/(1+\delta)$ if and only if*

$$\nabla f_s(\tilde{M})\tilde{U} = 0, \quad -\lambda_{min}(\nabla f_s(\tilde{M})) \leq (1+\delta)\sigma_r(\tilde{U}).$$

*The matrix $\tilde{M} = \tilde{U}\tilde{V}^T$ with $\tilde{U} \in \mathbb{R}^{n \times r}$ and $\tilde{V} \in \mathbb{R}^{m \times r}$ is a fixed point of the SVP algorithm run on problem* (2) *with the step size $1/(1+\delta)$ if and only if*

$$[\nabla f_a(\tilde{M})]^T \tilde{U} = 0, \quad \nabla f_a(\tilde{M})\tilde{V} = 0, \quad \|\nabla f_a(\tilde{M})\|_2 \leq (1+\delta)\sigma_r(\tilde{M}).$$

Next, we consider problems (3)-(5). Since the goal is to study only spurious local minima and saddle points, it is enough to focus on the second-order necessary optimality conditions. The following two theorems follow from basic calculations and we omit the proof.

**Theorem 10.** *The matrix $\tilde{U} \in \mathbb{R}^{n \times r}$ is a second-order critical point of problem* (4) *if and only if*

$$\nabla f_s(\tilde{U}\tilde{U}^T)\tilde{U} = 0$$

*and*

$$2\langle \nabla f_s(\tilde{U}\tilde{U}^T), \Delta\Delta^T \rangle + [\nabla^2 f_s(\tilde{U}\tilde{U}^T)](\tilde{U}\Delta^T + \Delta\tilde{U}^T, \tilde{U}\Delta^T + \Delta\tilde{U}^T) \geq 0$$

*holds for every $\Delta \in \mathbb{R}^{n \times r}$.*

**Theorem 11.** *The point $(\tilde{U}, \tilde{V})$ with $\tilde{U} \in \mathbb{R}^{n \times r}$ and $\tilde{V} \in \mathbb{R}^{m \times r}$ is a second-order critical point of problem* (3) *if and only if*

$$\nabla[f_a(\tilde{U}\tilde{V}^T)]^T \tilde{U} = 0, \quad \nabla f_a(\tilde{U}\tilde{V}^T)\tilde{V} = 0$$

*and*

$$2\langle \nabla f_a(\tilde{U}\tilde{V}^T), \Delta_U \Delta_V^T \rangle + [\nabla^2 f_a(\tilde{U}\tilde{V}^T)](\tilde{U}\Delta_V^T + \Delta_U \tilde{V}^T, \tilde{U}\Delta_V^T + \Delta_U \tilde{V}^T) \geq 0$$

*holds for every $\Delta_U \in \mathbb{R}^{n \times r}$ and $\Delta_V \in \mathbb{R}^{m \times r}$. Moreover, the given point is a a second-order critical point of problem* (5) *if and only if*

$$\nabla[f_a(\tilde{U}\tilde{V}^T)]^T \tilde{U} = 0, \quad \nabla f_a(\tilde{U}\tilde{V}^T)\tilde{V} = 0, \quad \tilde{U}^T\tilde{U} = \tilde{V}^T\tilde{V}$$

*and*

$$2\langle \nabla f_a(\tilde{U}\tilde{V}^T), \Delta_U \Delta_V^T \rangle + [\nabla^2 f_a(\tilde{U}\tilde{V}^T)](\tilde{U}\Delta_V^T + \Delta_U \tilde{V}^T, \tilde{U}\Delta_V^T + \Delta_U \tilde{V}^T)$$
$$+ \frac{\mu}{2}\|\tilde{U}^T\Delta_U + \Delta_U^T\tilde{U} - \tilde{V}^T\Delta_V - \Delta_V^T\tilde{V}\|_F^2 \geq 0$$

*holds for every $\Delta_U \in \mathbb{R}^{n \times r}$ and $\Delta_V \in \mathbb{R}^{m \times r}$.*

# B  Relation between the Symmetric and Asymmetric Problems

In this section, we study the relationship between problems (4)-(5). This relationship is more general than the topic of this paper, namely the non-existence of spurious second-order critical points and the strict saddle property, and holds for any property that is characterized by the RIP constant $\delta$ and the BDP constant $\kappa$. Specifically, we show that any property that holds for the symmetric problems (4) with $(\delta, \kappa)$ also holds for the regularized asymmetric problem (5) with another pair of constants $(\tilde{\delta}, \tilde{\kappa})$ decided by $\delta, \kappa$, and vice versa.

We first consider the transformation from the asymmetric case to the symmetric case. The transformation to the symmetric case has been established in Ge et al. (2017) for linear problem. Here, we show that the transformation can be revised and extended to the nonlinear measurements case.

**Theorem 12.** *Suppose that the function $f_a(\cdot)$ satisfies the $\delta$-RIP$_{2r,2s}$ and the $\kappa$-BDP$_{2t}$ properties. If we choose $\mu := (1 - \delta)/2$, then problem (5) is equivalent to a symmetric problem whose objective function satisfies the $2\delta/(1+\delta)$-RIP$_{2r,2s}$ and the $2\kappa/(1+\delta)$-BDP$_{2t}$ properties.*

*Proof of Theorem 12.* For any matrix $N \in \mathbb{R}^{(n+m) \times (n+m)}$, we divide the matrix into four blocks as

$$N = \begin{bmatrix} N_{11} & N_{12} \\ N_{21} & N_{22} \end{bmatrix},$$

where $N_{11} \in \mathbb{R}^{n \times n}, N_{12} \in \mathbb{R}^{n \times m}, N_{22} \in \mathbb{R}^{m \times m}$. Then, we define a new function

$$\tilde{f}(N) := f_a(N_{12}) + f_a(N_{21}^T).$$

We observe that $\tilde{f}(WW^T) = 2h_a(U, V)$, where

$$W := \begin{bmatrix} U \\ V \end{bmatrix} \in \mathbb{R}^{(n+m) \times r}.$$

For any $K \in \mathbb{R}^{(n+m) \times (n+m)}$, the Hessian of $\tilde{f}(\cdot)$ satisfies

$$[\nabla^2 \tilde{f}(N)](K, K) = [\nabla^2 f_a(N_{12})](K_{12}, K_{12}) + [\nabla^2 f_a(N_{21}^T)](K_{21}^T, K_{21}^T). \tag{8}$$

Similarly, we can define

$$\tilde{g}(N) := \|N_{11}\|_F^2 + \|N_{22}\|_F^2 - \|N_{12}\|_F^2 - \|N_{21}\|_F^2.$$

We can also verify that $\tilde{g}(WW^T) = g(U, V)$ and

$$[\nabla^2 \tilde{g}(N)](K, K) = 2 \left( \|K_{11}\|_F^2 + \|K_{22}\|_F^2 - \|K_{12}\|_F^2 - \|K_{21}\|_F^2 \right). \tag{9}$$

for every $K \in \mathbb{R}^{(n+m) \times (n+m)}$. The minimization problem (5) is then equivalent to

$$\min_{W \in \mathbb{R}^{(n+m) \times r}} F(WW^T) := \tilde{f}(WW^T) + \frac{\mu}{2} \cdot \tilde{g}(WW^T), \tag{10}$$

which is in the symmetric form as problem (4). For every $N, K \in \mathbb{R}^{(n+m) \times (n+m)}$ with $\text{rank}(N) \leq 2r$ and $\text{rank}(K) \leq 2s$, it results from relations (8) and (9) that

$$\begin{aligned}
&[\nabla^2 F(N)](K, K) \\
&\quad \geq (1 - \delta) \left( \|K_{12}\|_F^2 + \|K_{21}\|_F^2 \right) + \mu \left( \|K_{11}\|_F^2 + \|K_{22}\|_F^2 - \|K_{12}\|_F^2 - \|K_{21}\|_F^2 \right) \\
&\quad \geq \min\{1 - \delta - \mu, \mu\} \cdot \|K\|_F^2
\end{aligned}$$

and

$$\begin{aligned}
&[\nabla^2 F(N)](K, K) \\
&\quad \leq (1 + \delta) \left( \|K_{12}\|_F^2 + \|K_{21}\|_F^2 \right) + \mu \left( \|K_{11}\|_F^2 + \|K_{22}\|_F^2 - \|K_{12}\|_F^2 - \|K_{21}\|_F^2 \right) \\
&\quad \leq \max\{1 + \delta - \mu, \mu\} \cdot \|K\|_F^2.
\end{aligned}$$

Choosing $\mu := (1 - \delta)/2$, we obtain

$$\frac{1 - \delta}{2} \cdot \|K\|_F^2 \leq [\nabla^2 F(N)](K, K) \leq \frac{1 + 3\delta}{2} \cdot \|K\|_F^2.$$

Hence, it follows that the function $2F(\cdot)/(1+\delta)$ satisfies the $2\delta/(1+\delta)$-RIP$_{2r,2s}$ property.

Moreover, for every $N, N', K, L \in \mathbb{R}^{(n+m)\times(n+m)}$ with

$$\mathrm{rank}(N), \mathrm{rank}(N'), \mathrm{rank}(K), \mathrm{rank}(L) \leq 2t,$$

it holds that

$$
\begin{aligned}
[\nabla^2 \tilde{g}(N)](K, L) &= [\nabla^2 \tilde{g}(N')](K, L) \\
&= 2\left(\langle K_{11}, L_{11}\rangle + \langle K_{22}, L_{22}\rangle - \langle K_{12}, L_{12}\rangle - \langle K_{21}, L_{21}\rangle\right)
\end{aligned}
$$

and

$$
\begin{aligned}
&\left|[\nabla^2 F(N) - \nabla^2 F(N')](K, L)\right| \\
&= \left|[\nabla^2 f(N_{12}) - \nabla^2 f(N'_{12})](K_{12}, L_{12}) + [\nabla^2 f(N_{21}^T) - \nabla^2 f((N'_{21})^T)](K_{21}^T, L_{21}^T)\right| \\
&\leq \kappa \|K_{12}\|_F \|L_{12}\|_F + \kappa \|K_{21}\|_F \|L_{21}\|_F \leq \kappa \|K\|_F \|L\|_F,
\end{aligned}
$$

which implies that the function $\frac{2}{1+\delta} \cdot F(\cdot)$ satisfies the $2\kappa/(1+\delta)$-BDP$_{2r}$ property. Since problem (10) is equivalent to the minimization of $\frac{2}{1+\delta} \cdot F(WW^T)$, it is equivalent to a symmetric problem that satisfies the $2\delta/(1+\delta)$-RIP$_{2r,2s}$ and the $2\kappa/(1+\delta)$-BDP$_{2r}$ properties. $\qquad\square$

We can see that both constants $\delta$ and $\kappa$ are approximately doubled in the transformation. As an example, Bhojanapalli et al. (2016b) showed that the symmetric linear problem has no spurious local minima if the $\delta$-RIP$_{2r}$ property is satisfied with $\delta < 1/5$. Using Theorem 12, we know that the asymmetric linear problem has no spurious local minima if the $\delta$-RIP$_{2r}$ property is satisfied with $\delta < 1/9$.

The transformation from a symmetric problem to an asymmetric problem is more straightforward. We can equivalently solve the optimization problem

$$\min_{U,V\in\mathbb{R}^{n\times r}} \quad f_s\left[\frac{1}{2}\left(UV^T + VU^T\right)\right] \tag{11}$$

or its regularized version with any parameter $\mu > 0$. It can be easily shown that the above problem has the same RIP and BDP constants as the original symmetric problem. We omit the proof for brevity.

**Theorem 13.** *Suppose that the function $f_s(\cdot)$ satisfies the $\delta$-RIP$_{4r,2s}$ and the $\kappa$-BDP$_{4t}$ properties. For every $\mu > 0$, problem (4) is equivalent to an asymmetric problem and its regularized version with the $\delta$-RIP$_{2r,2s}$ and the $\kappa$-BDP$_{2t}$ properties.*

Note that the transformation from a symmetric problem to an asymmetric problem will not increase the constants $\kappa$ and $\delta$ but requires stronger RIP and BDP properties. Hence, a direct analysis on the symmetric case may establish the same property under a weaker condition. In addition to problem (11), we can also directly consider the problem $\min_{U,V} f_a(UV^T)$. However, in certain applications, the objective function is only defined for symmetric matrices and we can only use the formulation (11) to construct an asymmetric problem. In more restricted cases when the objective function is only defined for symmetric and positive semi-definite matrices, we can only apply the direct analysis to the symmetric case.

## C   Proofs for Section 2

### C.1   Proof of Theorem 3

*Proof of Theorem 3.* We denote $f(\cdot) := f_s(\cdot)$ and $f(\cdot) := f_a(\cdot)$ for the symmetric and asymmetric case, respectively. Using the mean value theorem and the $\delta$-RIP$_{2r,2r}$ property, there exists a constant $s \in [0, 1]$ such that

$$
\begin{aligned}
&f(M_{t+1}) - f(M_t) \\
&= \langle \nabla f(M_t), M_{t+1} - M_t\rangle + \frac{1}{2}[\nabla^2 f(M_t + s(M_{t+1} - M_t))](M_{t+1} - M_t, M_{t+1} - M_t)
\end{aligned}
$$

$$\leq \langle \nabla f(M_t), M_{t+1} - M_t \rangle + \frac{1+\delta}{2} \|M_{t+1} - M_t\|_F^2.$$

We define

$$\phi_t(M) := \langle \nabla f(M_t), M - M_t \rangle + \frac{1+\delta}{2} \|M - M_t\|_F^2 = \frac{1+\delta}{2} \|M - \tilde{M}_{t+1}\|_F^2 + constant,$$

where the last constant term is independent of $M$. Since the projection is orthogonal, the projected matrix $M_{t+1}$ achieves the minimal value of $\phi_t(M)$ over all matrices on the manifold $\mathcal{M}$. Therefore, we obtain

$$f(M_{t+1}) - f(M_t) \leq \phi_t(M_{t+1}) \leq \phi_t(M^*)$$
$$= \langle \nabla f(M_t), M^* - M_t \rangle + \frac{1+\delta}{2} \|M^* - M_t\|_F^2. \tag{12}$$

On the other hand, we can similarly prove that the $\delta$-RIP$_{2r,2r}$ property ensures

$$f(M^*) - f(M_t) \geq \langle \nabla f(M_t), M^* - M_t \rangle + \frac{1-\delta}{2} \|M^* - M_t\|_F^2,$$

$$f(M_t) - f(M^*) \geq \frac{1-\delta}{2} \|M^* - M_t\|_F^2.$$

Substituting the above two inequalities into (12), it follows that

$$f(M_{t+1}) - f(M_t) \leq f(M^*) - f(M_t) + \delta \|M^* - M_t\|_F^2$$
$$\leq f(M^*) - f(M_t) + \frac{2\delta}{1-\delta} [f(M_t) - f(M^*)]. \tag{13}$$

Therefore, using the condition that $\delta < 1/3$, we have

$$f(M_{t+1}) - f(M^*) \leq \frac{2\delta}{1-\delta} [f(M_t) - f(M^*)] := \alpha [f(M_t) - f(M^*)],$$

where $\alpha := 2\delta/(1-\delta) < 1$. Combining this single-step bound with the induction method proves the linear convergence of Algorithm 1. □

# D  Proofs for Section 3

## D.1  Proof of Theorem 4

*Proof of Theorem 4.* We only consider the case when $m$ and $n$ are at least $2r$. In this case, we have $\ell = 2r$. Other cases can be handled similarly. For the notational simplicity, we denote $M^* := M_a^*$ in this proof.

**Necessity.** We first consider problem (3). Suppose that $M^*$ and $\tilde{M}$ are the optimum and a spurious second-order critical point of problem (3), respectively. It has been proved in Ha et al. (2020) that the spurious second-order critical point $\tilde{M}$ has rank $r$ and is a fixed point of the SVP algorithm with the step size $(1+\delta)^{-1}$. Therefore, the point $\tilde{M}$ should be a minimizer of the projection step of the SVP algorithm. This implies that

$$\|\tilde{M} - [\tilde{M} - (1+\delta)^{-1}\nabla f_a(\tilde{M})]\|_F^2 \leq \|M^* - [\tilde{M} - (1+\delta)^{-1}\nabla f_a(\tilde{M})]\|_F^2,$$

which can be simplified to

$$\langle \nabla f_a(\tilde{M}), \tilde{M} - M^* \rangle \leq \frac{1+\delta}{2} \|\tilde{M} - M^*\|_F^2. \tag{14}$$

Let $\mathcal{U}$ and $\mathcal{V}$ denote the subspaces spanned by the columns and rows of $\tilde{M}$ and $M^*$, respectively. Namely, we have

$$\mathcal{U} := \{\tilde{M}v_1 + M^*v_2 \mid v_1, v_2 \in \mathbb{R}^m\}, \quad \mathcal{V} := \{\tilde{M}^T u_1 + (M^*)^T u_2 \mid u_1, u_2 \in \mathbb{R}^n\}.$$

Since the ranks of both matrices are bounded by $r$, the dimensions of $\mathcal{U}$ and $\mathcal{V}$ are bounded by $2r$. Therefore, we can find orthogonal matrices $U \in \mathbb{R}^{n \times 2r}$ and $V \in \mathbb{R}^{m \times 2r}$ such that

$$\mathcal{U} \subset \text{range}(U), \quad \mathcal{V} \subset \text{range}(V)$$

and write $\tilde{M}, M^*$ in the form

$$\tilde{M} = U \begin{bmatrix} \Sigma & 0_{r \times r} \\ 0_{r \times r} & 0_{r \times r} \end{bmatrix} V^T, \quad M^* = U R V^T,$$

where $\Sigma \in \mathbb{R}^{r \times r}$ is a diagonal matrix and $R \in \mathbb{R}^{2r \times 2r}$ has rank at most $r$. Recalling the first condition in Theorem 11, the column space and the row space of $\nabla f_a(\tilde{M})$ are orthogonal to the column space and the row space of $\tilde{M}$, respectively. Then, the $\delta$-RIP$_{2r,2r}$ property gives

$$\exists \alpha \in [1 - \delta, 1 + \delta] \quad \text{s.t.} \ -\langle \nabla f_a(\tilde{M}), M^* \rangle = \langle \nabla f_a(\tilde{M}), \tilde{M} - M^* \rangle$$
$$= \int_0^1 [\nabla^2 f_a(M^* + s(\tilde{M} - M^*))](\tilde{M} - M^*, \tilde{M} - M^*) \, ds$$
$$= \alpha \| \tilde{M} - M^* \|_F^2 > 0. \tag{15}$$

This means that

$$G := \mathcal{P}_U \nabla f_a(\tilde{M}) \mathcal{P}_V \neq 0,$$

where $\mathcal{P}_U$ and $\mathcal{P}_V$ are the orthogonal projections onto $\mathcal{U}$ and $\mathcal{V}$, respectively. Combining with inequality (14), we obtain $\alpha \leq (1 + \delta)/2$. By the definition of $G$, we have

$$\langle \nabla f_a(\tilde{M}), M^* \rangle = \langle G, M^* \rangle.$$

Since both the column space and the row space of $G$ are orthogonal to $\tilde{M}$, the matrix $G$ has the form

$$G = U \begin{bmatrix} 0_{r \times r} & 0_{r \times r} \\ 0_{r \times r} & -\Lambda \end{bmatrix} V^T, \tag{16}$$

where $\Lambda \in \mathbb{R}^{r \times r}$. We may assume without loss of generality that $\Lambda_{ii} \geq 0$ for all $i$; otherwise, one can flip the sign of some of the last $r$ columns of $U$. By another orthogonal transformation, we may assume without loss of generality that $\Lambda$ is a diagonal matrix. Then, Theorem 9 gives

$$(1 + \delta) \min_{1 \leq i \leq r} \Sigma_{ii} = (1 + \delta)\sigma_r(\tilde{M}) \geq \|\nabla f_a(\tilde{M})\|_2 \geq \|G\|_2 = \max_{1 \leq i \leq (\ell - r)} \Lambda_{ii}. \tag{17}$$

In addition, condition (15) is equivalent to

$$\langle \Lambda, R_{r+1:2r,r+1:2r} \rangle = \alpha \| \tilde{M} - M^* \|_F^2 = \alpha \left[ \text{tr}(\Sigma^2) - 2\langle \Sigma, R_{1:r,1:r} \rangle + \|R\|_F^2 \right]. \tag{18}$$

By the Taylor expansion, for every $Z \in \mathbb{R}^{n \times m}$, we have

$$\langle \nabla f_a(\tilde{M}), Z \rangle = \int_0^1 [\nabla^2 f_a(M^* + s(\tilde{M} - M^*))](\tilde{M} - M^*, Z) \, ds = (\tilde{M} - M^*) : \mathcal{H} : Z,$$

where the last expression is the tensor multiplication and $\mathcal{H}$ is the tensor such that

$$K : \mathcal{H} : L = \int_0^1 [\nabla^2 f_a(M^* + s(\tilde{M} - M^*))](K, L) \, ds, \quad \forall K, L \in \mathbb{R}^{n \times m}.$$

We define

$$\tilde{G} := G - \alpha(\tilde{M} - M^*).$$

By the definition of $\alpha$, we know that $\langle \tilde{G}, \tilde{M} - M^* \rangle = 0$. Furthermore, using the definition of $\mathcal{H}$, we obtain

$$(\tilde{M} - M^*) : \mathcal{H} : (\tilde{M} - M^*) = \alpha \| \tilde{M} - M^* \|_F^2,$$
$$(\tilde{M} - M^*) : \mathcal{H} : \tilde{G} = \tilde{G} : \mathcal{H} : (\tilde{M} - M^*) = \|\tilde{G}\|_F^2.$$

Suppose that

$$\tilde{G} : \mathcal{H} : \tilde{G} = \beta \|\tilde{G}\|_F^2$$

for some $\beta \in [1 - \delta, 1 + \delta]$. We consider matrices of the form

$$K(t) := t(\tilde{M} - M^*) + \tilde{G}, \quad \forall t \in \mathbb{R}.$$

Since $K(t)$ is a linear combination of $\tilde{M} - M^*$ and $G$, the column space of $K(t)$ is a subspace of $\mathcal{U}$, and thus $K(t)$ has rank at most $2r$ and the $\delta$-RIP$_{2r,2r}$ property implies

$$(1 - \delta)\|K(t)\|_F^2 \le K(t) : \mathcal{H} : K(t) \le (1 + \delta)\|K(t)\|_F^2. \tag{19}$$

Using the facts that

$$\|K(t)\|_F^2 = \|\tilde{M} - M^*\|_F^2 \cdot t^2 + \|\tilde{G}\|_F^2,$$

$$K(t) : \mathcal{H} : K(t) = \alpha\|\tilde{M} - M^*\|_F^2 \cdot t^2 + 2\|\tilde{G}\|_F^2 \cdot t + \beta\|\tilde{G}\|_F^2,$$

we can write the two inequalities in (19) as quadratic inequalities

$$[\alpha - (1 - \delta)]\|\tilde{M} - M^*\|_F^2 \cdot t^2 + 2\|\tilde{G}\|_F^2 \cdot t + [\beta - (1 - \delta)]\|\tilde{G}\|_F^2 \ge 0,$$

$$[(1 + \delta) - \alpha]\|\tilde{M} - M^*\|_F^2 \cdot t^2 - 2\|\tilde{G}\|_F^2 \cdot t + [(1 + \delta) - \beta]\|\tilde{G}\|_F^2 \ge 0. \tag{20}$$

If $\alpha = 1 - \delta$, then we must have $\|\tilde{G}\|_F = 0$ and thus $G = \alpha(\tilde{M} - M^*)$. Equivalently, we have $M^* = \tilde{M} - \alpha^{-1}G$. Since the column and row spaces of $G \ne 0$ are orthogonal to $\tilde{M}$, the rank of $M^*$ is at least $\text{rank}(\tilde{M}) + 1 = r + 1$, which is a contradiction. Since $\alpha \le (1 + \delta)/2$, we have $\alpha < 1 + \delta$. Thus, we have proved that

$$1 - \delta < \alpha < 1 + \delta.$$

Checking the condition for quadratic functions to be non-negative, we obtain

$$\|\tilde{G}\|_F^2 \le [\alpha - (1 - \delta)][\beta - (1 - \delta)] \cdot \|\tilde{M} - M^*\|_F^2,$$

$$\|\tilde{G}\|_F^2 \le [(1 + \delta) - \alpha][(1 + \delta) - \beta] \cdot \|\tilde{M} - M^*\|_F^2.$$

Since

$$\alpha - (1 - \delta) > 0, \quad (1 + \delta) - \alpha > 0,$$

the above two inequalities are equivalent to

$$\frac{\|\tilde{G}\|_F^2}{\alpha - (1 - \delta)} \le [\beta - (1 - \delta)] \cdot \|\tilde{M} - M^*\|_F^2,$$

$$\frac{\|\tilde{G}\|_F^2}{(1 + \delta) - \alpha} \le [(1 + \delta) - \beta] \cdot \|\tilde{M} - M^*\|_F^2.$$

Summing up the two inequalities and dividing both sides by $2\delta$ gives rise to

$$\frac{\|\tilde{G}\|_F^2}{\delta^2 - (1 - \alpha)^2} \le \|\tilde{M} - M^*\|_F^2. \tag{21}$$

We note that the above condition is also sufficient for the inequalities in (20) to hold by choosing $\beta = 2 - \alpha$. Using the relation $\|G\|_F^2 = \|\tilde{G}\|_F^2 + \alpha^2\|\tilde{M} - M^*\|_F^2$, one can write

$$\text{tr}(\Lambda^2) = \|G\|_F^2 \le (2\alpha - 1 + \delta^2)\|\tilde{M} - M^*\|_F^2 = \alpha^{-1}(2\alpha - 1 + \delta^2)\langle\Lambda, R_{r+1:2r,r+1:2r}\rangle. \tag{22}$$

Now, using the fact that $\text{rank}(M^*) \le r$, we can write the matrix $R$ as

$$R = \begin{bmatrix} A \\ C \end{bmatrix} \begin{bmatrix} B \\ D \end{bmatrix}^T = \begin{bmatrix} AB^T & AD^T \\ CB^T & CD^T \end{bmatrix},$$

where $A, B, C, D \in \mathbb{R}^{r \times r}$. Then, conditions (18) and (22) become

$$\langle\Lambda, CD^T\rangle = \alpha\left[\text{tr}(\Sigma^2) - 2\langle\Sigma, AB^T\rangle + \|AB^T\|_F^2 + \|AD^T\|_F^2 + \|CB^T\|_F^2 + \|CD^T\|_F^2\right] \tag{23}$$

and

$$\text{tr}(\Lambda^2) \le \alpha^{-1}(2\alpha - 1 + \delta^2) \cdot \langle\Lambda, CD^T\rangle. \tag{24}$$

If $\langle\Lambda, CD^T\rangle = 0$, we have

$$\text{tr}(\Sigma^2) - 2\langle\Sigma, AB^T\rangle + \|AB^T\|_F^2 + \|AD^T\|_F^2 + \|CB^T\|_F^2 + \|CD^T\|_F^2 = 0,$$

which implies that

$$AB^T = \Sigma, \quad AD^T = CB^T = CD^T = 0.$$

This contradicts the assumption that $\tilde{M} \ne M^*$. Combining this with conditions (17), (23) and (24), we arrive at the necessity part. For problem (5), Lemma 3 in Ha et al. (2020) ensures that $\tilde{M}$ is still a fixed point of the SVP algorithm. Recalling the necessary conditions in Theorem 11, we know that the same necessary conditions also hold in this case.

**Sufficiency.** Now, we study the sufficiency part. We first consider problem (3). We choose two orthogonal matrices $U \in \mathbb{R}^{n \times 2r}, V \in \mathbb{R}^{m \times 2r}$ and define

$$\tilde{M} = U \begin{bmatrix} \Sigma & 0_{r \times r} \\ 0_{r \times r} & 0_{r \times r} \end{bmatrix} V^T, \quad M^* := U \left( \begin{bmatrix} A \\ C \end{bmatrix} \begin{bmatrix} B \\ D \end{bmatrix}^T \right) V^T, \quad G := U \begin{bmatrix} 0_{r \times r} & 0_{r \times r} \\ 0_{r \times r} & -\Lambda \end{bmatrix} V^T.$$

Since $\langle \Lambda, CD^T \rangle \neq 0$, we have $\tilde{M} \neq M^*$. Then, we know that $\mathrm{rank}(\tilde{M}) \leq r$ and $\mathrm{rank}(M^*) \leq r$. We define

$$\tilde{G} := G - \alpha(\tilde{M} - M^*),$$

which satisfies $\langle \tilde{G}, \tilde{M} - M^* \rangle = 0$ by the condition in the second line of (6). If $\tilde{G} = 0$, then

$$\begin{bmatrix} 0_{r \times r} & 0_{r \times r} \\ 0_{r \times r} & -\Lambda \end{bmatrix} = \alpha \cdot \begin{bmatrix} \Sigma & 0_{r \times r} \\ 0_{r \times r} & 0_{r \times r} \end{bmatrix} - \alpha \cdot \begin{bmatrix} A \\ C \end{bmatrix} \begin{bmatrix} B \\ D \end{bmatrix}^T = \alpha \cdot \begin{bmatrix} \Sigma & 0_{r \times r} \\ 0_{r \times r} & 0_{r \times r} \end{bmatrix} - \alpha \cdot \begin{bmatrix} AB^T & 0 \\ 0 & CD^T \end{bmatrix},$$

where the second step is because of $CB^T = 0$ and $AD^T = 0$. The above relation is equivalent to

$$\Sigma = AB^T, \quad \Lambda = \alpha \cdot CD^T.$$

Since $\Sigma \succ 0$, the matrix $AB^T$ has rank $r$. Noticing that the decomposition of matrix $M^*$ ensures that the rank of $M^*$ is at most $r$, we have $CD^T = 0$, which is a contradiction to the condition that $\langle CD^T, \Lambda \rangle \neq 0$. Therefore, we have $\tilde{G} \neq 0$. We consider the rank-2 symmetric tensor

$$\mathcal{G}_1 := \frac{\alpha}{\|\tilde{M} - M^*\|_F^2} \cdot (\tilde{M} - M^*) \otimes (\tilde{M} - M^*) + \frac{2 - \alpha}{\|\tilde{G}\|_F^2} \cdot \tilde{G} \otimes \tilde{G}$$
$$+ \frac{1}{\|\tilde{M} - M^*\|_F^2} \left[ (\tilde{M} - M^*) \otimes \tilde{G} + \tilde{G} \otimes (\tilde{M} - M^*) \right].$$

For every matrix $K \in \mathbb{R}^{n \times m}$, we have the decomposition

$$K = t(\tilde{M} - M^*) + s\tilde{G} + \tilde{K}, \quad \langle \tilde{M} - M^*, \tilde{K} \rangle = \langle \tilde{G}, \tilde{K} \rangle = 0,$$

where $t, s \in \mathbb{R}$ are two suitable constants. Then, using the definition of $\mathcal{G}_1$, we have

$$K : \mathcal{G}_1 : K = \alpha \|\tilde{M} - M^*\|_F^2 \cdot t^2 + 2\|\tilde{G}\|_F^2 \cdot ts + (2 - \alpha)\|\tilde{G}\|_F^2 \cdot s^2.$$

By the conditions in the third line of (6), one can write

$$\|\tilde{G}\|_F^2 \leq [\alpha - (1 - \delta)][(1 + \delta) - \alpha] \cdot \|\tilde{M} - M^*\|_F^2,$$

which leads to

$$[\alpha - (1 - \delta)]\|\tilde{M} - M^*\|_F^2 \cdot t^2 + 2\|\tilde{G}\|_F^2 \cdot ts + [(1 + \delta) - \alpha]\|\tilde{G}\|_F^2 \cdot s^2 \geq 0,$$
$$[(1 + \delta) - \alpha]\|\tilde{M} - M^*\|_F^2 \cdot t^2 - 2\|\tilde{G}\|_F^2 \cdot ts + [\alpha - (1 - \delta)]\|\tilde{G}\|_F^2 \cdot s^2 \geq 0.$$

The above two inequalities are equivalent to

$$(1 - \delta)[\|\tilde{M} - M^*\|_F^2 \cdot s^2 + \|\tilde{G}\|_F^2 \cdot t^2] \leq K : \mathcal{G}_1 : K \leq (1 + \delta)[\|\tilde{M} - M^*\|_F^2 \cdot s^2 + \|\tilde{G}\|_F^2 \cdot t^2]. \tag{25}$$

By restricting to the subspace

$$\mathcal{S} := \mathrm{span}\{\tilde{M} - M^*, \tilde{G}\} = \{s(\tilde{M} - M^*) + t\tilde{G} \mid s, t \in \mathbb{R}\},$$

the tensor $\mathcal{G}_1$ can be viewed as a $2 \times 2$ matrix. Then, inequality (25) implies that the matrix has two eigenvalues $\lambda_1$ and $\lambda_2$ such that

$$1 - \delta \leq \lambda_1, \lambda_2 \leq 1 + \delta.$$

Therefore, we can rewrite the tensor $\mathcal{G}_1$ restricted to $\mathcal{S}$ as

$$[\mathcal{G}_1]_{\mathcal{S}} = \lambda_1 \cdot G_1 \otimes G_1 + \lambda_2 \cdot G_2 \otimes G_2,$$

where $G_1, G_2$ are linear combinations of $\tilde{M} - M^*, \tilde{G}$ and have the unit norm. Since the orthogonal complementary $\mathcal{S}^\perp$ is in the null space of $\mathcal{G}_1$, we have

$$\mathcal{G}_1 = [\mathcal{G}_1]_{\mathcal{S}} = \lambda_1 \cdot G_1 \otimes G_1 + \lambda_2 \cdot G_2 \otimes G_2.$$

Now, we choose matrices $G_3, \ldots, G_N$ such that $G_1, \ldots, G_N$ form an orthonormal basis of the linear vector space $\mathbb{R}^{n \times m}$, where $N := nm$. We define another symmetric tensor by

$$\mathcal{H} := \mathcal{G}_1 + \sum_{i=3}^{N} (1 + \delta) \cdot G_i \otimes G_i.$$

Then, inequality (25) implies that the quadratic form $K : \mathcal{H} : K$ satisfies the $\delta$-RIP$_{2r, 2r}$ property.

Therefore, we can choose the Hessian to be the constant tensor $\mathcal{H}$ and define the function $f_a(\cdot)$ as

$$f_a(K) := \frac{1}{2}(K - M^*) : \mathcal{H} : (K - M^*), \quad \forall K \in \mathbb{R}^{n \times m}.$$

Combining with the definition of $\mathcal{H}$, we know

$$\nabla f_a(\tilde{M}) = \mathcal{H} : (\tilde{M} - M^*) = G, \quad \nabla^2 f_a(\tilde{M}) = \mathcal{H}.$$

We choose matrices $\bar{U} \in \mathbb{R}^{n \times r}, \bar{V} \in \mathbb{R}^{m \times r}$ such that $\tilde{M} = \bar{U}\bar{V}^T$ and $\bar{U}^T\bar{U} = \bar{V}^T\bar{V}$. By the definitions of $\tilde{M}$ and $G$, we know that $\tilde{M}$ and $G$ have orthogonal column and row spaces, i.e.,

$$\bar{U}^T G = 0, \quad G\bar{V} = 0.$$

This means that the first-order optimality conditions are satisfied at the point $(\bar{U}, \bar{V})$. For the second-order necessary optimality conditions, we consider the direction

$$\Delta := \begin{bmatrix} \Delta_U \\ \Delta_V \end{bmatrix} \in \mathbb{R}^{(n+m) \times r}.$$

We consider the decomposition

$$\Delta_U = \mathcal{P}_{\bar{U}}\Delta_U + \mathcal{P}_{\bar{U}}^{\perp}\Delta_U := \Delta_U^1 + \Delta_U^2, \quad \Delta_V = \mathcal{P}_{\bar{V}}\Delta_V + \mathcal{P}_{\bar{V}}^{\perp}\Delta_V := \Delta_V^1 + \Delta_V^2,$$

where $\mathcal{P}_{\bar{U}}, \mathcal{P}_{\bar{V}}$ are the orthogonal projection onto the column space of $\bar{U}, \bar{V}$, respectively. Then, using the conditions in the first line of (6), we have

$$\langle \nabla f_a(\tilde{M}), \Delta_U \Delta_V^T \rangle = \langle G, \Delta_U \Delta_V^T \rangle = \langle G, \Delta_U^2 (\Delta_V^2)^T \rangle \geq -\|G^T \Delta_U^2\|_F \|\Delta_V^2\|_F$$

$$\geq -(1 + \delta)\sigma_r(\tilde{M})\|\Delta_U^2\|_F\|\Delta_V^2\|_F \geq -(1 + \delta)\sigma_r(\tilde{M}) \cdot \frac{\|\Delta_U^2\|_F^2 + \|\Delta_V^2\|_F^2}{2}. \tag{26}$$

We define

$$\Delta_1 := \bar{U}(\Delta_V^1)^T + \Delta_U^1 \bar{V}^T, \quad \Delta_2 := \bar{U}(\Delta_V^2)^T + \Delta_U^2 \bar{V}^T.$$

Then, we know that $\langle \Delta_1, \Delta_2 \rangle = 0$. Using the assumption that $CB^T = AD^T = 0$, we know that $M^*$ has the form

$$M^* = U \begin{bmatrix} AB^T & 0 \\ 0 & CD^T \end{bmatrix} V^T = \mathcal{P}_{\bar{U}} M^* \mathcal{P}_{\bar{V}} + \mathcal{P}_{\bar{U}}^{\perp} M^* \mathcal{P}_{\bar{V}}^{\perp}. \tag{27}$$

Then, the special form (27) implies that

$$\langle M^*, \Delta_2 \rangle = \langle M^*, \bar{U}(\Delta_V^2)^T + \Delta_U^2 \bar{V}^T \rangle = \langle M^*, \bar{U}\Delta_V^T \mathcal{P}_{\bar{V}}^{\perp} + \mathcal{P}_{\bar{U}}^{\perp}\Delta_U \bar{V}^T \rangle = 0.$$

Using the definitions of $\tilde{M}$ and $G$, it can be concluded that

$$\langle \tilde{M}, \Delta_2 \rangle = 0, \quad \langle G, \Delta_2 \rangle = \langle G, \bar{U}(\Delta_V^2)^T + \Delta_U^2 \bar{V}^T \rangle = 0.$$

Since $G_1, G_2$ are linear combinations of $\tilde{M} - M^*$ and $G$, the last three relations lead to

$$\langle G_1, \Delta_2 \rangle = \langle G_2, \Delta_2 \rangle = 0.$$

Therefore, there exist constants $a_3, \ldots, a_N$ such that

$$\Delta_2 = \sum_{i=3}^{N} a_i G_i.$$

Suppose that the constants $b_1, \ldots, b_N$ satisfy

$$\Delta_1 = \sum_{i=1}^{N} b_i G_i.$$

Then, the fact $\langle \Delta_1, \Delta_2 \rangle = 0$ and the orthogonality of $G_1, \ldots, G_N$ imply that

$$\sum_{i=3}^{N} a_i b_i = 0.$$

We can calculate that

$$[\nabla^2 f_a(\tilde{M})](\bar{U}\Delta_V^T + \Delta_U \bar{V}^T, \bar{U}\Delta_V^T + \Delta_U \bar{V}^T) = (\Delta_1 + \Delta_2) : \mathcal{H} : (\Delta_1 + \Delta_2)$$

$$= \lambda_1 \cdot b_1^2 + \lambda_2 \cdot b_2^2 + (1+\delta)\sum_{i=3}^{N}(a_i + b_i)^2 \geq (1+\delta)\sum_{i=3}^{N}(a_i + b_i)^2$$

$$= (1+\delta)\sum_{i=3}^{N}\left(a_i^2 + b_i^2\right) \geq (1+\delta)\sum_{i=3}^{N} a_i^2 = (1+\delta)\|\bar{U}(\Delta_V^2)^T + \Delta_U^2 \bar{V}^T\|_F^2,$$

where the third last step is due to $\sum_{i=3}^{N} a_i b_i = 0$. Noticing that $\langle \bar{U}(\Delta_V^2)^T, \Delta_U^2 \bar{V}^T \rangle = 0$, the above inequality gives that

$$[\nabla^2 f_a(\tilde{M})](\bar{U}\Delta_V^T + \Delta_U \bar{V}^T, \bar{U}\Delta_V^T + \Delta_U \bar{V}^T) \geq (1+\delta)\|\bar{U}(\Delta_V^2)^T\|_F^2 + (1+\delta)\|\Delta_U^2 \bar{V}^T\|_F^2$$

$$\geq (1+\delta)\sigma_r(\bar{U})^2\|\Delta_V^2\|_F^2 + (1+\delta)\sigma_r(\bar{V})^2\|\Delta_U^2\|_F^2 = (1+\delta)\sigma_r(\tilde{M})(\|\Delta_V^2\|_F^2 + \|\Delta_U^2\|_F^2),$$

where the last equality is because of $\sigma_r(\bar{U})^2 = \sigma_r(\bar{V})^2 = \sigma_r(\tilde{M})$ when $\bar{U}^T\bar{U} = \bar{V}^T\bar{V}$. Combining with inequality (26), one can write

$$[\nabla^2 h_a(U,V)](\Delta, \Delta) = 2\langle \nabla f_a(\tilde{M}), \Delta_U \Delta_V^T \rangle + [\nabla^2 f_a(\tilde{M})](\bar{U}\Delta_V^T + \Delta_U \bar{V}^T, \bar{U}\Delta_V^T + \Delta_U \bar{V}^T)$$

$$\geq -(1+\delta)\sigma_r(\tilde{M})(\|\Delta_V^2\|_F^2 + \|\Delta_U^2\|_F^2) + (1+\delta)\sigma_r(\tilde{M})(\|\Delta_V^2\|_F^2 + \|\Delta_U^2\|_F^2) = 0.$$

This shows that $(\bar{U}, \bar{V})$ satisfies the second-order necessary optimality conditions, and therefore it is a spurious second-order critical point.

Now, we consider problem (5). Since the point $(\bar{U}, \bar{V})$ satisfies $\bar{U}^T\bar{U} = \bar{V}^T\bar{V}$, it is also a local minimum of the regularization term. Hence, the point $(\bar{U}, \bar{V})$ is also a spurious second-order critical point of the regularized problem (5). $\qquad \square$

## D.2 Proof of Corollary 1

*Proof of Corollary 1.* We assume that problem (3) has a spurious second-order critical point. By the necessity part of Theorem (6), there exist $\alpha \in (1-\delta, 1+\delta)$ and real numbers $\sigma, \lambda, a, b, c, d$ such that

$$(1+\delta)\sigma \geq \lambda > 0, \quad \alpha^{-1}(2\alpha - 1 + \delta^2)cd \cdot \lambda \geq \lambda^2 > 0,$$
$$cd \cdot \lambda = \alpha[\sigma^2 - 2ab \cdot \sigma + (ab)^2 + (ad)^2 + (cb)^2 + (cd)^2]. \tag{28}$$

We first relax the second line to

$$cd \cdot \lambda \geq \alpha[\sigma^2 - 2|ab| \cdot \sigma + (ab)^2 + 2|ab| \cdot |cd| + (cd)^2]. \tag{29}$$

Then, we denote $x := |ab|$ and consider the quadratic programming problem

$$\min_{x \geq 0} x^2 + 2(|cd| - \sigma) \cdot x,$$

whose optimal value is

$$-(\sigma - |cd|)_+^2,$$

where $(t)_+ := \max\{t, 0\}$. Substituting into inequality (29), we obtain

$$cd \cdot \lambda \geq \alpha[\sigma^2 - (\sigma - |cd|)_+^2 + (cd)^2]. \tag{30}$$

Then, we consider two different cases.

**Case I.**  We first consider the case when $\sigma \geq |cd|$. In this case, the inequality (30) becomes
$$cd \cdot \lambda \geq 2\alpha \cdot \sigma|cd| = 2\alpha \cdot \sigma cd,$$
where the last equality is due to $cd > 0$. Therefore,
$$\lambda \geq 2\alpha \cdot \sigma.$$
The second inequality in (28) implies $\lambda \leq \alpha^{-1}(2\alpha - 1 + \delta^2) \cdot cd$. Combining with the above inequality and the assumption of this case, it follows that
$$\alpha^{-1}(2\alpha - 1 + \delta^2) \cdot \sigma \geq \alpha^{-1}(2\alpha - 1 + \delta^2) \cdot cd \geq 2\alpha \cdot \sigma,$$
which is further equivalent to
$$\alpha^{-1}(2\alpha - 1 + \delta^2) \geq 2\alpha \iff \delta^2 \geq 2\alpha^2 - 2\alpha + 1.$$
Since $2\alpha^2 - 2\alpha + 1 \geq 1/2$, we arrive at $\delta^2 \geq 1/2$, which is a contradiction to $\delta < 1/2$.

**Case II.**  We then consider the case when $\sigma \leq |cd|$. In this case, the inequality (30) becomes
$$cd \cdot \lambda \geq \alpha[\sigma^2 + (cd)^2].$$
Combining with the second inequality in (28), we obtain $\lambda \leq \alpha^{-1}(2\alpha - 1 + \delta^2) \cdot (cd)$. Therefore,
$$\alpha^{-1}(2\alpha - 1 + \delta^2) \cdot (cd)^2 \geq cd \cdot \lambda \geq \alpha[\sigma^2 + (cd)^2].$$
Moreover, the first inequality in (28) gives
$$(1 + \delta)\sigma \cdot cd \geq cd \cdot \lambda \geq \alpha[\sigma^2 + (cd)^2].$$
By denoting $y := cd$, the above two inequalities become
$$\alpha^{-1}(2\alpha - 1 + \delta^2) \cdot y^2 \geq \alpha[\sigma^2 + y^2],$$
$$(1 + \delta)\sigma \cdot y \geq \alpha[\sigma^2 + y^2]. \tag{31}$$
By denoting $z := y/\sigma$, the first inequality in (31) implies
$$z^2 \geq \frac{\alpha^2}{\delta^2 - (1 - \alpha)^2}. \tag{32}$$
Since $\delta < 1/2$, one can write
$$(1 - \alpha)^2 + \alpha^2 \geq \frac{1}{2} > \frac{1}{4} > \delta^2,$$
which is equivalent to $\alpha^2 \geq \delta^2 - (1 - \alpha)^2$. Therefore, inequality (32) implies that $z^2 \geq 1$ and
$$z^2 + \frac{1}{z^2} \geq \frac{\alpha^2}{\delta^2 - (1 - \alpha)^2} + \frac{\delta^2 - (1 - \alpha)^2}{\alpha^2}. \tag{33}$$
On the other hand, the second inequality in (31) implies
$$z + \frac{1}{z} \leq \frac{1 + \delta}{\alpha} \quad \text{and thus} \quad z^2 + \frac{1}{z^2} + 2 \leq \frac{(1 + \delta)^2}{\alpha^2}.$$
Combining with inequality (33), it follows that
$$\frac{\alpha^2}{\delta^2 - (1 - \alpha)^2} + \frac{\delta^2 - (1 - \alpha)^2}{\alpha^2} + 2 \leq \frac{(1 + \delta)^2}{\alpha^2}. \tag{34}$$
By some calculation, the above inequality is equivalent to
$$(\delta^2 + 2\delta + 5) \cdot \alpha^2 + (2\delta^2 - 4\delta - 6) \cdot \alpha + 2(1 + \delta)(1 - \delta^2) \leq 0.$$
Checking the discriminant of the above quadratic function, we obtain
$$(2\delta^2 - 4\delta - 6)^2 - 8(\delta^2 + 2\delta + 5)(1 + \delta)(1 - \delta^2) \geq 0,$$
which is equivalent to
$$4(2\delta - 1)(\delta + 1)^4 \geq 0.$$
However, the above claim contradicts the assumption that $\delta < 1/2$.

In summary, the contradictions in the two cases imply that the condition (28) cannot hold, and therefore there does not exist spurious second-order critical points.  $\square$

### D.3 Counterexample for the Rank-one Case

**Example 3.** Let $e_i \in \mathbb{R}^n$ be the $i$-th standard basis of $\mathbb{R}^n$. We define the tensor

$$\mathcal{H} := \sum_{i,j=1}^n (e_i e_j^T) \otimes (e_i e_j^T) + \frac{1}{2}(e_1 e_1^T) \otimes (e_2 e_2^T) + \frac{1}{2}(e_2 e_2^T) \otimes (e_1 e_1^T)$$

$$+ \frac{1}{4}\left[(e_1 e_2^T) \otimes (e_1 e_2^T) + (e_2 e_1^T) \otimes (e_2 e_1^T)\right] + \frac{1}{4}(e_1 e_2^T) \otimes (e_2 e_1^T) + \frac{1}{4}(e_2 e_1^T) \otimes (e_1 e_2^T)$$

and the objective function

$$f_a(M) := (M - e_1 e_1^T) : \mathcal{H} : (M - e_1 e_1^T) \quad \forall M \in \mathbb{R}^{n \times n}.$$

The global minimizer of $f_a(\cdot)$ is the rank-1 matrix $M^* := e_1 e_1^T$. It has been proved in Zhang et al. (2019) that the function $f_a(\cdot)$ satisfies the $\delta$-RIP$_{2,2}$ property with $\delta = 1/2$. Moreover, we define

$$U := \frac{1}{\sqrt{2}} e_2, \quad V := U, \quad \tilde{M} := UU^T \neq M^*.$$

It has been proved in Zhang et al. (2019) that the first-order optimality condition is satisfied. To verify the second-order necessary condition, we can calculate that

$$[\nabla^2 h_a(U, U)](\Delta, \Delta) = 2\langle \nabla f_a(\tilde{M}), \Delta_U \Delta_V^T \rangle + (U\Delta_V^T + \Delta_U U^T) : \mathcal{H} : (U\Delta_V^T + \Delta_U U^T)$$

$$= -\frac{3}{2}(\Delta_U)_1(\Delta_V)_1 + \frac{5}{8}\left[(\Delta_U)_1^2 + (\Delta_V)_1^2\right] + \frac{1}{4}(\Delta_U)_1(\Delta_V)_1$$

$$+ \frac{1}{2}\left[(\Delta_U)_2 + (\Delta_V)_2\right]^2 + \frac{1}{2}\sum_{i=3}^n \left[(\Delta_U)_i^2 + (\Delta_V)_i^2\right]$$

$$= \frac{5}{8}\left[(\Delta_U)_1 - (\Delta_V)_1\right]^2 + \frac{1}{2}\left[(\Delta_U)_2 + (\Delta_V)_2\right]^2 + \frac{1}{2}\sum_{i=3}^n \left[(\Delta_U)_i^2 + (\Delta_V)_i^2\right],$$

which is non-negative for every $\Delta \in \mathbb{R}^n$. Hence, we conclude that the point $\tilde{M}$ is a spurious second-order critical point of problem (3). Moreover, since we choose $V = U$, the point $\tilde{M}$ is a global minimizer of the regularizer $\|U^T U - V^T V\|_F^2$ and thus $\tilde{M}$ is also a spurious second-order critical point of problem (5).

### D.4 Proof of Corollary 2

*Proof of Corollary 2.* We first consider the case when $\delta \leq 1/3$. We assume that there exists a spurious second-order critical point $\tilde{M}$. Then, by Theorem 4, we know that there exists a constant $\alpha \in (1 - \delta, (1 + \delta)/2]$. This means that

$$1 - \delta < \frac{1 + \delta}{2},$$

which contradicts the assumption that $\delta \leq 1/3$.

Then, we consider the case when $\delta < 1/2$. With no loss of generality, assume that $\tilde{M} \neq M^*$ and $M^* \neq 0$; otherwise, the inequality in this theorem is trivially true. Define

$$m_{11} := \|\Sigma\|_F^2, \quad m_{12} := \langle \Sigma, AB^T \rangle, \quad m_{22} := \|AB^T\|_F^2 + \|AD^T\|_F^2 + \|CB^T\|_F^2 + \|CD^T\|_F^2.$$

By our construction in Theorem 4, we know that

$$m_{11} = \|\tilde{M}\|_F^2, \quad m_{12} = \langle \tilde{M}, M^* \rangle, \quad m_{22} = \|M^*\|_F^2.$$

Therefore, we only need to prove $m_{12} \geq C(\delta) \cdot \sqrt{m_{11} m_{22}}$ for some constant $C(\delta) > 0$. By the analysis in Ha et al. (2020), we know that the second-order critical point $\tilde{M}$ must have rank $r$ and thus $m_{11} \neq 0$. The remainder of the proof is split into two steps.

**Step I.** First, we prove that

$$\frac{(m_{11} + m_{22} - 2m_{12})^2}{m_{11}m_{22} - m_{12}^2} \leq \frac{(1+\delta)^2}{\alpha^2}, \quad \frac{(m_{11} - m_{12})^2}{m_{11}m_{22} - m_{12}^2} \leq \frac{\delta^2 - (1-\alpha)^2}{\alpha^2}. \tag{35}$$

We first rule out the case when $m_{11}m_{22} - m_{12}^2 = 0$. In this case, the equality condition of the Cauchy inequality shows that there exists a constant $t$ such that

$$\tilde{M} = tM^*.$$

Since $\tilde{M} \neq 0$, the constant $t$ is not 0. Using the mean value theorem, for any $Z \in \mathbb{R}^{n \times m}$, there exists a constant $c \in [0, 1]$ such that

$$\langle \nabla f_a(\tilde{M}), Z \rangle = \nabla^2 f[M^* + c(\tilde{M} - M^*)](\tilde{M} - M^*, Z)$$
$$= \nabla^2 f[M^* + c(\tilde{M} - M^*)][(t-1)M^*, Z].$$

The $\delta$-RIP$_{2r,2r}$ property gives

$$\langle \nabla f_a(\tilde{M}), \tilde{M} \rangle = \nabla^2 f[M^* + c(\tilde{M} - M^*)][(t-1)M^*, tM^*] \geq t(t-1)(1-\delta)\|M^*\|_F^2.$$

If $t = 1$, we conclude that $\tilde{M} = M^*$, which contradicts the assumption that $\tilde{M} \neq M^*$. Therefore, it holds that

$$\langle \tilde{M}, \nabla f_a(\tilde{M}) \rangle \neq 0.$$

This contradicts the first-order optimality condition, which states that $\langle \tilde{M}, \nabla f_a(\tilde{M}) \rangle = 0$. Hence, we have proved that inequality (35) is well defined. We consider the decomposition

$$\begin{bmatrix} 0 & 0 \\ 0 & \Lambda \end{bmatrix} = c_1 \begin{bmatrix} \Sigma & 0 \\ 0 & 0 \end{bmatrix} + c_2 \begin{bmatrix} A \\ C \end{bmatrix} \begin{bmatrix} B \\ D \end{bmatrix}^T + K, \quad \left\langle K, \begin{bmatrix} \Sigma & 0 \\ 0 & 0 \end{bmatrix} \right\rangle = \left\langle K, \begin{bmatrix} A \\ C \end{bmatrix} \begin{bmatrix} B \\ D \end{bmatrix}^T \right\rangle = 0.$$

Using the conditions in Theorem 4, it follows that

$$\left\langle \begin{bmatrix} 0 & 0 \\ 0 & \Lambda \end{bmatrix}, \begin{bmatrix} \Sigma & 0 \\ 0 & 0 \end{bmatrix} \right\rangle = 0, \quad \left\langle \begin{bmatrix} 0 & 0 \\ 0 & \Lambda \end{bmatrix}, \begin{bmatrix} A \\ C \end{bmatrix} \begin{bmatrix} B \\ D \end{bmatrix}^T \right\rangle = \alpha(m_{11} - 2m_{12} + m_{22}).$$

The pair of coefficients $(c_1, c_2)$ can be uniquely solved as

$$c_1 = -\alpha \cdot \frac{m_{11} + m_{22} - 2m_{12}}{m_{11}m_{22} - m_{12}^2} \cdot m_{12}, \quad c_2 = \alpha \cdot \frac{m_{11} + m_{22} - 2m_{12}}{m_{11}m_{22} - m_{12}^2} \cdot m_{11}.$$

Using the orthogonality of the decomposition, we have

$$\|\Lambda\|_F^2 \geq \left\| c_1 \begin{bmatrix} \Sigma & 0 \\ 0 & 0 \end{bmatrix} + c_2 \begin{bmatrix} A \\ C \end{bmatrix} \begin{bmatrix} B \\ D \end{bmatrix}^T \right\|_F^2 = c_1^2 m_{11} + 2c_1 c_2 m_{12} + c_2^2 m_{22}$$
$$= \alpha^2 \cdot \frac{m_{11}(m_{11} + m_{22} - 2m_{12})^2}{m_{11}m_{22} - m_{12}^2}. \tag{36}$$

Using the last two lines of condition (6), one can write

$$\alpha^2 \cdot \frac{m_{11}(m_{11} + m_{22} - 2m_{12})^2}{m_{11}m_{22} - m_{12}^2} \leq \|\Lambda\|_F^2$$
$$\leq (2\alpha - 1 + \delta^2) \left[ \text{tr}(\Sigma^2) - 2\langle \Sigma, AB^T \rangle + \|AB^T\|_F^2 + \|AD^T\|_F^2 + \|CB^T\|_F^2 + \|CD^T\|_F^2 \right]$$
$$= (2\alpha - 1 + \delta^2)(m_{11} - 2m_{12} + m_{22}).$$

Simplifying the above inequality, we arrive at the second inequality in (35). Now, the first inequality in condition (6) implies that

$$\|\Lambda\|_F^2 \leq (1+\delta)^2 \|\Sigma\|_F^2 = (1+\delta)^2 m_{11}.$$

Substituting inequality (36) into the left-hand side, it follows that

$$\alpha^2 \cdot \frac{m_{11}(m_{11} + m_{22} - 2m_{12})^2}{m_{11}m_{22} - m_{12}^2} \leq (1+\delta)^2 m_{11},$$

which is equivalent to the first inequality in (35).

**Step II.** Next, we prove the existence of $C(\delta)$. We denote

$$\kappa := \frac{m_{12}}{\sqrt{m_{11}m_{22}}} \in (-1, 1).$$

and

$$C_1 := \frac{\delta^2 - (1-\alpha)^2}{\alpha^2}, \quad C_2 := \frac{(1+\delta)^2}{\alpha^2}, \quad t := \sqrt{\frac{m_{11}}{m_{22}}}.$$

Since $\tilde{M} \neq 0$, we have $t > 0$. The inequalities in (35) can be written as

$$(t - \kappa)^2 \leq (1 - \kappa^2)C_1, \quad (t + 1/t - 2\kappa)^2 \leq (1 - \kappa^2)C_2. \tag{37}$$

Using the assumption that $\delta < 1/2$, we can write

$$\delta^2 < \frac{1}{4} < (1 - \alpha)^2 + \frac{1}{2}\alpha^2,$$

which leads to

$$C_1 = \frac{\delta^2 - (1-\alpha)^2}{\alpha^2} < \frac{1}{2}.$$

If $\kappa + \sqrt{(1 - \kappa^2)C_1} \geq 1$, then

$$|\kappa| \geq \frac{1 - C_1}{1 + C_1} \geq \frac{1}{3} > 0. \tag{38}$$

If $\kappa < 0$, then it holds that

$$\kappa + \sqrt{(1 - \kappa^2)C_1} \leq -\frac{1}{3} + \sqrt{\frac{1}{2}} < 1,$$

which contradicts the assumption. Therefore, we have $\kappa \geq 0$ and inequality (38) gives $\kappa \geq 1/3$.

Now, we assume that $\kappa + \sqrt{(1 - \kappa^2)C_1} \leq 1$. Then, the first inequality in (37) gives

$$0 < t \leq \kappa + \sqrt{(1 - \kappa^2)C_1} \leq 1,$$

which further leads to

$$t + \frac{1}{t} - 2\kappa \geq -\kappa + \sqrt{(1 - \kappa^2)C_1} + \frac{1}{\kappa + \sqrt{(1 - \kappa^2)C_1}}.$$

Combining with the second inequality in (37), we obtain

$$-\kappa + \sqrt{(1 - \kappa^2)C_1} + \frac{1}{\kappa + \sqrt{(1 - \kappa^2)C_1}} \leq \sqrt{(1 - \kappa^2)C_2}.$$

The above inequality can be simplified to

$$\sqrt{1 - \kappa^2}(1 + C_1 - \sqrt{C_1 C_2}) \leq \kappa\sqrt{C_2}.$$

We notice that the inequality $1 + C_1 - \sqrt{C_1 C_2} \leq 0$ is equivalent to inequality (34), which cannot hold when $\delta < 1/2$. Therefore, we have $1 + C_1 - \sqrt{C_1 C_2} > 0$ and $\kappa > 0$. Then, the above inequality is equivalent to

$$(1 - \kappa^2)(1 + C_1 - \sqrt{C_1 C_2})^2 \leq \kappa^2 \cdot C_2.$$

Therefore, we have

$$\kappa^2 \geq \frac{(1 + C_1 - \sqrt{C_1 C_2})^2}{(1 + C_1 - \sqrt{C_1 C_2})^2 + C_2} = 1 - \frac{1}{1 + \eta^2},$$

where we define

$$\eta := \frac{1 + C_1 - \sqrt{C_1 C_2}}{\sqrt{C_2}}.$$

To prove the existence of $C(\delta)$ such that $\kappa \geq C(\delta) > 0$, we only need to show that $\eta$ is lower bounded by a positive constant. With $\delta$ fixed, $\eta$ can be viewed as a continuous function of $\alpha$. Since $\eta = (1 - \delta)/(1 + \delta) > 0$ when $\alpha = 1 - \delta$, the function/parameter $\eta$ is defined for all $\alpha$ in the compact set $[1 - \delta, (1 + \delta)/2]$. Combining with the fact that $1 + C_1 - \sqrt{C_1 C_2} > 0$, the function $\eta$ is positive on a compact set, and thus there exists a positive lower bound $\bar{C}(\delta) > 0$.

In summary, we can define the function

$$C(\delta) := \min\left\{\frac{1}{3}, \bar{C}(\delta)\right\} > 0$$

such that $\kappa \geq C(\delta)$ for every spurious second-order critical point $\tilde{M}$. $\qquad\square$

## D.5 Counterexample for the General Rank Case with Linear Measurements

**Example 4.** Using the previous rank-1 example, we design a counterexample with linear measurement for the rank-$r$ case. Let $n \geq 2r$ be an integer and $e_i \in \mathbb{R}^n$ be the $i$-th standard basis of $\mathbb{R}^n$. We define the tensor

$$
\mathcal{H} := \frac{3}{2} \sum_{i,j=1}^{n} (e_i e_j^T) \otimes (e_i e_j^T) + \sum_{i=1}^{r} \Big\{ -\frac{1}{2} \big[ (e_{2i-1} e_{2i-1}^T) \otimes (e_{2i-1} e_{2i-1}^T) + (e_{2i} e_{2i}^T) \otimes (e_{2i} e_{2i}^T) \big]
$$

$$
+ \frac{1}{2} \big[ (e_{2i-1} e_{2i-1}^T) \otimes (e_{2i} e_{2i}^T) + (e_{2i} e_{2i}^T) \otimes (e_{2i-1} e_{2i-1}^T) \big]
$$

$$
- \frac{1}{4} \big[ (e_{2i-1} e_{2i}^T) \otimes (e_{2i-1} e_{2i}^T) + (e_{2i} e_{2i-1}^T) \otimes (e_{2i} e_{2i-1}^T) \big]
$$

$$
+ \frac{1}{4} \big[ (e_{2i-1} e_{2i}^T) \otimes (e_{2i} e_{2i-1}^T) + (e_{2i} e_{2i-1}^T) \otimes (e_{2i-1} e_{2i}^T) \big] \Big\}
$$

and the rank-$r$ global minimum

$$
U^* := \begin{bmatrix} e_1 & e_3 & \cdots & e_{2r-1} \end{bmatrix}, \quad M^* := U^*(U^*)^T = \sum_{i=1}^{r} e_{2i-1} e_{2i-1}^T.
$$

The objective function is defined as

$$
f_a(M) := (M - M^*) : \mathcal{H} : (M - M^*) \quad \forall M \in \mathbb{R}^{n \times n}.
$$

We can similarly prove that the function $f_a(\cdot)$ satisfies the $\delta$-RIP$_{2r,2r}$ property with $\delta = 1/2$. Moreover, we define

$$
\tilde{U} := \frac{1}{\sqrt{2}} \begin{bmatrix} e_2 & e_4 & \cdots & e_{2r} \end{bmatrix}, \quad \tilde{M} := \tilde{U}\tilde{U}^T = \frac{1}{2} \sum_{i=1}^{r} e_{2i} e_{2i}^T \neq M^*.
$$

The gradient of $f_a(\cdot)$ at point $\tilde{M}$ is

$$
\nabla f_a(\tilde{M}) = -\frac{3}{4} \sum_{i=1}^{r} e_{2i-1} e_{2i-1}^T \in \mathbb{R}^{2r \times 2r}.
$$

Since the column and row spaces of the gradient are orthogonal to those of $\tilde{M}$, the first-order optimality condition is satisfied. To verify the second-order necessary condition, we can similarly calculate that

$$
[\nabla^2 h_a(\tilde{U}, \tilde{U})](\Delta, \Delta)
$$

$$
= 2 \langle \nabla f_a(\tilde{M}), \Delta_U \Delta_V^T \rangle + (\tilde{U}\Delta_V^T + \Delta_V \tilde{U}^T) : \mathcal{H} : (\tilde{U}\Delta_V^T + \Delta_U \tilde{U}^T)
$$

$$
= -\frac{3}{2} \sum_{i=1}^{r} \left[ \sum_{j=1}^{r} (\Delta_U)_{2i-1,j} \right] \left[ \sum_{j=1}^{r} (\Delta_V)_{2i-1,j} \right] + \sum_{i=1}^{r} \Big\{ \frac{5}{8} \big[ (\Delta_U)_{2i-1,i}^2 + (\Delta_V)_{2i-1,i}^2 \big]
$$

$$
+ \frac{1}{4} (\Delta_U)_{2i-1,i} (\Delta_V)_{2i-1,i} + \frac{1}{2} \big[ (\Delta_U)_{2i,i} + (\Delta_V)_{2i,i} \big]^2 \Big\}
$$

$$
+ \sum_{1 \leq i,j \leq n, i \neq j} \frac{3}{4} \big[ (\Delta_U)_{2j,i} + (\Delta_V)_{2i,j} \big]^2 + \sum_{1 \leq i,j \leq n, i \neq j} \frac{3}{4} \big[ (\Delta_U)_{2j-1,i}^2 + (\Delta_V)_{2j-1,i}^2 \big]
$$

$$
= \sum_{i=1}^{r} \Big\{ \frac{5}{8} \big[ (\Delta_U)_{2i-1,i} - (\Delta_V)_{2i-1,i} \big]^2 + \frac{1}{2} \big[ (\Delta_U)_{2i,i} + (\Delta_V)_{2i,i} \big]^2 \Big\}
$$

$$
+ \sum_{1 \leq i,j \leq n, i \neq j} \frac{3}{4} \big[ (\Delta_U)_{2j,i} + (\Delta_V)_{2i,j} \big]^2 + \sum_{1 \leq i,j \leq n, i \neq j} \frac{3}{4} \big[ (\Delta_U)_{2j-1,i} - (\Delta_V)_{2j-1,i} \big]^2,
$$

which is non-negative for every $\Delta \in \mathbb{R}^{n \times r}$. Hence, the point $\tilde{M}$ is a spurious second-order critical point of problem (3). Moreover, since we choose $\tilde{V} = \tilde{U}$, the point $\tilde{M}$ is a global minimizer of the regularizer $\|\tilde{U}^T \tilde{U} - \tilde{V}^T \tilde{U}\|_F^2$ and thus $\tilde{M}$ is also a spurious second-order critical point of problem (5).

# E Proofs for Section 4

## E.1 Proof of Theorem 6

In this subsection, we use the following notations:

$$M := UV^T, \; M^* := U^*(V^*)^T, \quad W := \begin{bmatrix} U \\ V \end{bmatrix}, \; W^* := \begin{bmatrix} U^* \\ V^* \end{bmatrix}, \quad \hat{W} := \begin{bmatrix} U \\ -V \end{bmatrix}, \; \hat{W}^* := \begin{bmatrix} U^* \\ -V^* \end{bmatrix},$$

where $M^* := M_a^*$ is the global optimum. We always assume that $U^*$ and $V^*$ satisfy $(U^*)^T U^* = (V^*)^T V^*$. When there is no ambiguity about $W$, we use $W^*$ to denote the minimizer of $\min_{X \in \mathcal{X}^*} \|W - X\|_F$, where $\mathcal{X}^*$ is the set of global minima of problem (5). We note that the set $\mathcal{X}^*$ is the trajectory of a global minimum $(U^*, V^*)$ under the orthogonal group:

$$\mathcal{X}^* = \{(U^*R, V^*R) \mid R \in \mathbb{R}^{r \times r}, R^T R = RR^T = I_r\}.$$

Therefore, the set $\mathcal{X}^*$ is a compact set and its minimum can be attained. With this choice, it holds that

$$\text{dist}(W, \mathcal{X}^*) = \|W - W^*\|_F.$$

We first summarize some technical results in the following lemma.

**Lemma 1** (Tu et al. (2016); Zhu et al. (2018)). *The following statements hold for every $U \in \mathbb{R}^{n \times r}$, $V \in \mathbb{R}^{m \times r}$ and $W \in \mathbb{R}^{(n+m) \times r}$:*

- $4\|M - M^*\|_F^2 \geq \|WW^T - W^*(W^*)^T\|_F^2 - \|U^T U - V^T V\|_F^2.$

- $\|W^*(W^*)^T\|_F^2 = 4\|M^*\|_F^2.$

- *If* $\text{rank}(W^*) = r$, *then* $\|WW^T - W^*(W^*)^T\|_F^2 \geq 2(\sqrt{2} - 1)\sigma_r^2(W^*)\|W - W^*\|_F^2.$

- *If* $\text{rank}(U^*) = r$, *then* $\|UU^T - U^*(U^*)^T\|_F^2 \geq 2(\sqrt{2} - 1)\sigma_r^2(U^*)\|U - U^*\|_F^2.$

The proof of Theorem 6 follows from the following sequence of lemmas. We first identify two cases when the gradient is large. The following lemma proves that an unbalanced solution cannot be a first-order critical point.

**Lemma 2.** *Given a constant $\epsilon > 0$, if*

$$\|U^T U - V^T V\|_F \geq \epsilon,$$

*then*

$$\|\nabla \rho(U, V)\|_F \geq \mu(\epsilon/r)^{3/2}.$$

*Proof.* Using the relationship between the 2-norm and the Frobenius norm, we have

$$\|U^T U - V^T V\|_2 \geq r^{-1}\|U^T U - V^T V\|_F \geq \epsilon/r.$$

Let $q \in \mathbb{R}^r$ be an eigenvector of $U^T U - V^T V$ such that

$$\|q\|_2 = 1, \quad |q^T(U^T U - V^T V)q| = \|U^T U - V^T V\|_2.$$

We consider the direction

$$\Delta := \hat{W}qq^T.$$

Then, we can calculate that

$$\|\Delta\|_F^2 = \text{tr}\left(\hat{W}qq^T qq^T \hat{W}^T\right) = \text{tr}\left(q^T \hat{W}^T \hat{W}q\right) = q^T(U^T U + V^T V)q.$$

In addition, we have

$$\langle \nabla h_a(U, V), \Delta \rangle = \left\langle \begin{bmatrix} \nabla f_a(M)V \\ [\nabla f_a(M)]^T U \end{bmatrix}, \begin{bmatrix} Uqq^T \\ -Vqq^T \end{bmatrix} \right\rangle$$
$$= \text{tr}\left[V^T[\nabla f_a(M)]^T Uqq^T\right] - \text{tr}\left[U^T \nabla f_a(M)Vqq^T\right]$$
$$= q^T\left[V^T[\nabla f_a(M)]^T U\right]q - q^T\left[U^T \nabla f_a(M)V\right]q = 0.$$

and

$$\left|\left\langle \frac{\mu}{4}\nabla g(U,V),\Delta\right\rangle\right| = \mu\left|\left\langle \hat{W}\hat{W}^TW, Wqq^T\right\rangle\right|$$

$$= \mu\left|\text{tr}\left[(U^TU - V^TV)(U^TU + V^TV)qq^T\right]\right|$$

$$= \mu\left|q^T(U^TU - V^TV)(U^TU + V^TV)q\right|$$

$$= \mu\|U^TU - V^TV\|_2 \cdot q^T(U^TU + V^TV)q$$

$$= \mu\|U^TU - V^TV\|_2 \cdot \sqrt{q^T(U^TU + V^TV)q} \cdot \|\Delta\|_F.$$

Hence, Cauchy's inequality implies that

$$\|\nabla\rho(U,V)\|_F \geq \frac{|\langle\nabla\rho(U,V),\Delta\rangle|}{\|\Delta\|_F} = \mu\|U^TU - V^TV\|_2 \cdot \sqrt{q^T(U^TU + V^TV)q}.$$

Using the fact that

$$q^T(U^TU + V^TV)q \geq \left|q^T(U^TU - V^TV)q\right| = \|U^TU - V^TV\|_2,$$

we obtain

$$\|\nabla\rho(U,V)\|_F \geq \mu\|U^TU - V^TV\|_2^{3/2} \geq \mu(\epsilon/r)^{3/2}.$$

$\square$

The next lemma proves that a solution with large norm cannot be a first-order critical point.

**Lemma 3.** *Given a constant $\epsilon > 0$, if*

$$\frac{1-\delta}{3} \leq \mu < 1 - \delta, \quad \|WW^T\|_F^{3/2} \geq \max\left\{\left(\frac{1+\delta}{1-\mu-\delta}\right)^2 \|W^*(W^*)^T\|_F^{3/2}, \frac{4\sqrt{r}\lambda}{1-\mu-\delta}\right\},$$

*then*

$$\|\nabla\rho(U,V)\|_F \geq \lambda.$$

*Proof.* Choosing the direction $\Delta := W$, we can calculate that

$$\langle\nabla\rho(U,V),\Delta\rangle = 2\langle\nabla f_a(UV^T), UV^T\rangle + \mu\|U^TU - V^TV\|_F^2. \tag{39}$$

Using the $\delta$-RIP$_{2r,2r}$ property, we have

$$[\nabla^2 f_a(N)](M,M) \geq (1-\delta)\|M\|_F^2, \quad [\nabla^2 f_a(N)](M^*,M) \leq (1+\delta)\|M\|_F\|M^*\|_F,$$

where $N \in \mathbb{R}^{n\times m}$ is every matrix with rank at most $2r$. Then, the first term can be estimated as

$$\langle\nabla f_a(UV^T), UV^T\rangle = \int_0^1 [\nabla^2 f_a(M^* + s(M - M^*))][M - M^*, M]\,ds$$

$$\geq (1-\delta)\|M\|_F^2 - (1+\delta)\|M^*\|_F\|M\|_F.$$

The second term is

$$\mu\|U^TU - V^TV\|_F^2 = \mu\left(\|UU^T\|_F^2 + \|VV^T\|_F^2\right) - 2\mu\|M\|_F^2.$$

Substituting into equation (39), it follows that

$$\langle\nabla\rho(U,V),\Delta\rangle \geq \mu\left(\|UU^T\|_F^2 + \|VV^T\|_F^2\right) + 2(1-\delta-\mu)\|M\|_F^2 - 2(1+\delta)\|M^*\|_F\|M\|_F$$

$$\geq \mu\left(\|UU^T\|_F^2 + \|VV^T\|_F^2\right) + 2(1-\delta-\mu)\|M\|_F^2 - 2c\|M\|_F^2 - \frac{(1+\delta)^2}{2c}\|M^*\|_F^2$$

$$\geq \min\{\mu, 1-\delta-\mu-c\}\|WW^T\|_F^2 - \frac{(1+\delta)^2}{2c}\|M^*\|_F^2,$$

where $c \in (0, 1-\delta-\mu)$ is a constant to be designed later. Using equality that $(U^*)^TU^* = (V^*)^TV^*$, Lemma 1 gives

$$\|W^*(W^*)^T\|_F^2 = 4\|M^*\|_F^2.$$

As a result,

$$\langle \nabla \rho(U,V), \Delta \rangle \geq \min\{\mu, 1 - \delta - \mu - c\} \|WW^T\|_F^2 - \frac{(1+\delta)^2}{8c} \|W^*(W^*)^T\|_F^2.$$

Now, choosing

$$c = \frac{1 - \delta - \mu}{2}$$

and noticing that $\mu \geq (1 - \delta - \mu)/2$, it yields that

$$\langle \nabla \rho(U,V), \Delta \rangle \geq \frac{1 - \delta - \mu}{2} \|WW^T\|_F^2 - \frac{(1+\delta)^2}{4(1 - \delta - \mu)} \|W^*(W^*)^T\|_F^2. \tag{40}$$

On the other hand,

$$\|\Delta\|_F = \|W\|_F \leq \sqrt{r} \|WW^T\|_F^{1/2}.$$

Combining with inequality (40) and using the assumption of this lemma, one can write

$$
\begin{aligned}
\|\nabla \rho(U,V)\|_F &\geq \frac{\langle \nabla \rho(U,V), \Delta \rangle}{\|\Delta\|_F} \\
&\geq \frac{1 - \delta - \mu}{2\sqrt{r}} \|WW^T\|_F^{3/2} - \frac{(1+\delta)^2}{4\sqrt{r}(1 - \delta - \mu)} \|W^*(W^*)^T\|_F^2 \|WW^T\|_F^{-1/2} \\
&\geq \frac{1 - \delta - \mu}{2\sqrt{r}} \|WW^T\|_F^{3/2} - \frac{(1+\delta)^2}{4\sqrt{r}(1 - \delta - \mu)} \|W^*(W^*)^T\|_F^{3/2} \\
&\geq \frac{1 - \delta - \mu}{4\sqrt{r}} \|WW^T\|_F^{3/2} \geq \lambda.
\end{aligned}
$$

$\square$

Using the above two lemmas, we only need to focus on points such that

$$\|U^TU - V^TV\|_F = o(1), \quad \|WW^T\|_F = O(1).$$

The following lemma proves that if $(U,V)$ is an approximate first-order critical point with a small singular value $\sigma_r(W)$, then the Hessian of the objective function at this point has a negative curvature.

**Lemma 4.** *Consider positive constants $\alpha, C, \epsilon, \lambda$ such that*

$$\epsilon^2 \leq (\sqrt{2} - 1)\sigma_r^2(W^*) \cdot \alpha^2, \quad G > \mu\left(\epsilon + \frac{4H^2}{G^2}\right) + \frac{(1+\delta)H^2}{G^2}, \tag{41}$$

*where $G := \|\nabla f_a(M)\|_2$ and $H := \lambda + \mu\epsilon C$. If*

$$\|U^TU - V^TV\|_F^2 \leq \epsilon^2, \quad \|WW^T\|_F \leq C^2, \quad \|W - W^*\|_F \geq \alpha, \quad \|\nabla \rho(U,V)\|_F \leq \lambda$$

*and*

$$\sigma_r^2(W) \leq \frac{2}{1+\delta}\left[G - \mu\left(\epsilon + \frac{4H^2}{G^2}\right) - \frac{(1+\delta)H^2}{G^2}\right] - 2\tau \tag{42}$$

*for some positive constant $\tau$, then it holds that*

$$\lambda_{min}(\nabla^2 \rho(U,V)) \leq -(1+\delta)\tau.$$

*Proof.* We choose a singular vector $q$ of $W$ such that

$$\|q\|_2 = 1, \quad \|Wq\|_2 = \sigma_r(W).$$

Since $\|Wq\|_2 = \sqrt{\|Uq\|_2^2 + \|Vq\|_2^2}$, we have

$$\|Uq\|_2^2 + \|Vq\|_2^2 = \sigma_r^2(W).$$

We choose singular vectors $u$ and $v$ such that

$$\|u\|_2 = \|v\|_2 = 1, \quad \|\nabla f_a(M)\|_2 = u^T \nabla f_a(M)v.$$

We define the direction as

$$\Delta_U := -uq^T, \quad \Delta_V := vq^T, \quad \Delta := \begin{bmatrix} \Delta_U \\ \Delta_V \end{bmatrix}, \quad \hat{\Delta} := \begin{bmatrix} \Delta_U \\ -\Delta_V \end{bmatrix}.$$

For the Hessian of $h_a(\cdot, \cdot)$, we can calculate that

$$\langle \nabla f_a(M), \Delta_U \Delta_V^T \rangle = -\|\nabla f_a(M)\|_2 = -G \tag{43}$$

and the $\delta$-RIP$_{2r,2r}$ property gives

$$
\begin{aligned}
[\nabla^2 f_a(M)](\Delta_U V^T & + U \Delta_V^T, \Delta_U V^T + U \Delta_V^T) \\
& \leq (1+\delta)\|\Delta_U V^T + U \Delta_V^T\|_F^2 = (1+\delta)\| - u(Vq)^T + (Uq)v^T\|_F^2 \\
& = (1+\delta)\left(\|Vq\|_F^2 + \|Uq\|_F^2\right) - 2(1+\delta)[q^T(U^T u)] \cdot [q^T(V^T v)] \\
& \leq (1+\delta)\sigma_r^2(W) + 2(1+\delta) \cdot \|U^T u\|_F \|V^T v\|_F.
\end{aligned}
\tag{44}
$$

Then, we consider the terms coming from the Hessian of the regularizer. First, we have

$$
\begin{aligned}
\langle \hat{\Delta}\hat{W}^T, \Delta W^T \rangle & \leq \|U^T U - V^T V\|_F \cdot \|\Delta_U^T \Delta_U - \Delta_V^T \Delta_V\|_F \\
& \leq \epsilon \cdot \left[\|\Delta_U^T \Delta_U\|_F + \|\Delta_V^T \Delta_V\|_F\right] = 2\epsilon.
\end{aligned}
\tag{45}
$$

Next, we can estimate that

$$
\begin{aligned}
\langle \hat{W}\hat{\Delta}^T, \Delta W^T \rangle + \langle \hat{W}\hat{W}^T, \Delta \Delta^T \rangle & = \frac{1}{2}\|U^T \Delta_U + \Delta_U^T U - V^T \Delta_V - \Delta_V^T V\|_F^2 \\
& \leq 4\left(\|U^T \Delta_U\|_F^2 + \|V^T \Delta_V\|_F^2\right) \\
& = 4\left(\|(U^T u)q^T\|_F^2 + \|(V^T v)q^T\|_F^2\right) \\
& = 4\left(\|U^T u\|_F^2 + \|V^T v\|_F^2\right).
\end{aligned}
\tag{46}
$$

Using the assumption that $\|WW^T\|_F \leq C^2$ and $\|U^T U - V^T V\|_F^2 \leq \epsilon^2$, one can write

$$\|\hat{W}\hat{W}^T W\|_F^2 \leq \|U^T U - V^T V\|_F^2 \cdot \|U^T U + V^T V\|_F \leq \epsilon^2 \|WW^T\|_F \leq \epsilon^2 C^2$$

and

$$\left\| \begin{bmatrix} \nabla f_a(UV^T)V \\ \nabla f_a(UV^T)^T U \end{bmatrix} \right\|_F = \|\nabla\rho(U,V) - \mu\hat{W}\hat{W}^T W\|_F \leq \lambda + \mu\epsilon C = H. \tag{47}$$

The second relation implies that

$$\|\nabla f_a(UV^T)V\|_2 \leq \|\nabla f_a(UV^T)V\|_F \leq H, \quad \|U^T \nabla f_a(UV^T)\|_2 \leq \|U^T \nabla f_a(UV^T)\|_F \leq H. \tag{48}$$

By the definition of $u$ and $v$, it holds that

$$\|v\|_2 = 1, \quad \|\nabla f_a(M)\|_2 u = \nabla f_a(M)v.$$

Therefore,

$$\|U^T u\|_F^2 = \frac{\|U^T \nabla f_a(M)v\|_F^2}{\|\nabla f_a(M)\|_2^2} \leq \frac{\|U^T \nabla f_a(M)\|_F^2 \|v\|_2^2}{\|\nabla f_a(M)\|_2^2} \leq \frac{H^2}{G^2}.$$

Similarly,

$$\|V^T v\|_F^2 \leq \frac{H^2}{G^2}.$$

Substituting into (44) and (46) yields that

$$[\nabla^2 f_a(M)](\Delta_U V^T + U\Delta_V^T, \Delta_U V^T + U\Delta_V^T) \leq (1+\delta)\sigma_r^2(W) + 2(1+\delta) \cdot \frac{H^2}{G^2} \tag{49}$$

and

$$\langle \hat{W}\hat{\Delta}^T, \Delta W^T \rangle + \langle \hat{W}\hat{W}^T, \Delta\Delta^T \rangle \leq 8 \cdot \frac{H^2}{G^2}. \tag{50}$$

Combining (43), (45), (49) and (50), it follows that

$$[\nabla^2 \rho(U,V)](\Delta,\Delta) \leq -2G + (1+\delta)\sigma_r^2(W) + 2\mu\epsilon + [8\mu + 2(1+\delta)] \cdot \frac{H^2}{G^2}.$$

Since $\|\Delta\|_F^2 = 2$, the above relation implies

$$\lambda_{min}(\nabla^2 \rho(U,V)) \leq -G + \frac{1+\delta}{2}\sigma_r^2(W) + \mu\epsilon + (4\mu + 1 + \delta) \cdot \frac{H^2}{G^2} \leq -(1+\delta)\tau.$$

$\square$

*Remark* 2. The positive constants $\epsilon$ and $\lambda$ in the proof of Lemma 4 can be chosen to be arbitrarily small with $\alpha, C$ fixed. Hence, we may choose small enough $\epsilon$ and $\lambda$ such that the assumptions given in inequality (41) are satisfied. This lemma resolves the case when the minimal singular value $\sigma_r^2(W)$ is on the order of $\|\nabla f_a(M)\|_2/(2+2\delta)$. In the next lemma, we will show that this is the only case when $\delta < 1/3$.

The final step is to prove that condition (42) always holds provided that $\delta < 1/3$ and $\epsilon, \lambda, \tau = o(1)$.

**Lemma 5.** *Given positive constants $\alpha, C, \epsilon, \lambda$, if*

$$\|U^T U - V^T V\|_F^2 \leq \epsilon^2, \quad \max\{\|WW^T\|_F, \|W^*(W^*)^T\|_F\} \leq C^2,$$
$$\|W - W^*\|_F \geq \alpha, \quad \|\nabla\rho(U,V)\|_F \leq \lambda, \quad \delta < 1/3,$$

*then the inequality $G \geq c\alpha$ holds for some constant $c > 0$ independent of $\alpha, \epsilon, \lambda, C$. Furthermore, there exist two positive constants*

$$\epsilon_0(\delta, \mu, \sigma_r(M_a^*), \|M_a^*\|_F, \alpha, C), \quad \lambda_0(\delta, \mu, \sigma_r(M_a^*), \|M_a^*\|_F, \alpha, C)$$

*such that*

$$\sigma_r^2(W) \leq \frac{2}{1+\delta}\left[G - \mu\left(2\epsilon + \frac{4H^2}{G^2}\right) - \frac{(1+\delta)H^2}{G^2}\right] \tag{51}$$

*whenever*

$$0 < \epsilon \leq \epsilon_0(\delta, \mu, \sigma_r(M_a^*), \|M_a^*\|_F, \alpha, C),$$
$$0 < \lambda \leq \lambda_0(\delta, \mu, \sigma_r(M_a^*), \|M_a^*\|_F, \alpha, C).$$

*Here, $G$ and $H$ are defined in Lemma 4.*

*Proof.* We first prove the existence of the constant $c$. Using Lemma 1, one can write
$$4\|M - M^*\|_F^2 \geq \|WW^T - W^*(W^*)^T\|_F^2 - \|U^T U - V^T V\|_F^2 \geq \|WW^T - W^*(W^*)^T\|_F^2 - \epsilon^2.$$
Using Lemma 1 and the assumption that $\|W - W^*\|_F \geq \alpha$, we have

$$\|M - M^*\|_F^2 \geq \frac{\sqrt{2}-1}{2}\sigma_r^2(W^*)\|W - W^*\|_F^2 - \frac{\epsilon^2}{4} \geq \frac{\sqrt{2}-1}{2}\sigma_r^2(W^*) \cdot \alpha^2 - \frac{\epsilon^2}{4}. \tag{52}$$

By the definition of $\epsilon$, it follows that

$$\|M - M^*\|_F^2 \geq \frac{\sqrt{2}-1}{4}\sigma_r^2(W^*) \cdot \alpha^2 > 0.$$

Thus, the $\delta$-RIP$_{2r,2r}$ property gives

$$\|\nabla f_a(M)\|_F \geq \frac{\langle \nabla f_a(M), M - M^*\rangle}{\|M - M^*\|_F} \geq (1-\delta)\|M - M^*\|_F \geq \sqrt{\frac{\sqrt{2}-1}{4}} \cdot \sigma_r(W^*)(1-\delta) \cdot \alpha.$$

Hence, we have

$$G = \|\nabla f_a(M)\|_2 \geq \sqrt{\frac{\sqrt{2}-1}{4r}} \cdot \sigma_r(W^*)(1-\delta) \cdot \alpha = c\alpha,$$

where we define

$$c := \sqrt{\frac{\sqrt{2}-1}{4r}} \cdot \sigma_r(W^*)(1-\delta).$$

Next, we prove inequality (51) by contradiction, i.e., we assume

$$\sigma_r^2(W) > \frac{2}{1+\delta}\left[G - \mu\left(2\epsilon + \frac{4H^2}{G^2}\right) - \frac{(1+\delta)H^2}{G^2}\right] \geq \frac{2c\alpha}{1+\delta} + \text{poly}(\epsilon, \lambda). \tag{53}$$

The remainder of the proof is divided into three steps.

**Step I.** We first develop a lower bound for $\sigma_r(M)$. We choose a vector $p \in \mathbb{R}^r$ such that

$$\|p\|_F = 1, \quad U^T U p = \sigma_r^2(U) \cdot p.$$

It can be shown that

$$\|(Wp)^T W\|_F = \|p^T U^T U + p^T V^T V\|_F \leq 2\|p^T U^T U\|_F + \|p^T (V^T V - U^T U)\|_F$$
$$\leq 2\sigma_r^2(U) + \|p^T\|_F \|V^T V - U^T U\|_F \leq 2\sigma_r^2(U) + \epsilon.$$

On the other hand, since $W$ has rank $r$, it holds that

$$\left\|(Wp)^T W\right\|_F \geq \sigma_r^2(W) \cdot \|p\|_F = \sigma_r^2(W).$$

Combining the above two estimates, we arrive at

$$2\sigma_r^2(U) \geq \sigma_r^2(W) - \epsilon > 0,$$

where the last inequality is from the assumption that $\epsilon, \lambda$ are small and $\sigma_r(W)$ is lower bounded by a positive value in (53). Using the inequality that $\sqrt{1-x} \geq 1 - x$ for every $x \in [0,1]$, the above inequality implies that

$$\sigma_r(U) \geq \frac{1}{\sqrt{2}}\sigma_r(W) \cdot \sqrt{1 - \frac{\epsilon}{\sigma_r^2(W)}} \geq \frac{1}{\sqrt{2}}\sigma_r(W) - \frac{\epsilon}{\sqrt{2}\sigma_r(W)}. \tag{54}$$

Similarly, one can prove that

$$\sigma_r(V) \geq \frac{1}{\sqrt{2}}\sigma_r(W) - \frac{\epsilon}{\sqrt{2}\sigma_r(W)}.$$

When $\epsilon$ is small enough, we know that $\sigma_r(U), \sigma_r(V) \neq 0$ and both $U, V$ have rank $r$. To lower bound the singular value $\sigma_r(M)$, we consider vectors $x$ such that $\|x\|_2 = 1$ and lower bound $x^T V(U^T U)V^T x$. Since the range of $V(U^T U)V^T$ is a subspace of the range of $V$ and the range of $V$ has exactly dimension $r$, directions $x$ that are in the orthogonal complement of the range of $V$ correspond to exactly $m - r$ zero singular values. Hence, to estimate the $r$-th largest singular value of $M$, we only need to consider directions that are in the range of $V$. Namely, we only consider directions that have the form $x = Vy$ for some vector $y$. Then, we have

$$x^T V(U^T U)V^T x = y^T (V^T V)(U^T U)(V^T V)y$$
$$= y^T (V^T V)^3 y + y^T (V^T V)(U^T U - V^T V)(V^T V)y.$$

First, we bound the second term by calculating that

$$\|V(V^T V - U^T U)V^T\|_2 \leq \|V\|_2^2 \|U^T U - V^T V\|_2 \leq \|V^T V\|_F \|U^T U - V^T V\|_F$$
$$\leq \|W^T W\|_F \|U^T U - V^T V\|_F \leq C^2\epsilon.$$

This implies that

$$y^T (V^T V)(U^T U - V^T V)(V^T V)y \geq -C^2\epsilon \cdot \|Vy\|_F^2.$$

Next, we assume that $y$ has the decomposition

$$y = \sum_{i=1}^{r} c_i v_i,$$

where $v_i$ is an eigenvector of $V^T V$ associated with the eigenvalue $\sigma_i^2(V)$. Then, we can calculate that

$$y^T (V^T V)^3 y = \sum_{i=1}^{r} c_i^2 \sigma_i^6(V), \quad \|Vy\|_F^2 = \sum_{i=1}^{r} c_i^2 \sigma_i^2(V) = 1.$$

Combining the above estimates leads to

$$x^T V(U^T U)V^T x \geq \left[\frac{\sum_{i=1}^{r} c_i^2 \sigma_i^6(V)}{\sum_{i=1}^{r} c_i^2 \sigma_i^2(V)} - C^2\epsilon\right] \cdot \|Vy\|_F^2$$
$$= \frac{\sum_{i=1}^{r} c_i^2 \sigma_i^6(V)}{\sum_{i=1}^{r} c_i^2 \sigma_i^2(V)} - C^2\epsilon \geq \sigma_r^4(V) - C^2\epsilon.$$

This implies that

$$\sigma_r^2(M) \geq \sigma_r^4(V) - C^2\epsilon \geq \left[\frac{1}{\sqrt{2}}\sigma_r(W) - \frac{\epsilon}{\sqrt{2}\sigma_r(W)}\right]^4 - C^2\epsilon$$

$$\geq \frac{1}{4}\sigma_r^4(W) - \sigma_r^2(W)\epsilon - \sigma_r^{-2}(W)\epsilon^3 - C^2\epsilon$$

$$\geq \frac{1}{4}\sigma_r^4(W) - \sigma_r^{-2}(W)\epsilon^3 - 2C^2\epsilon$$

$$\geq \frac{1}{4}\sigma_r^4(W) - \frac{1+\delta}{G} \cdot \epsilon^3 - 2C^2\epsilon$$

$$\geq \frac{1}{4}\sigma_r^4(W) - \frac{1+\delta}{c\alpha} \cdot \epsilon^3 - 2C^2\epsilon. \tag{55}$$

where the second last inequality is due to (53) and the assumption that $\epsilon$ and $\lambda$ are sufficiently small.

**Step II.** Next, we derive an upper bound for $\sigma_r(M)$. We define

$$\bar{M} := \mathcal{P}_r\left[M - \frac{1}{1+\delta}\nabla f_a(M)\right],$$

where $\mathcal{P}_r$ is the orthogonal projection onto the low-rank set via SVD. Since $M \neq M^*$ and $\delta < 1/3$, we recall that inequality (13) gives

$$-\phi(\bar{M}) \geq \frac{1-3\delta}{1-\delta}[f_a(M) - f_a(M^*)] \geq \frac{1-3\delta}{2}\|M - M^*\|_F^2$$

$$\geq \frac{1-3\delta}{2}\left[\frac{\sqrt{2}-1}{2}\sigma_r^2(W^*)\alpha^2 - \frac{\epsilon^2}{4}\right] := K,$$

where the second inequality follows from (52) and

$$-\phi(\bar{M}) = \langle \nabla f_a(M), M - \bar{M}\rangle - \frac{1+\delta}{2}\|M - \bar{M}\|_F^2.$$

Hence,

$$\langle \nabla f_a(M), M - \bar{M}\rangle - \frac{1+\delta}{2}\|M - \bar{M}\|_F^2 \geq K. \tag{56}$$

When we choose $\epsilon$ to be small enough, it holds that $K > 0$. For simplicity, we define

$$N := -\frac{1}{1+\delta}\nabla f_a(M).$$

Then, $\bar{M} = \mathcal{P}_r(M + N)$ and the left-hand side of (56) is equal to

$$\langle \nabla f_a(M), M - \bar{M}\rangle - \frac{1+\delta}{2}\|M - \bar{M}\|_F^2$$

$$= (1+\delta)\langle N, \mathcal{P}_r(M+N) - M\rangle - \frac{1+\delta}{2}\|\mathcal{P}_r(M+N) - M\|_F^2$$

$$= \frac{1+\delta}{2}\left[\|N\|_F^2 - \|N + M - \mathcal{P}_r(M+N)\|_F^2\right]$$

$$= \frac{1+\delta}{2}\left[\|N\|_F^2 - \|N + M\|_F^2 + \|\mathcal{P}_r(M+N)\|_F^2\right]. \tag{57}$$

Similar to the proof of inequality (48), we can prove that

$$\|NV\|_F \leq \tilde{H} := \frac{H}{1+\delta}, \quad \|U^T N\|_F \leq \tilde{H}.$$

Then, we have

$$-\operatorname{tr}[N^T(UV^T)] \leq \|U^T N\|_F\|V\|_F \leq \tilde{H} \cdot \|W\|_F \leq \tilde{H} \cdot \sqrt{\sqrt{r}\|WW^T\|_F} \leq \sqrt[4]{r}C \cdot \tilde{H}.$$

Using the above relation, we obtain

$$\|N\|_F^2 - \|N + M\|_F^2 = -2\operatorname{tr}[N^T(UV^T)] - \|M\|_F^2 \leq 2\sqrt[4]{r}C \cdot \tilde{H} - \|M\|_F^2.$$

Suppose that $\mathcal{P}_U$ and $\mathcal{P}_V$ are the orthogonal projections onto the column spaces of $U$ and $V$, respectively. We define

$$N_1 := \mathcal{P}_U N \mathcal{P}_V, \quad N_2 := \mathcal{P}_U N (I - \mathcal{P}_V), \quad N_3 := (I - \mathcal{P}_U) N \mathcal{P}_V, \quad N_4 := (I - \mathcal{P}_U) N (I - \mathcal{P}_V).$$

Then, recalling the assumption (53) and inequality (54), it follows that

$$\|N_1\|_F = \|\mathcal{P}_U N \mathcal{P}_V\|_F \leq \sigma_r^{-1}(U)\|U^T \mathcal{P}_U N \mathcal{P}_V\|_F \leq \sigma_r^{-1}(U)\|U^T N\|_F \leq \frac{\sqrt{2}\sigma_r(W)}{\sigma_r^2(W) - \epsilon} \cdot \tilde{H}$$

$$\leq \left[\sqrt{\frac{1+\delta}{G}} + \operatorname{poly}(\epsilon, \lambda)\right] \cdot \tilde{H} \leq \left[\sqrt{\frac{1+\delta}{c\alpha}} + \operatorname{poly}(\epsilon, \lambda)\right] \cdot \tilde{H} := \kappa\tilde{H}.$$

Similarly, we can prove that

$$\|N_1 + N_2\|_F = \|\mathcal{P}_U N\|_F \leq \kappa\tilde{H}, \quad \|N_1 + N_3\|_F = \|N\mathcal{P}_V\|_F \leq \kappa\tilde{H},$$

which leads to

$$\|N_2\|_F \leq 2\kappa\tilde{H}, \quad \|N_3\|_F \leq 2\kappa\tilde{H}.$$

Using Weyl's theorem, the following holds for every $1 \leq i \leq r$:

$$|\sigma_i(M + N) - \sigma_i(M + N_4)| \leq \|N_1 + N_2 + N_3\|_2 \leq \|N_1 + N_2 + N_3\|_F \leq 3\kappa\tilde{H}.$$

Therefore, we have

$$\|\mathcal{P}_r(M + N)\|_F^2 = \sum_{i=1}^r \sigma_i^2(M + N)$$

$$\geq \sum_{i=1}^r \sigma_i^2(M + N_4) - r \cdot 3\kappa\tilde{H} \cdot (\|M + N\|_2 + \|M + N_4\|_2)$$

$$\geq \sum_{i=1}^r \sigma_i^2(M + N_4) - 6r\kappa\tilde{H} \cdot (\|M\|_2 + \|N\|_2)$$

$$\geq \sum_{i=1}^r \sigma_i^2(M + N_4) - 6r\kappa\tilde{H} \cdot \left(\|M\|_F + \frac{G}{1+\delta}\right). \tag{58}$$

Using the assumption (53) and the inequality (55), one can write

$$\frac{G}{1+\delta} \leq \frac{\sigma_r^2(W)}{2} + \operatorname{poly}(\sqrt{\epsilon}, \lambda) \leq \sigma_r(M) + \operatorname{poly}(\sqrt{\epsilon}, \lambda) \leq \|M\|_F + \operatorname{poly}(\sqrt{\epsilon}, \lambda), \tag{59}$$

where $\operatorname{poly}(\sqrt{\epsilon}, \lambda)$ means a polynomial of $\sqrt{\epsilon}$ and $\lambda$. Therefore, we attain the bound

$$\|M\|_F + \|N\|_F \leq 2\|M\|_F + \operatorname{poly}(\sqrt{\epsilon}, \lambda) \leq 2 \cdot \frac{\|WW^T\|_F}{\sqrt{2}} + \operatorname{poly}(\sqrt{\epsilon}, \lambda)$$

$$\leq \sqrt{2}C^2 + \operatorname{poly}(\sqrt{\epsilon}, \lambda). \tag{60}$$

Substituting back into the previous estimate (58), it follows that

$$\|\mathcal{P}_r(M+N)\|_F^2 \geq \sum_{i=1}^r \sigma_i^2(M+N_4) - 6\sqrt{2}r\kappa\tilde{H}C^2 + \operatorname{poly}(\sqrt{\epsilon}, \lambda) = \sum_{i=1}^r \sigma_i^2(M+N_4) + \operatorname{poly}(\sqrt{\epsilon}, \lambda).$$

Now, since $M$ and $N_4$ have orthogonal column and row spaces, the maximal $r$ singular values of $M + N_4$ are simply the maximal $r$ singular values of the singular values $M$ and $N_4$, which we assume to be

$$\sigma_i(M), \ i = 1, \ldots, k \quad \text{and} \quad \sigma_i(N_4), \ i = 1, \ldots, r - k.$$

Now, it follows from (57) that

$$\frac{2}{1+\delta}\left[\langle \nabla f_a(M), M - \bar{M}\rangle - \frac{1+\delta}{2}\|M - \bar{M}\|_F^2\right]$$

$$= \|N\|_F^2 - \|N + M\|_F^2 + \|\mathcal{P}_r(M + N)\|_F^2$$

$$\leq -\sum_{i=1}^{r} \sigma_i^2(M) + \sum_{i=1}^{k} \sigma_i^2(M) + \sum_{i=1}^{r-k} \sigma_i^2(N_4) + \mathrm{poly}(\sqrt{\epsilon}, \lambda) + 2\sqrt[4]{r}C \cdot \tilde{H}$$

$$= -\sum_{i=k+1}^{r} \sigma_i^2(M) + \sum_{i=1}^{r-k} \sigma_i^2(N_4) + \mathrm{poly}(\sqrt{\epsilon}, \lambda)$$

$$\leq -(r-k)\sigma_r^2(M) + (r-k)\|N_4\|_2^2 + \mathrm{poly}(\sqrt{\epsilon}, \lambda)$$

$$\leq -(r-k)\sigma_r^2(M) + (r-k)\|N\|_2^2 + \mathrm{poly}(\sqrt{\epsilon}, \lambda).$$

If $k = r$, then the above inequality and inequality (56) imply that

$$\mathrm{poly}(\sqrt{\epsilon}, \lambda) \geq K = O(\alpha^2),$$

which contradicts the assumption that $\epsilon$ and $\lambda$ are small. Hence, it can be concluded that $r - k \geq 1$. Combining with (56), we obtain the upper bound

$$\sigma_r^2(M) \leq -\frac{2}{1+\delta} \cdot \frac{K}{r-k} + \|N\|_2^2 + \frac{1}{r-k} \cdot \mathrm{poly}(\sqrt{\epsilon}, \lambda)$$

$$= -\frac{2}{1+\delta} \cdot \frac{K}{r} + \|N\|_2^2 + \mathrm{poly}(\sqrt{\epsilon}, \lambda). \tag{61}$$

**Step III.** In the last step, we combine the inequalities (55) and (61), which leads to

$$\frac{1}{4}\sigma_r^4(W) - \frac{1+\delta}{c\alpha} \cdot \epsilon^3 - 2C^2\epsilon \leq -\frac{2}{1+\delta} \cdot \frac{K}{r} + \frac{1}{(1+\delta)^2}G^2 + \mathrm{poly}(\sqrt{\epsilon}, \lambda).$$

This means that

$$\sigma_r^4(W) + \frac{8}{1+\delta} \cdot \frac{K}{r} \leq \frac{4}{(1+\delta)^2}G^2 + \mathrm{poly}(\sqrt{\epsilon}, \lambda).$$

Since $K > 0$ has lower bounds that are independent of $\epsilon$ and $\lambda$, we can choose $\epsilon$ and $\lambda$ to be small enough such that

$$\sigma_r^4(W) + \frac{4}{1+\delta} \cdot \frac{K}{r} \leq \frac{4}{(1+\delta)^2}G^2.$$

However, recalling the assumption (53), we have

$$\sigma_r^4(W) > \frac{4}{(1+\delta)^2}\left[G - \mu\left(2\epsilon + \frac{4H^2}{G^2}\right) - \frac{(1+\delta)H^2}{G^2}\right]^2$$

$$\geq \frac{4}{(1+\delta)^2}G^2 - \frac{16}{(1+\delta)^2}G \cdot \mu\epsilon + \mathrm{poly}(\sqrt{\epsilon}, \lambda)$$

$$\geq \frac{4}{(1+\delta)^2}G^2 - \frac{16}{(1+\delta)^2}\mu\epsilon \cdot \frac{1}{\sqrt{2}}(1+\delta)C^2 + \mathrm{poly}(\sqrt{\epsilon}, \lambda)$$

$$= \frac{4}{(1+\delta)^2}G^2 + \mathrm{poly}(\sqrt{\epsilon}, \lambda),$$

where in the third inequality we use inequalities (59)-(60) to conclude that

$$G \leq (1+\delta)\|M\|_F + \mathrm{poly}(\sqrt{\epsilon}, \lambda) \leq \frac{1}{\sqrt{2}}(1+\delta)C^2 + \mathrm{poly}(\sqrt{\epsilon}, \lambda).$$

The above two inequalities cannot hold simultaneously when $\lambda$ and $\epsilon$ are small enough. This contradiction means that the condition (51) holds by choosing

$$0 < \epsilon \leq \epsilon_0(\delta, \mu, \sigma_r(M_a^*), \|M_a^*\|_F, \alpha, C),$$
$$0 < \lambda \leq \lambda_0(\delta, \mu, \sigma_r(M_a^*), \|M_a^*\|_F, \alpha, C),$$

for some small enough positive constants

$$\epsilon_0(\delta, \mu, \sigma_r(M_a^*), \|M_a^*\|_F, \alpha, C), \quad \lambda_0(\delta, \mu, \sigma_r(M_a^*), \|M_a^*\|_F, \alpha, C).$$

$\square$

The only thing left is to piecing everything together.

*Proof of Theorem 6.* We first choose

$$C := \left[ \left( \frac{1+\delta}{1-\mu-\delta} \right)^2 \|W^*(W^*)^T\|_F^{3/2} \right]^{1/3}.$$

Then, we select $\epsilon_1$ and $\lambda_1$ as

$$\epsilon_1(\delta, r, \mu, \sigma_r(M_a^*), \|M_a^*\|_F, \alpha) := \epsilon_0(\delta, r, \mu, \sigma_r(M_a^*), \|M_a^*\|_F, \alpha, C),$$

$$\lambda_1(\delta, r, \mu, \sigma_r(M_a^*), \|M_a^*\|_F, \alpha) := \min \left\{ \lambda_0(\delta, r, \mu, \sigma_r(M_a^*), \|M_a^*\|_F, \alpha, C), \right.$$

$$\left. \frac{(1-\mu-\delta)C^3}{4\sqrt{r}} \right\}.$$

Finally, we combine Lemmas 4-5 to get the bounds for the gradient and the Hessian. $\square$

### E.2 Proof of Theorem 7

In this subsection, we use similar notations:

$$M := UU^T, \quad M^* := U^*(U^*)^T,$$

where $M^* := M_s^*$ is the global optimum. We also assume that $U^*$ is the minimizer of $\min_{X \in \mathcal{X}^*} \|U - X\|_F$ when there is no ambiguity about $U$. In this case, the distance is given by

$$\text{dist}(U, \mathcal{X}^*) = \|U - U^*\|_F.$$

The proof of Theorem 7 is similar to that of Theorem 6. We first consider the case when $\|UU^T\|_F$ is large.

**Lemma 6.** *Given a constant $\epsilon > 0$, if*

$$\|UU^T\|_F^2 \geq \max \left\{ \frac{2(1+\delta)}{1-\delta} \|U^*(U^*)^T\|_F^2, \left( \frac{2\lambda\sqrt{r}}{1-\delta} \right)^{4/3} \right\},$$

*then*

$$\|\nabla h_s(U)\|_F \geq \lambda.$$

*Proof.* Choosing the direction $\Delta := U$, we can calculate that

$$\langle \nabla h_s(U), \Delta \rangle = \langle \nabla f_s(UU^T), UU^T \rangle.$$

Using the $\delta$-RIP$_{2r,2r}$ property, we have

$$\langle \nabla f_s(UU^T), UU^T \rangle = \int_0^1 [\nabla^2 f_s(M^* + s(M - M^*)][M - M^*, M]$$
$$\geq (1-\delta)\|M\|_F^2 - (1+\delta)\|M^*\|_F\|M\|_F$$
$$\geq \frac{1-\delta}{2}\|M\|_F^2.$$

Moreover,

$$\|\Delta\|_F = \|U\|_F \leq \sqrt{r}\|UU^T\|_F^{1/2}.$$

This leads to

$$\|\nabla h_s(U)\|_F \geq \frac{\langle \nabla h_s(U), \Delta \rangle}{\|\Delta\|_F} = \frac{\langle \nabla f_s(UU^T), UU^T \rangle}{\|U\|_F} \geq \frac{1-\delta}{2\sqrt{r}}\|UU^T\|_F^{3/2} \geq \lambda.$$

$\square$

The next lemma is a counterpart of Lemma 4.

**Lemma 7.** *Consider positive constants $\alpha, C, \lambda$ such that*

$$\lambda \leq 2(\sqrt{r}C)^{-1}(\sqrt{2}-1)\sigma_r^2(U^*) \cdot \alpha^2, \quad G > \frac{(1+\delta)\lambda^2}{4G^2},$$

*where $G := -\lambda_{min}(\nabla f_s(M))$. If*

$$\|UU^T\|_F \leq C^2, \quad \|U - U^*\|_F \geq \alpha, \quad \|\nabla h_s(U)\|_F \leq \lambda,$$

*then the inequality $G \geq c\alpha^2$ holds for some constant $c > 0$ independent of $\alpha, \lambda, C$. Moreover, if there exists some positive constant $\tau$ such that*

$$\sigma_r^2(U) \leq \frac{1}{1+\delta}\left[G - \frac{(1+\delta)\lambda^2}{4G^2}\right] - \tau, \tag{62}$$

*then*

$$\lambda_{min}(\nabla^2 h_s(U)) \leq -2(1+\delta)\tau.$$

*Proof.* We choose a singular vector $q$ of $U$ such that

$$\|q\|_2 = 1, \quad \|Uq\|_2 = \sigma_r(U).$$

We first prove the existence of the constant $c$. The $\delta$-$\text{RIP}_{2r,2r}$ property gives

$$\langle \nabla f_s(M), M^* - M \rangle \leq -(1-\delta)\|M - M^*\|_F^2.$$

Using the assumption of this lemma, we have

$$\|\nabla f_s(M)U\|_2 \leq \|\nabla f_s(M)U\|_F = \frac{1}{2}\|\nabla h_s(U)\|_F \leq \frac{1}{2}\lambda, \tag{63}$$

which leads to

$$\langle \nabla f_s(M), M \rangle = \langle \nabla f_s(M)U, U \rangle \leq \|\nabla f_s(M)U\|_F \|U\|_F \leq \frac{1}{2}\lambda \cdot \sqrt{r}C.$$

Substituting into (63), it follows that

$$\langle \nabla f_s(M), M^* \rangle \leq -(1-\delta)\|M - M^*\|_F^2 + \frac{1}{2}\lambda \cdot \sqrt{r}C.$$

Using Lemma 1, we have

$$\|M - M^*\|_F^2 \geq 2(\sqrt{2}-1)\sigma_r^2(U^*)\|U - U^*\|_F^2 \geq 2(\sqrt{2}-1)\sigma_r^2(U^*) \cdot \alpha^2.$$

By the condition on $\lambda$, it follows that

$$\langle \nabla f_s(M), M^* \rangle \leq -(1-\delta)\|M - M^*\|_F^2 + \frac{1}{2}\lambda \cdot \sqrt{r}C \leq -(\sqrt{2}-1)(1-\delta)\sigma_r^2(U^*) \cdot \alpha^2. \tag{64}$$

The above inequality also indicates that $\lambda_{min}(\nabla f_s(M)) < 0$. Using the relations that

$$\nabla f_s(M) \succeq \lambda_{min}(\nabla f_s(M)) \cdot I_n, \quad M^* \succeq 0,$$

we arrive at

$$\langle \nabla f_s(M), M^* \rangle \geq \lambda_{min}(\nabla f_s(M)) \operatorname{tr}(M^*) \geq \sqrt{r}\|M^*\|_F \cdot \lambda_{min}(\nabla f_s(M)).$$

Combining the last inequality with (64), we obtain

$$\lambda_{min}(\nabla f_s(M)) \leq -(\sqrt{r}\|M^*\|_F)^{-1}(\sqrt{2}-1)(1-\delta)\sigma_r^2(U^*) \cdot \alpha^2 = -c\alpha^2$$

and thus $G \geq c\alpha^2$, where

$$c := (\sqrt{r}\|M^*\|_F)^{-1}(\sqrt{2}-1)(1-\delta)\sigma_r^2(U^*)$$

Next, we prove the upper bound on the minimal eigenvalue. We choose an eigenvector $u$ such that

$$\|u\|_2 = 1, \quad \lambda_{min}(\nabla f_s(M)) = u^T \nabla f_s(M)u.$$

The direction is chosen to be
$$\Delta := uq^T.$$
For the Hessian of $h_s(\cdot, \cdot)$, we can calculate that
$$\langle \nabla f_s(M), \Delta\Delta^T \rangle = \lambda_{min}(\nabla f_s(M)) = -G \tag{65}$$
and the $\delta$-RIP$_{2r,2r}$ property gives
$$
\begin{aligned}
[\nabla^2 f_s(M)](\Delta U^T + U\Delta^T, \Delta U^T + U\Delta^T) & \\
\leq (1+\delta)\|\Delta U^T + U\Delta^T\|_F^2 &= (1+\delta)\|u(Uq)^T + (Uq)u^T\|_F^2 \\
= 2(1+\delta)\|Uq\|_F^2 + 2(1+\delta)[q^T(U^T u)]^2 & \\
\leq 2(1+\delta)\sigma_r^2(U) + 2(1+\delta) \cdot \|U^T u\|_F^2. &
\end{aligned} \tag{66}
$$
By letting the vector $\tilde{v}$ be
$$\|\tilde{v}\|_2 = 1, \quad \lambda_{min}(\nabla f_s(M))u = \nabla f_s(M)\tilde{v},$$
the inequality (63) implies that
$$\|U^T u\|_F^2 = \frac{\|U^T \nabla f_s(M)\tilde{v}\|_F^2}{\lambda_{min}^2(\nabla f_s(M))} = \frac{\|U^T \nabla f_s(M)\tilde{v}\|_2^2}{\lambda_{min}^2(\nabla f_s(M))} \leq \frac{\|U^T \nabla f_s(M)\|_2^2 \|\tilde{v}\|_2^2}{\lambda_{min}^2(\nabla f_s(M))} \leq \frac{\lambda^2}{4G^2}.$$
Substituting into (66), we obtain
$$[\nabla^2 f_s(M)](\Delta U^T + U\Delta^T, \Delta U^T + U\Delta^T) \leq 2(1+\delta)\sigma_r^2(U) + (1+\delta) \cdot \frac{\lambda^2}{2G^2}. \tag{67}$$
Combining (65) and (67), it follows that
$$[\nabla^2 h_s(U)](\Delta, \Delta) \leq -2G + 2(1+\delta)\sigma_r^2(U) + (1+\delta) \cdot \frac{\lambda^2}{2G^2}.$$
Since $\|\Delta\|_F^2 = 1$, the above inequality implies
$$\lambda_{min}(\nabla^2 h_s(U)) \leq -2G + 2(1+\delta)\sigma_r^2(U) + (1+\delta) \cdot \frac{\lambda^2}{2G^2} \leq -(1+\delta)\tau.$$

$\square$

We finally give the counterpart of Lemma 5, which states that the condition (62) always holds when $\delta < 1/3$.

**Lemma 8.** *Given positive constants $\alpha, C, \epsilon, \lambda$, if*
$$\max\{\|UU^T\|_F, \|U^*(U^*)^T\|_F\} \leq C^2, \ \|U - U^*\|_F \geq \alpha, \ \|\nabla h_s(U)\|_F \leq \lambda, \ \delta < 1/3,$$
*then there exists a positive constant $\lambda_0(\delta, W^*, \alpha, C)$ such that*
$$\sigma_r^2(U) \leq \frac{1}{1+\delta}\left[ G - \frac{(1+\delta)\lambda^2}{4G^2} - \lambda \right] \tag{68}$$
*whenever*
$$0 < \lambda \leq \lambda_0(\delta, \sigma_r(M_s^*), \|M_s^*\|_F, \alpha, C).$$

*Proof.* We prove by contradiction, i.e., we assume
$$\sigma_r^2(U) > \frac{1}{1+\delta}\left[ G - \frac{(1+\delta)\lambda^2}{4G^2} - \lambda \right] \geq \frac{c\alpha^2}{1+\delta} + \text{poly}(\lambda). \tag{69}$$
To follow the proof of Lemma 5, we also divide the argument into three steps, although the first step is superficial.

**Step I.** We first give a lower bound for $\lambda_r(M)$. In the symmetric case, this step is straightforward, since we always have
$$\lambda_r^2(M) = \sigma_r^4(U). \tag{70}$$

**Step II.** Next, we derive an upper bound for $\lambda_r(M)$. We define

$$\bar{M} := \mathcal{P}_r \left[ M - \frac{1}{1+\delta} \nabla f_s(M) \right],$$

where $\mathcal{P}_r$ is the orthogonal projection onto the low-rank manifold (we do not drop negative eigenvalues in this proof). Since $M \neq M^*$ and $\delta < 1/3$, we recall that inequality (13) gives

$$-\phi(\bar{M}) \geq \frac{1-3\delta}{1-\delta}[f_s(M) - f_s(M^*)] \geq \frac{1-3\delta}{2} \|M - M^*\|_F^2$$
$$\geq (1-3\delta) \cdot (\sqrt{2} - 1)\sigma_r^2(W^*)\alpha^2 := K > 0,$$

where the second inequality comes from Lemma 1 and

$$-\phi(\bar{M}) = \langle \nabla f_s(M), M - \bar{M} \rangle - \frac{1+\delta}{2} \|M - \bar{M}\|_F^2.$$

Hence,

$$\langle \nabla f_s(M), M - \bar{M} \rangle - \frac{1+\delta}{2} \|M - \bar{M}\|_F^2 \geq K. \tag{71}$$

For simplicity, we define

$$N := -\frac{1}{1+\delta} \nabla f_s(M).$$

Then, $\bar{M} = \mathcal{P}_r(M + N)$ and the left-hand side of (71) is equal to

$$\langle \nabla f_s(M), M - \bar{M} \rangle - \frac{1+\delta}{2} \|M - \bar{M}\|_F^2$$
$$= (1+\delta)\langle N, \mathcal{P}_r(M + N) - M \rangle - \frac{1+\delta}{2} \|\mathcal{P}_r(M + N) - M\|_F^2$$
$$= \frac{1+\delta}{2} \left[ \|N\|_F^2 - \|N + M - \mathcal{P}_r(M + N)\|_F^2 \right]$$
$$= \frac{1+\delta}{2} \left[ \|N\|_F^2 - \|N + M\|_F^2 + \|\mathcal{P}_r(M + N)\|_F^2 \right]. \tag{72}$$

Similar to the proof of inequality (63), we can prove that

$$\|U^T N\|_F \leq \tilde{H} := \frac{\lambda}{2(1+\delta)}.$$

Then, we have

$$-\operatorname{tr}[N^T(UU^T)] \leq \|U^T N\|_F \|U\|_F \leq \tilde{H} \cdot \|U\|_F \leq \tilde{H} \cdot \sqrt{\sqrt{r}\|UU^T\|_F} \leq \sqrt[4]{r}C \cdot \tilde{H}.$$

Using the above relation, one can write

$$\|N\|_F^2 - \|N + M\|_F^2 = -2\operatorname{tr}[N^T(UU^T)] - \|M\|_F^2 \leq 2\sqrt[4]{r}C \cdot \tilde{H} - \|M\|_F^2.$$

Suppose that $\mathcal{P}_U$ is the orthogonal projections onto the column space of $U$. We define

$$N_1 := \mathcal{P}_U N \mathcal{P}_U, \quad N_2 := \mathcal{P}_U N(I - \mathcal{P}_U), \quad N_3 := (I - \mathcal{P}_U)N\mathcal{P}_U, \quad N_4 := (I - \mathcal{P}_U)N(I - \mathcal{P}_U).$$

Then, it follows from (69) that

$$\|N_1\|_F = \|\mathcal{P}_U N \mathcal{P}_U\|_F \leq \sigma_r^{-1}(U)\|U^T \mathcal{P}_U N \mathcal{P}_U\|_F \leq \sigma_r^{-1}(U)\|U^T N\|_F \leq \sigma_r^{-1}(U) \cdot \tilde{H}$$
$$\leq \left[ \sqrt{\frac{1+\delta}{G}} + \operatorname{poly}(\lambda) \right] \cdot \tilde{H} \leq \left[ \sqrt{\frac{1+\delta}{c\alpha^2}} + \operatorname{poly}(\lambda) \right] \cdot \tilde{H} := \kappa\tilde{H}.$$

Similarly, we can prove that

$$\|N_1 + N_2\|_F = \|\mathcal{P}_U N\|_F \leq \kappa\tilde{H}, \quad \|N_1 + N_3\|_F = \|N\mathcal{P}_V\|_F \leq \kappa\tilde{H},$$

which leads to

$$\|N_2\|_F \leq 2\kappa\tilde{H}, \quad \|N_3\|_F \leq 2\kappa\tilde{H}.$$

Using Weyl's theorem, the following holds for every $1 \leq i \leq r$:

$$|\lambda_i(M + N) - \lambda_i(M + N_4)| \leq \|N_1 + N_2 + N_3\|_2 \leq \|N_1 + N_2 + N_3\|_F \leq 3\kappa\tilde{H}.$$

Therefore, we have

$$
\begin{aligned}
\|\mathcal{P}_r(M + N)\|_F^2 &= \sum_{i=1}^{r} \lambda_i^2(M + N) \\
&\geq \sum_{i=1}^{r} \lambda_i^2(M + N_4) - r \cdot 3\kappa\tilde{H} \cdot (\|M + N\|_2 + \|M + N_4\|_2) \\
&\geq \sum_{i=1}^{r} \lambda_i^2(M + N_4) - 6r\kappa\tilde{H} \cdot (\|M\|_2 + \|N\|_2) \\
&\geq \sum_{i=1}^{r} \lambda_i^2(M + N_4) - 6r\kappa\tilde{H} \cdot \left( \|M\|_F + \frac{G}{1 + \delta} \right). \qquad (73)
\end{aligned}
$$

Similar to the asymmetric case, we can prove that

$$\frac{G}{1 + \delta} \leq \|M\|_F + \mathrm{poly}(\lambda).$$

holds under the assumption (69). Therefore, we obtain the bound

$$\|M\|_F + \|N\|_F \leq 2\|M\|_F + \mathrm{poly}(\lambda) \leq 2C^2 + \mathrm{poly}(\lambda).$$

Substituting back into the previous estimate (73), it follows that

$$\|\mathcal{P}_r(M + N)\|_F^2 \geq \sum_{i=1}^{r} \lambda_i^2(M + N_4) + \mathrm{poly}(\lambda).$$

Now, since $M$ and $N_4$ have orthogonal column and row spaces, the maximal $r$ eigenvalues of $M + N_4$ are simply the maximal $r$ eigenvalues of the eigenvalues of $M$ and $N_4$, which we assume to be

$$\lambda_i(M), \ i = 1, \ldots, k \quad \text{and} \quad \lambda_i(N_4), \ i = 1, \ldots, r - k.$$

Now, it follows from (72) that

$$
\begin{aligned}
\frac{2}{1 + \delta} &\left[ \langle \nabla f_s(M), M - \bar{M} \rangle - \frac{1 + \delta}{2} \|M - \bar{M}\|_F^2 \right] \\
&= \|N\|_F^2 - \|N + M\|_F^2 + \|\mathcal{P}_r(M + N)\|_F^2 \\
&\leq -\sum_{i=1}^{r} \lambda_i^2(M) + \sum_{i=1}^{k} \lambda_i^2(M) + \sum_{i=1}^{r-k} \lambda_i^2(N_4) + \mathrm{poly}(\lambda) + 2\sqrt[4]{r}C \cdot \tilde{H} \\
&= -\sum_{i=k+1}^{r} \lambda_i^2(M) + \sum_{i=1}^{r-k} \lambda_i^2(N_4) + \mathrm{poly}(\lambda). \qquad (74)
\end{aligned}
$$

Using the assumption (69) and the fact that $\lambda$ is small, we know that $\lambda_i(N_4) > 0$ for all $i \in \{1, \ldots, k\}$. Therefore,

$$-\sum_{i=k+1}^{r} \lambda_i^2(M) + \sum_{i=1}^{r-k} \lambda_i^2(N_4) \leq -(r - k)\lambda_r^2(M) + (r - k)\lambda_{max}(N_4)^2.$$

Substituting into (74) gives rise to

$$
\begin{aligned}
\frac{2}{1 + \delta} &\left[ \langle \nabla f_s(M), M - \bar{M} \rangle - \frac{1 + \delta}{2} \|M - \bar{M}\|_F^2 \right] \\
&\leq -(r - k)\lambda_r^2(M) + (r - k)\lambda_{max}(N_4)^2 + \mathrm{poly}(\lambda) \\
&\leq -(r - k)\lambda_r^2(M) + (r - k)\lambda_{max}(N)^2 + \mathrm{poly}(\lambda).
\end{aligned}
$$

If $k = r$, then the above inequality and inequality (71) imply that

$$\text{poly}(\lambda) \geq K = O(\alpha^2),$$

which contradicts the assumption that $\lambda$ is small. Hence, we conclude that $r - k \geq 1$. Combining with (71), we obtain the upper bound

$$\lambda_r^2(M) \leq -\frac{2}{1+\delta} \cdot \frac{K}{r-k} + \lambda_{max}(N)^2 + \frac{1}{r-k} \cdot \text{poly}(\lambda)$$

$$= -\frac{2}{1+\delta} \cdot \frac{K}{r} + \lambda_{max}(N)^2 + \text{poly}(\lambda). \tag{75}$$

**Step III.**    In the last step, we combine the relations (70) and (75), which leads to

$$\sigma_r^4(U) \leq -\frac{2}{1+\delta} \cdot \frac{K}{r} + \frac{1}{(1+\delta)^2} G^2 + \text{poly}(\lambda).$$

This means that

$$\sigma_r^4(U) + \frac{2}{1+\delta} \cdot \frac{K}{r} \leq \frac{1}{(1+\delta)^2} G^2 + \text{poly}(\lambda).$$

Since $K > 0$ has lower bounds that are independent of $\lambda$, we can choose $\lambda$ to be small enough such that

$$\sigma_r^4(U) + \frac{1}{1+\delta} \cdot \frac{K}{r} \leq \frac{1}{(1+\delta)^2} G^2.$$

However, considering the assumption (69), we have

$$\sigma_r^4(U) \geq \frac{1}{(1+\delta)^2} \left[ G - \frac{(1+\delta)\lambda^2}{4G^2} - \lambda \right]^2 = \frac{1}{(1+\delta)^2} G^2 - 2\lambda \cdot G + \text{poly}(\lambda)$$

$$\geq \frac{1}{(1+\delta)^2} G^2 - 2\lambda \cdot (1+\delta)C^2 + \text{poly}(\lambda) = \frac{1}{(1+\delta)^2} G^2 + \text{poly}(\lambda),$$

where the second inequality is due to $G \leq (1+\delta)C^2$, which can be proved similar to the asymmetric case. The above two inequalities cannot hold simultaneously when $\lambda$ is small enough. This contradiction means that the condition (68) holds by choosing

$$0 < \lambda \leq \lambda_0(\delta, \sigma_r(M_s^*), \|M_s^*\|_F, \alpha, C),$$

for a small enough positive constant $\lambda_0(\delta, \sigma_r(M_s^*), \|M_s^*\|_F, \alpha, C)$.

$\square$

*Proof of Theorem 7.*   We first choose

$$C := \left[ \frac{2(1+\delta)}{1-\delta} \|U^*(U^*)^T\|_F^2 \right]^{1/4}.$$

Then, we select $\lambda_1$ as

$$\lambda_1(\delta, r, \sigma_r(M_s^*), \|M_s^*\|_F, \alpha) := \min \left\{ \lambda_0(\delta, r, \sigma_r(M_s^*), \|M_s^*\|_F, \alpha, C), \frac{(1-\delta)C^3}{2\sqrt{r}} \right\}.$$

Finally, we combine Lemmas 6-8 to get the bounds for the gradient and the Hessian.   $\square$