# OpenReview forum: "General Low-rank Matrix Optimization: Geometric Analysis and Sharper Bounds"
_NeurIPS.cc/2021/Conference — NeurIPS 2021 Poster_

### Official Review · Reviewer_mEBV · 2021-06-25

**Rating:** 7
**Confidence:** 3

**Summary:**

This paper studies low-rank matrix recovery problems, where a loss function is optimized, which satisfies the restricted isometry property. The paper considers a formalization based on matrix factorization, as it is typically studied in non-convex optimization.  The paper is concerned with the question when the landscape of the problem is “nice”, i.e. when are there no spurious second-order saddle points and when does the loss landscape have the strict saddle property.


**Main Review:**

The paper improves both in the linear and non-linear upon existing work, both also for symmetric and non symmetric matrices. Moreover, the paper is clearly written and easy to read. Although the result Is “only” of theoretical interest, I feel that this is a strong paper, which merits publication.

I do not see any particular strong weakness, but I only have the following remarks:
-It might be helpful for the reader if the authors could make precise what they mean by “the problems are equivalent” in Theorem 2
-In the formulation of Theorem 4 it is required that alpha \in (1-delta,(1+delta)/2]. Is this a typo?! The interval is the empty set.
-typo in l. 319 “gradient decent”
-I think it might be helpful for the reader if the authors could motivate the non-linear formulation by some particular example.


**Time Spent Reviewing:**

4

---

> ### Author Response · Authors · 2021-08-08
> **Response to Reviewer mEBV**
>
> We thank the reviewer for the detailed comments and suggestions. Please find our response to the comments below.
>
> 1. It might be helpful for the reader if the authors could make precise what they mean by “the problems are equivalent” in Theorem 2.
>
> Response: We thank the reviewer for the valuable comment. By saying “the problems are equivalent”, we mean that any first-order critical point of problem (2) corresponds to a first-order critical point of problem (5) with balanced energy, i.e. $U^TU=V^TV$. We are sorry for the confusion and will make the definition precise in the revised paper.
>
> 2. In the formulation of Theorem 4 it is required that alpha \in (1-delta,(1+delta)/2]. Is this a typo?! The interval is the empty set.
>
> Response: We thank the reviewer for the helpful comment. This is not a typo. Indeed, the set is non-empty when delta > 1/3 and this is the reason why there does not exist spurious SOCP when delta <= 1/3. We will mention the requirement that delta <= 1/3 in the revised paper.
>
> 3. I think it might be helpful for the reader if the authors could motivate the non-linear formulation by some particular example.
>
> Response: We thank the reviewer for the constructive comment. The general low-rank matrix optimization problem has been well-studied in literature. One common example with non-linearity is the one-bit matrix sensing problem; please see [1-3] for more concrete discussions. We will provide some examples in the revised paper.
>
> References:
>
> [1] Bi, Y. and Lavaei, J., 2020. Global and local analyses of nonlinear low-rank matrix recovery problems. arXiv preprint arXiv:2010.04349.
>
> [2] Zhu, Z., Li, Q., Tang, G. and Wakin, M.B., 2018. Global optimality in low-rank matrix optimization. IEEE Transactions on Signal Processing, 66(13), pp.3614-3628.
>
> [3] Li, Q., Zhu, Z. and Tang, G., 2019. The non-convex geometry of low-rank matrix optimization. Information and Inference: A Journal of the IMA, 8(1), pp.51-96.

---

> > ### Comment · Reviewer_mEBV · 2021-08-22
> > **Response**
> >
> > I thank the authors for their response. My evaluation of the paper has been unchanged. Hence, I keep my score.

---

### Official Review · Reviewer_b54e · 2021-07-07

**Rating:** 7
**Confidence:** 3

**Summary:**

This paper studies low-rank matrix optimization, in particular solving $\underset{M\in\mathbb{R}^{m\times n}}{\min} f(M)$ under a rank constraint $\mathrm{rank}(M) \leq r$, where $f$ is a convex function. This captures a lot of problems like matrix completion, phase retrieval, etc.

The authors assume that $f$ is smooth and strongly convex along directions of rank $2r$, where $r$ is the rank of the global minimizer, and establish results in terms of the Restricted Isometry Property (RIP) constant $\delta_{2r}$ of $f$. The main contribution of the paper is to establish sharper upper and lower bounds on $\delta_{2r}$, under which every saddle point has a strictly negative eigenvalue, which by known results implies that the function be efficiently optimized using gradient descent. Specifically, the positive results work as long as $\delta_{2r} \leq 1/3$ and there are counterexamples when $\delta_{2r} \geq `1/2$.

Additionally, the authors show that their results transfer to the case when the matrix $M$ is constrained to be symmetric PSD, strengthening previous results, and also strengthen the RIP upper bound to $\delta_{2} \leq 1/2$ in the case of rank-1 matrices.

**Limitations And Societal Impact:**

A potential limitation is the constant RIP upper bound that is required, which seems like a strong assumption. Also, the strict saddle point results (Section 4) require the additional non-trivial assumption that the global optimizer has low rank. This is not usually the case in practice, because of noise.

**Main Review:**

**Originality**

To the best of my knowledge, the results are original. I find such structural results interesting, as they immediately imply convergence of a large family of gradient descent-based algorithms. That said, the RIP condition seems restrictive, and I wonder if one can achieve similar (but weaker) structural results after dropping the strong convexity assumption.

**Quality**

The results of the paper look correct to me. I have the following comments/suggestions to make:

* In the introduction, the authors mention convex relaxation approaches as the only alternatives for the problem of low-rank optimization. However, this is ignoring another line of work, which uses a technique similar to Orthogonal Matching Pursuit (OMP) (and variants) but generalized to the low-rank matrix domain. Additionally, some of these algorithms don't require a singular value thresholding of the solution, and also do not require a constant RIP upper bound (with the caveat of relaxing the rank constraint). Such results can be found in the following papers:
[1] Shai Shalev-Shwartz, Alon Gonen, Ohad Shamir: Large-Scale Convex Minimization with a Low-Rank Constraint. ICML 2011: 329-336
[2] Kyriakos Axiotis, Maxim Sviridenko: Local Search Algorithms for Rank-Constrained Convex Optimization. ICLR 2021

* In Theorem 3, it should be clarified in which sense the proof of Jain et al. (2010) is extended. Apart from the fact that it works for general functions rather than quadratic, is there some other difference in the statement? Also, a high level sentence on how the proof differs would be useful.

* In Corollary 2, what does the fact that the solution is positively correlated with the optimal solution imply? What is the practical significance of this property?

**Clarity**

The writing in the main body is good and accurate. Unfortunately, the proofs in the appendix are very technical without providing some sort of intuition. I have the following suggestions:

* The authors should add a short explanation on how the proofs work at a technical level. In particular, what are the new technical insights compared to Ha et al. (2020) and Zhu et al. (2021)?

* I couldn't exactly figure out how the regularization term in (5) helps in the analysis. It would be helpful if the authors could include a short (technical) explanation on why it is necessary for their analysis.

* The statements of Theorems 4 and 5 are not clear. Various symbols such as $\Sigma, \Lambda$ are defined but are not related to the function $f_a(\cdot)$. These should be fixed, and also a short description of what each condition means or why it is important would be helpful.

**Significance**

I think that the results are significant in that they are the first structural results that show that gradient descent can _efficiently_ solve a large family of low-rank optimization problems via the Burer-Monterio factorization approach.

**Time Spent Reviewing:**

2

---

> ### Author Response · Authors · 2021-08-08
> **Response to Reviewer b54e**
>
> We thank the reviewer for the detailed comments and suggestions. Please find our response to the comments below.
>
> 1. The RIP condition seems restrictive, and I wonder if one can achieve similar (but weaker) structural results after dropping the strong convexity assumption.
>
> Response: We thank the reviewer for the valuable comment. The RIP condition appears in a variety of applications of the low-rank matrix optimization problem; please see the survey paper [1]. In certain applications, even the RIP condition cannot be established over the whole low-rank manifold, we are able to establish similar strongly convex and smooth conditions on part of the manifold. If the iteration points of algorithms are constrained to or regularized (either explicitly or implicitly) towards those benign regions, the proof techniques in this work may still be applicable. Examples include the phase retrieval problem [2] and the matrix completion problem [2,3]. However, the analysis of the case when the strong convexity does not hold is usually application-specific and cannot be generalized to general low-rank problems. Moreover, the RIP assumption is standard in the literature of general low-rank matrix optimization problem. Furthermore, if we drop the strong convexity assumption, we are unable to achieve linear convergence in general [4]. The work [6] shows that the existence of RIP is enough to obtain guarantees on the local landscape of the problem and the size of this local region depends on the RIP that can be anything between 0 and 1 (however, the provided bounds on RIP are not sharp). Although we aimed to obtain sharp bounds on RIP for global landscape of the problem in this paper, we believe that our analysis can be adopted to obtain sharp RIP bounds for local regions. We leave the precise derivation to a future work since it needs a number of lemma and we have space restrictions.
>
> 2. In the introduction, the authors mention convex relaxation approaches as the only alternatives for the problem of low-rank optimization. However, this is ignoring another line of work, which uses a technique similar to Orthogonal Matching Pursuit (OMP) (and variants) but generalized to the low-rank matrix domain. Additionally, some of these algorithms don't require a singular value thresholding of the solution, and also do not require a constant RIP upper bound (with the caveat of relaxing the rank constraint). Such results can be found in the following papers.
>
> Response: We thank the reviewer for the very helpful comment. The algorithms in both references are interesting and we will add a discussion in the revised paper. It seems to the authors that these algorithms also utilize the factorized formulation of low-rank problems and require implementing the truncated SVD for r times. In addition, the algorithms need to solve inner problems that have a similar form of the low-rank optimization problem considered in this work. The total running time of those algorithms relies on the efficiency of solving the inner problems. Although the inner problems may be solved efficiently for certain examples, in general we can only solve the inner problems efficiently when some conditions similar to the RIP condition hold. We will discuss these interesting papers in our revised manuscript.
>
> 3. In Theorem 3, it should be clarified in which sense the proof of Jain et al. (2010) is extended. Apart from the fact that it works for general functions rather than quadratic, is there some other difference in the statement? Also, a high level sentence on how the proof differs would be useful.
>
> Response: We thank the reviewer for the constructive comment. The proof is almost identical to that in Jain et al. (2010) except that we have replaced the quadratic function with the RIP bounds. However, the result of this proof (inequality (13)) plays a key role in the subsequent proofs in Section 4. We note that Theorem 3 itself is not our major contribution and only serves as a hint that the landscape of the objective function may be benign when delta < 1/3. We will provide more explanations in the revised paper.
>
> 4. In Corollary 2, what does the fact that the solution is positively correlated with the optimal solution imply? What is the practical significance of this property?
>
> Response: We thank the reviewer for the valuable comment. The positive correlation result is largely of theoretical interest. We included this result to provide a transition between the regime delta >= 1/2, where spurious SOCPs may exist, and the regime delta <= 1/3, where the SOCP is unique. In practice, our result helps us find possible locations of spurious solutions relative to the ground truth in the regime with RIP between 1/3 and 1/2.
>
> 5. The authors should add a short explanation on how the proofs work at a technical level. In particular, what are the new technical insights compared to Ha et al. (2020) and Zhu et al. (2021)?
>
> Response: We thank the reviewer for the suggestion. Compared with Zhu et al. (2021), our results require only $RIP_{2r,2r}$ condition and the RIP constant can be as large as 1/3, instead of 1/5. Their proof requires reducing the original problem to the case where the RIP constant is 0, while our proof directly tackles the original problem. Compared with Ha et al. (2020), we have improved the existing inequality (ln.348) to a stronger inequality (ln.345); please see the discussion after Theorem 6. In addition, we briefly explain the intuition of each lemma in the proof of Theorem 6:
>
> - Lemma 2 proves that unbalanced solution cannot be a first-order critical point;
> - Lemma 3 proves that solution with large norm cannot be a first-order critical point;
> - Lemma 4 provides an upper bound on $\sigma_r(W)$ for first-order critical point;
> - Lemma 5 is to show that when $\sigma_r(W)$ is small for a first-order critical point, its Hessian has a negative curvature.
>
> We will provide these intuitions in the revised paper.
>
> 6. I couldn't exactly figure out how the regularization term in (5) helps in the analysis. It would be helpful if the authors could include a short (technical) explanation on why it is necessary for their analysis.
>
> Response: We thank the reviewer for the constructive comment. The regularization term is only needed in the proofs of the strict saddle property. More specifically, we need the regularization term to prove that points with a large norm cannot be an approximate SOCP (Lemma 3). Intuitively, the regularization term helps balance the "energies" of U and V and therefore helps reduce the symmetry of the objective function, which further reduces the severe non-convexity of the objective function; please see more explanations in [5]. In addition, the regularized asymmetric problem can be transformed to a symmetric problem and therefore the regularizer makes the landscapes of asymmetric problems similar to those of symmetric problems; please see the analysis in Appendix B. Moreover, the non-existence of SOCPs does not rely on the regularization term.
>
> 7. The statements of Theorems 4 and 5 are not clear. Various symbols such as $\Sigma, \Lambda$ are defined but are not related to the function $f_a$. These should be fixed, and also a short description of what each condition means or why it is important would be helpful.
>
> Response: We thank the reviewer for the valuable comment. Intuitively, $\Sigma$ and $\Lambda$ correspond to the singular values of the SOCP and the gradient at the SOCP, respectively. Matrices $A,B,C,D$ correspond to the SVD of the global optimum. We will include this intuition together with some discussions on the conditions in the revised paper.
>
> 8. The strict saddle point results (Section 4) require the additional non-trivial assumption that the global optimizer has low rank. This is not usually the case in practice, because of noise.
>
> Response: We thank the reviewer for the valuable comment. We leave the analysis of the noisy case to future works. In the noisy scenario, there may exist spurious solutions but all of them are in the vicinity of the ground truth. The design of an algorithm to reach to this neighborhood of the solution requires the ground truth to be low rank. In other words, the strict saddle point property for the noisy case is somewhat different due to the existence of saddles and spurious solutions in a small vicinity of the ground truth, and so our assumption should naturally be revised in the noisy case.
>
>
> References:
>
> [1] Chi, Y., Lu, Y.M. and Chen, Y., 2019. Nonconvex optimization meets low-rank matrix factorization: An overview. IEEE Transactions on Signal Processing, 67(20), pp.5239-5269.
>
> [2] Ma, C., Wang, K., Chi, Y. and Chen, Y., 2018, July. Implicit regularization in nonconvex statistical estimation: Gradient descent converges linearly for phase retrieval and matrix completion. In International Conference on Machine Learning (pp. 3345-3354). PMLR.
>
> [3] Chen, Y. and Wainwright, M.J., 2015. Fast low-rank estimation by projected gradient descent: General statistical and algorithmic guarantees. arXiv preprint arXiv:1509.03025.
>
> [4] Bhojanapalli, S., Kyrillidis, A. and Sanghavi, S., 2016, June. Dropping convexity for faster semi-definite optimization. In Conference on Learning Theory (pp. 530-582). PMLR.
>
> [5] Tu, S., Boczar, R., Simchowitz, M., Soltanolkotabi, M. and Recht, B., 2016, June. Low-rank solutions of linear matrix equations via procrustes flow. In International Conference on Machine Learning (pp. 964-973). PMLR.
>
> [6] Zhang, R.Y., Sojoudi, S. and Lavaei, J., 2019. Sharp Restricted Isometry Bounds for the Inexistence of Spurious Local Minima in Nonconvex Matrix Recovery. J. Mach. Learn. Res., 20(114), pp.1-34.

---

### Official Review · Reviewer_ogYY · 2021-07-08

**Rating:** 6
**Confidence:** 4

**Summary:**

This article considers the problem of minimizing a function $f$ over the set of matrices, either in the symmetric case ($f$ is defined for symmetric semidefinite positive matrices only) or in the asymmetric one ($f$ is defined over the set of all matrices with prescribed dimensions). It is assumed that
- the global minimizer of $f$ has low-rank (at most $r$) ;
- the Hessian of $f$, when restricted to matrices with rank at most $2r$, satisfies the restricted isometry property with constant $\delta$.

The authors introduce the corresponding "factorized" version of $f$ ($U \to f(UU^T)$ in the symmetric case, $(U,V) \to f(UV^T)$ in the asymmetric one, possibly with an additional regularizer - here, $U$ and $V$ have r columns) and try to establish conditions on $\delta$, as tight as possible, which guarantee that the factorized version has no spurious second-order critical point (SSOCP).

The main results are:
- a general (more or less) necessary and sufficient condition for the existence of a function with a SSOCP (Theorems 4 and 5);
- when $r=1$, if $\delta<1/2$, there is no SSOCP; the constant $1/2$ is tight (Corollary 1; known in the symmetric case, novel in the asymmetric one);
- for arbitrary $r$, if $\delta\leq 1/3$, there is no SSOCP (Corollary 2; minor improvement over Ha et al (2020));
- for arbitrary $r$, if $1/3<\delta<1/2$, SSOCP are positively correlated with the ground truth (Corollary 2);
- if $\delta<1/3$, $f$ has the strict saddle property (in the symmetric case, or for the regularized version in the asymmetric one) (Theorems 6 and 7).

**Limitations And Societal Impact:**

Not applicable to this article.

**Main Review:**

I think this is a good article, notably for the following two reasons:
1) The link between the RIP constant of a function and the non-existence of SSOCP for its factorized version has raised significant interest in the last years, and this article almost closes the question, by considering the most general setting (non-linear functions, symmetric or asymmetric case) and providing almost tight answers (only Corollary 2 is not tight, because of the gap between $1/3$ and $1/2$). The contribution of this article is therefore susceptible, in my opinion, to be of interest for a relatively wide audience.
2) From a technical point of view, it seems to me that this article is perfectly correct. (I did not find the time to read the thirty pages of proofs, but still read around half of them, and did not see any possible issue.)

However, in my opinion, this article also suffers from the following two drawbacks:
1) It seems to me that the question of finding the optimal RIP constant which guarantees the non-existence of SSOCP is very theoretical (although I admit it is a natural one). Indeed, in concrete settings, it is in general difficult to determine whether a function satisfies the RIP, and for which constant. The RIP is also not the only property which determines the existence or non-existence of SSOCP: there are functions which do not satisfy the RIP but for which no SSOCP exists. This slightly narrows down the implications of the presented results for concrete applications.
2) The presentation of the results and proofs does not seem optimal to me, which I think could reduce the impact and "reusability" of the article.
     1) In my opinion, Theorem 4 (which is the first "main" theorem) is too complicated to be easily understandable and reapplicable. It seems to me that the authors should try to simplify it into a simpler set of necessary and sufficient conditions. For instance, I think that the following property is equivalent to the one of Theorem 4 (I include a proof at the end of the review):
$$\begin{gather*}
\mbox{There exists diagonal matrices }\Sigma, \Lambda\mbox{ and matrices }A,B,C,D\mbox{ such that}
\\\\
(1+\delta)\min_i \Sigma_{ii} \geq \max_i \Lambda_{ii}, \Sigma \succ 0, \Lambda \succeq 0,\\\\
\mathrm{Tr}(\Lambda^2) \leq 2 \langle \Lambda, CD^T\rangle - (1-\delta^2)
(\mathrm{Tr}(\Sigma^2)-2 \langle\Sigma,AB^T\rangle + ||AB^T||^2_F + ||AD^T||^2_F + ||CB^T||_F^2 + ||CD^T||_F^2), \\\\
\langle \Lambda, CD^T \rangle \ne 0.
\end{gather*}$$
Why not replace the property in Theorem 4 by this simpler one? And there may be further simplifications which I have not seen.
     2) I found the proofs a bit painful to read. No intuition or high-level description is provided. To me, it looked like a long sequence of computations and technical arguments seemingly coming out of nowhere. I think this will make it difficult for other researchers to reuse some of the proof techniques.
     3) I found Section 4 (about the strict saddle property) quite confusing. I may have missed something, but my impression is that under very mild additional assumptions (continuity of the Hessian of f, for instance), the non-existence of SSOCP directly implies that, for all $\alpha$, there exists $\beta,\gamma$ such that Problem (4) (or (5)) has the $(\alpha,\beta,\gamma)$-strict saddle property. (This can be shown with standard continuity arguments.) As a consequence, from my understanding, what is non-trivial in Theorems 6 and 7 is that $\beta$ and $\gamma$ only depend on $f$ through a limited set of properties (RIP constant, rank of the global minimizer ...). But this is not discussed in the article; I was therefore unable to fully understand the interest of Theorems 6 and 7.
     4) Section 2, presented as a "motivating example", seems a bit unrelated with the rest of the article. Theorem 3 is not an example or a particular case of one of the subsequent results. And since the proof is not even sketched outside the appendix, it does not provide insights into proof techniques which will be used in the following sections.
     5) I did not understand the small insight into the proof of Theorem 6 provided on page 9. I did not understand what the equation after line 345 meant, why it implied the result, and I did not find it in the appendix.

In addition, I have the following minor remarks:
- The expression "necessary and sufficient conditions for the existence of spurious second-order critical points" can be slightly misleading. I think that something like "necessary and sufficient conditions for the existence of a function with the $\delta$-RIP and a spurious second-order critical point" would be more precise.
- Page 2, line 43: shouldn't "$O(n)$" and "$O(n+m)$" be "$O(rn)$" and "$O(r(n+m))$"?
- Page 2, line 47: I disagree with "is required", since convergence can occur even when the RIP does not hold; I would rather say something like "can be used".
- Page 2, line 54, "the curvature of the Hessian": why not simply "the Hessian"?
- Page 3, lines 88-89: "the result of second-order critical points" is unclear to me.
- Page 5, line 188: shouldn't there be a logarithmic term inside "$O(nmr)$"?
- Page 5, line 189: I did not find that "matrix multiplication" was a very clear comparison point (especially with no precision on the size of the multiplied matrices).
- Page 5, line 191: "combining" $\to$ "combined".
- Page 6, line 211: "iteration points" $\to$ "iterates"?
- Page 6, line 218: "Burer-Monterio" $\to$ "Burer-Montero"
- Page 6, line 237: "there exist" $\to$ "there exists"
- Page 7, line 245: "a basis of" $\to$ "a basis for"?
- Page 7, line 282, as well as the corresponding section of the appendix: why does the symmetric function need to satisfy $RIP_{4r,2r}$ and not $RIP_{4r,4r}$?
- Page 8, equation after line 316: I think this is only true if $||U-U^*||_F$ is small *and $\mathcal{X}^*$ is bounded*.
- Page 8, lines 324 to 326: this assertion is not clear enough. The boundedness of the trajectories does not ensure that the Hessian is Lipschitz continuous without any assumption on $f$ besides the fact that it is twice differentiable. (If the Hessian is discontinuous at $0$, for instance, it is not Lipschitz continuous over any set containing $0$, even if trajectories are bounded.)
- Page 13, Theorem 8: in my opinion, "first-order critical point" should be defined. Since the constraint set of Problem (1) is not a manifold, I do not think there is a fully standard definition.
- Theorem 9, equation after line 503: I think "$\sigma_r(\tilde{U})$" should be "$\sigma_r(\tilde{U})^2$".
- Page 13, line 506: "the goal" $\to$ "Since the goal"?
- Page 13, lines 511, 515, 518: "hold" $\to$ "holds"
- Page 16, line 583: $\alpha$ should be defined as $\frac{2\delta}{1-\delta}$.
- Page 16, line 591: "that" $\to$ "that the"?
- Sufficiency part of the proof, starting on page 19: why not set $U=\left(\begin{smallmatrix}I_{2r}\\\\ 0_{n-2r,2r}\end{smallmatrix}\right), V=\left(\begin{smallmatrix}I_{2r}\\\\ 0_{m-2r,2r}\end{smallmatrix}\right)$? This would slightly simplify the definitions.
- Page 21, line 686: "$\sigma(\tilde{M})$" should be "$\sigma_r(\tilde{M})$".
- Appendix D.3: I think that the definition of $V$ has been forgotten.
- Page 24, equation after line 748: "$\forall Z\in\mathbb{R}^{n\times m}$" should come before "there exists", since $c$ can depend on $Z$.
- Page 24, equation after line 740: I think that there should be absolute values (otherwise the right-hand side can be negative, if $t>1$, for instance).
- Page 25, equation after line 784: I do not understand the reason for taking the minimum with $\frac{1}{3}$.

Proof of the "simpler property": if the property in Theorem 4 holds, then the simpler one also does. Conversely, let us assume that the simpler one holds. We set
$$\alpha = \frac{\langle \Lambda, CD^T \rangle}{\mathrm{Tr}(\Sigma^2)-2 \langle\Sigma,AB^T\rangle + ||AB^T||^2_F + ||AD^T||^2_F + ||CB^T||_F^2 + ||CD^T||_F^2}.$$
With this definition, the three lines of Inequality (6) are true. We must only verify that $1-\delta< \alpha < \frac{1+\delta}{2}$.

The third line of the simpler property can be rewritten as $||X_1|| \leq \delta ||X_2||$, where
$$X_1=\begin{pmatrix}AB^T-\Sigma & AD^T \\\\ CB^T & CD^T-\Lambda \end{pmatrix},\qquad
X_2=\begin{pmatrix}AB^T-\Sigma & AD^T \\\\ CB^T & CD^T \end{pmatrix}.$$
As a consequence, by Cauchy-Schwarz,
$$\alpha = \frac{\langle X_2-X_1,X_2\rangle}{||X_2||^2} = 1 - \frac{\langle X_1,X_2\rangle}{||X_2||^2} \geq 1-\delta$$
(and we can show that equality is impossible.)

For the inequality $\alpha < \frac{1+\delta}{2}$, we set
$$X_3 = \begin{pmatrix}\Sigma&0\\\\0&\frac{\Lambda}{1+\delta}\end{pmatrix},\qquad
X_4 = \begin{pmatrix}A\\\\C\end{pmatrix} \begin{pmatrix}B^T&D^T\end{pmatrix}.$$
From the first line of the simpler condition, the projection of $X_3$ onto the set of rank $r$ matrices is
$$X_5 = \begin{pmatrix}\Sigma&0\\\\0&0\end{pmatrix},$$
which means that
$$\begin{gather*}
||X_3-X_5||^2 \leq ||X_3-X_4||^2 \\\\
\iff
(1+\delta)^{-2}\mathrm{Tr}(\Lambda^2) \leq (1+\delta)^{-2}\mathrm{Tr}(\Lambda^2) - \frac{2}{1+\delta} \langle \Lambda, CD^T\rangle + ||X_2||^2 \\\\
\iff \alpha \leq \frac{1+\delta}{2}.
\end{gather*}$$
(Again, one can show that equality is impossible.)

**Added after author feedback:**

I thank the authors for their feedback. On the interest of the Restricted Isometry Property, I do agree with the authors regarding the applications mentionned in their response. I do not agree with the fact that RIP "captures the complexity of the class of all operators". However, it does not make the question considered by the authors uninteresting. I simply think it is of more theoretical than practical interest.

Regarding the presentation issues mentioned in my first review, I was a bit disappointed by the first response provided by the authors, as explained during the discussion. The second response was more convincing for me, but a bit too vague to be fully convincing.

**Time Spent Reviewing:**

10

---

> ### Author Response · Authors · 2021-08-08
> **Response to Reviewer ogYY**
>
> We thank the reviewer for the detailed comments and suggestions. Please find our response to the comments below.
>
> 1. It seems to me that the question of finding the optimal RIP constant which guarantees the non-existence of SSOCP is very theoretical (although I admit it is a natural one). Indeed, in concrete settings, it is in general difficult to determine whether a function satisfies the RIP, and for which constant. The RIP is also not the only property which determines the existence or non-existence of SSOCP: there are functions which do not satisfy the RIP but for which no SSOCP exists. This slightly narrows down the implications of the presented results for concrete applications.
>
> Response: We thank the reviewer for the very helpful comment. We agree that the RIP constant is hard to estimate for an arbitrary function. However, there are several cases where the RIP constant can be estimated in terms of the sample complexity [1]. For problems that do not satisfy the RIP condition on the whole low-rank manifold, it suffices to satisfy the RIP condition in a local region around the solution to be able to obtain local results rather than global results [4]. As another example, the matrix completion problem has benign landscape in the region where the incoherence parameter is bounded [2,3]. The proof techniques proposed in this work under the RIP assumption may also provide inspirations to those more general cases. In addition, for several applications, the gradient-based algorithms can successfully find the global optimum with random initialization, even the RIP constant is larger than the bounds available in the literature. This work provides a non-trivial step towards closing the gap between the practice and the theory by providing more accurate bounds that characterize the landscape of the underlying optimization.
>
> 2. In my opinion, Theorem 4 (which is the first "main" theorem) is too complicated to be easily understandable and reapplicable. It seems to me that the authors should try to simplify it into a simpler set of necessary and sufficient conditions. For instance, I think that the following property is equivalent to the one of Theorem 4 (I include a proof at the end of the review). Why not replace the property in Theorem 4 by this simpler one? And there may be further simplifications which I have not seen.
>
> Response: We thank the reviewer for the constructive comment, and for working through out conditions to simplify them. We admit that there may exist equivalent forms of Theorem 4 that have a simpler expression. However, we chose to present the conditions in Theorem 4 in the current form since it directly helps with deriving Corollaries 1-2, which are of more practical importance. Although the conditions in Theorem 4 are complicated, they are powerful in deriving RIP bounds and constructing counterexamples, especially the condition in the third line of (6). Since the major role of Theorem 4 is to provide an intermediate step, we will rename Theorem 4 to be a lemma. We will also add a discussion on other equivalent forms of the conditions, as kindly derived by the reviewer.
>
> 3. I found the proofs a bit painful to read. No intuition or high-level description is provided. To me, it looked like a long sequence of computations and technical arguments seemingly coming out of nowhere. I think this will make it difficult for other researchers to reuse some of the proof techniques.
>
> Response: We thank the reviewer for the valuable comment. For the proofs in Section 3, the new technical tool is the conditions in the third line of Theorem 4; please see the discussion after Theorem 4. For the proofs in Section 3, we have improved the existing inequality (ln.348) to a stronger inequality (ln.345); please see the discussion after Theorem 6. We agree that the proofs are highly technical and we briefly explain the intuition of each lemma in the proof of Theorem 6:
>
> - Lemma 2 proves that unbalanced solution cannot be a first-order critical point;
> - Lemma 3 proves that solution with large norm cannot be a first-order critical point;
> - Lemma 4 provides an upper bound on $\sigma_r(W)$ for first-order critical point;
> - Lemma 5 is to show that when $\sigma_r(W)$ is small for a first-order critical point, its Hessian has a negative curvature.
>
> We will provide these intuitions in the revised paper.
>
> 4. I found Section 4 (about the strict saddle property) quite confusing. I may have missed something, but my impression is that under very mild additional assumptions (continuity of the Hessian of f, for instance), the non-existence of SSOCP directly implies that, for all $\alpha$, there exists $\beta,\gamma$ such that Problem (4) (or (5)) has the $(\alpha,\beta,\gamma)$-strict saddle property. (This can be shown with standard continuity arguments.) As a consequence, from my understanding, what is non-trivial in Theorems 6 and 7 is that $\beta$ and $\gamma$ only depend on f through a limited set of properties (RIP constant, rank of the global minimizer ...). But this is not discussed in the article; I was therefore unable to fully understand the interest of Theorems 6 and 7.
>
> Response: We thank the reviewer for the valuable comment. The limited dependence is important since the parameters $\alpha,\beta,\gamma$ can lead to the concrete convergence rate of saddle-escaping algorithms for a class of objective functions. Otherwise, the parameters $\alpha,\beta,\gamma$ can be arbitrarily close to 0 for functions in this class and the convergence rate will be arbitrarily slow. More importantly, the non-existence of spurious SOCPs cannot guarantee the non-existence of spurious approximate SOCPs, i.e., points with a small gradient and an "almost" positive semi-definite Hessian. We note that this cannot be derived from the standard continuity arguments since approximate SOCPs may be far away from the set of (exact) SOCPs. Saddle-escaping algorithms may spend exponential time to escape all approximate SOCPs. We will provide the discussion in the revised paper.
>
> 5. Section 2, presented as a "motivating example", seems a bit unrelated with the rest of the article. Theorem 3 is not an example or a particular case of one of the subsequent results. And since the proof is not even sketched outside the appendix, it does not provide insights into proof techniques which will be used in the following sections.
>
> Response: We thank the reviewer for the constructive comment. The results in Section 2 provide a hint that the landscape may be benign when the RIP constant is smaller than 1/3 and we may be able to establish linear convergence in this case. In addition, the inequality (13) in the proof of Theorem 3 is the key inequality for the proofs in Section 4. We will make this point clear in the revised paper.
>
> 6. I did not understand the small insight into the proof of Theorem 6 provided on page 9. I did not understand what the equation after line 345 meant, why it implied the result, and I did not find it in the appendix.
>
> Response: We thank the reviewer for the valuable comment. The inequality after ln.345 is the key inequality we have used in the proof of Theorem 6. More specifically, we have used it in Step II of the proof of Lemma 5. This is an improvement over the existing inequality, which is listed after ln.348. This improvement leads to the sharper results in this work.
>
> References:
>
> [1] Candès, E.J. and Recht, B., 2009. Exact matrix completion via convex optimization. Foundations of Computational mathematics, 9(6), pp.717-772.
>
> [2] Chen, Y. and Wainwright, M.J., 2015. Fast low-rank estimation by projected gradient descent: General statistical and algorithmic guarantees. arXiv preprint arXiv:1509.03025.
>
> [3] Ge, R., Jin, C. and Zheng, Y., 2017, July. No spurious local minima in nonconvex low rank problems: A unified geometric analysis. In International Conference on Machine Learning (pp. 1233-1242). PMLR.
>
> [4] Zhang, R.Y., Sojoudi, S. and Lavaei, J., 2019. Sharp Restricted Isometry Bounds for the Inexistence of Spurious Local Minima in Nonconvex Matrix Recovery. J. Mach. Learn. Res., 20(114), pp.1-34.

---

> > ### Comment · Reviewer_ogYY · 2021-08-10
> > **Response to author feedback**
> >
> > I thank the authors for their reply. I am slightly disappointed to see that they disagree with almost all my suggestions, although I do respect their opinion.
> >
> > A few additional thoughts and clarifications:
> > 1) I agree with the examples provided by the authors of when the RIP is useful. However, I do not agree with the assertion
> >     > In addition, for several applications, the gradient-based algorithms can successfully find the global optimum with random initialization, even the RIP constant is larger than the bounds available in the literature. This work provides a non-trivial step towards closing the gap between the practice and the theory by providing more accurate bounds that characterize the landscape of the underlying optimization.
> >
> >     Indeed, it seems to me that the success of randomly-initialized gradient-based algorithms in settings where the RIP constant is worse than what is required by results like Corollaries 1 to 3 precisely hints at the fact that the RIP constant alone does not characterize the properties of the optimization landscape, and that a RIP-based analysis necessarily fails at explaining many situations where no SSOCP exists.
> > 2) I do not agree with
> >     > However, we chose to present the conditions in Theorem 4 in the current form since it directly helps with deriving Corollaries 1-2, which are of more practical importance. Although the conditions in Theorem 4 are complicated, they are powerful in deriving RIP bounds and constructing counterexamples, especially the condition in the third line of (6).
> >
> >     Indeed, I tried to prove Corollary 1 using the "simpler" condition I proposed, and obtained a quite similar, but simpler proof. As to constructing counterexamples, since the "simpler" condition is trivially implied by the current one, constructing objects which satisfy the "simpler" condition cannot be more complex than constructing objects which satisfy the current one.
> > 3) I thank the authors for highlighting some key technical points. However, this is not sufficient, in my opinion, to make the 29 pages of appendix easy to read (which was the subject of my comment).
> > 4) I do agree. Thanks to te authors for these clarifications.
> > 5) Thanks to the authors for this explanation.
> > 6)
> >     > The inequality after ln.345 is the key inequality we have used in the proof of Theorem 6. More specifically, we have used it in Step II of the proof of Lemma 5.
> >
> >     I thank the authors for this clarification. However, I still do not see what this equation means, and have not found where it is used in Step II of the proof of Lemma 5.

---

> > > ### Author Response · Authors · 2021-08-10
> > > **Response to Reviewer ogYY**
> > >
> > > We are extremely grateful to the reviewer for generously spending time on reading our manuscript and continuing to provide constructive comments. We sincerely apologize that our responses may have implied that we may not implement the suggestions, which was by no means our intention. Indeed, we strive to improve the quality of our work and the suggestions by the reviewers are very helpful in this regard. In particular, we would like to discuss the main ones here:
> > >
> > >
> > > 1. We thank the reviewer for the feedback on RIP. We think the point of disagreement on to what extent RIP is useful is related to whether a particular function or a class of functions is to be studied using RIP. Finding a necessary and sufficient condition for the non-existence of spurious SOCPs for a specific function is almost impossible, and the only way is to find better necessary conditions and sufficient conditions. We believe that improving the RIP constant is the most direct step. To understand this point, consider the minimization of $||\mathcal A(xx^T)-b||^2$. If $\mathcal A$ belongs to the class of all possible operators, then the our work has shown that there is an operator $\mathcal A$ with RIP=0.5 that has a spurious solution. So, RIP=0.5 captures the complexity of the class of all operators. However, although in this paper we have given the sharpest RIP constant in literature, this is not the end, as the RIP condition can be further combined with other conditions. For example, a given operator in a particular application may be sparse or structured, and therefore it makes sense to restrict the analysis to that class of operators if a tight bound is needed. The paper [1] has shown that there are necessary and sufficient RIP-type conditions for the non-existence of spurious solutions for a given class of structured or sparse problems. This shows the usefulness of RIP. So, if for a particular operator the RIP is greater than 1/2 but there is no spurious solution, that is not due to the failure of RIP but due to not incorporating the problem structure. We mentioned in our previous response that "This work provides a non-trivial step towards closing the gap between the practice and the theory by providing more accurate bounds that characterize the landscape of the underlying optimization.” By this statement, we meant that since we believe RIP is important, it is essential to find the right bound to be able to somehow find the boundary of easy and hard problems for classes of operators.
> > >
> > >
> > > 2. To address the reviewer’s suggestion, we mentioned in our previous response that "We will also add a discussion on other equivalent forms of the conditions, as kindly derived by the reviewer.” So, we planned to keep the current conditions but add the equivalent form derived by the reviewer to the paper as well. Based on the new feedback from the reviewer about being able to prove Corollary 1 using the reviewer’s conditions, we will certainly work on the proof on our part and after checking that all of the required proofs in the paper can be derived from the new conditions, we will simply eliminate our current conditions, use the simpler equivalent form, and acknowledge the reviewer in the paper for simplifying our conditions. We would love to make our paper easier to understand and simplify the proofs as much as possible, and the review comment has been quite constructive.
> > >
> > >
> > > 3. We thank the reviewer for the constructive comment. For the proof of the strict saddle property, Lemmas 2-4 show that the gradient cannot be small in certain regions. For points outside these regions and are not close to the global minima, Lemma 4 proves an upper bound on $\sigma_r(W)$ and Lemma 5 utilizes this upper bound to derive the existence of a negative curvature. Combining the two parts, we know that for any point that is not close to the global minima, either the gradient is large or the Hessian has a negative curvature. More specifically, we mentioned in our previous response to include the following:
> > >
> > > “1- Lemma 2 proves that unbalanced solution cannot be a first-order critical point;
> > >
> > > 2- Lemma 3 proves that solution with large norm cannot be a first-order critical point;
> > >
> > > 3- Lemma 4 provides an upper bound on  $\sigma_r(W)$  for first-order critical point;
> > >
> > > 4- Lemma 5 is to show that when $\sigma_r(W)$ is small for a first-order critical point, its Hessian has a negative curvature.”
> > >
> > > Of course, adding four lines to the paper will not make the proofs easier to understand, and what we meant to say (which was not clear and we apologize) is that the above four lines are the summaries of the intuitions to be included in the paper, but we will definitely elaborate on each intuition. For example, for Lemma 2 we say that the goal is to prove the unbalanced solution is not a critical point and then at a high level discuss the steps leading to this conclusion. Since there is no page limit in the appendix, we will provide some short intuitions in the main paper and more detailed but high-level intuitions in the appendix.
> > >
> > >
> > > 6. We thank the reviewer for the valuable comment and we are sorry for the confusion. There was a typo in the key inequality and we have corrected it to:
> > >
> > > $$ ||\nabla f(M)||_2^2 \geq (1+\delta)^2\sigma_r^2(M) + O[ f_a(M) - f_a(M^*) ]. $$
> > >
> > > The above inequality is not directly proved in the paper. However, if we replace $K$ with $O[ f_a(M) - f_a(M^*) ]$, the above inequality can be similarly proved as inequality (61) (up to an approximation error term). More specifically, combining the first inequality in ln.917:
> > >
> > > $$ -\phi(\bar{M}) \geq \frac{1-3\delta}{1-\delta}[ f_a(M) - f_a(M^*) ], $$
> > >
> > > where $\phi(\bar{M})$ is defined after ln.918, and the inequality after ln.938:
> > >
> > > $$ -\frac{2}{1+\delta}\phi(\bar{M}) \leq -(r-k)\sigma_r^2(M) + (r-k)\left|| \frac{1}{1+\delta}\nabla f_a(M) \right||_2^2 + \mathrm{poly}(\sqrt{\epsilon},\lambda), $$
> > >
> > > we get
> > >
> > > $$ || \nabla f_a(M) ||_2^2 \geq (1+\delta)^2 \sigma_r^2(M) + \frac{2(1-3\delta)}{r(1-\delta^2)}[ f_a(M) - f_a(M^*) ] + \mathrm{poly}(\sqrt{\epsilon},\lambda). $$
> > >
> > > The key inequality does not directly lead to the conclusion, but it is the key step that leads to a sharper bound compared to existing results and is the main intuition behind the proof of Theorem 6. More specifically, the key inequality provides a relation between the smallest singular value $\sigma_r(W)$ and the norm $||\nabla f_a(M)||_2$, which further leads to inequality (51). Inequality (51) proves the existence of negative curvature direction for the Hessian of an approximate first-order critical point. We present the key inequality instead of calling (61) the key inequality in the main text because it requires fewer notation and is simpler. In the revised paper, we will correct this typo and make this point clear.
> > >
> > > References:
> > >
> > > [1] I. Molybog, S. Sojoudi, and J. Lavaei, Role of Sparsity and Structure in the Optimization Landscape of Non-convex Matrix Sensing,  Mathematical Programming, 2021.

---

### Official Review · Reviewer_vdzf · 2021-07-19

**Rating:** 8
**Confidence:** 5

**Summary:**


This paper considers the global geometry of general low-rank minimization problems via the Burer-Monterio factorization approach. Compared to the existing state-of-the-art results, this work improves the RIP constant required for the general low-rank minimization problem to have a global benign landscape.

**Limitations And Societal Impact:**

(1) The work is motivated by the high computational complexity of the Singular Value Projection (SVP) Algorithm. However, without theoretical or empirical comparisons, it is not clear why the Burer-Monterio factorization approach achieves better overall computational complexities. I think the Burer-Monterio factorization approach has better computation complexity within per iteration, but might require high iteration complexities.

(2) It is not clear why there is a gap between the RIP constants for the rank-1 case and the rank-r case.

**Main Review:**


This paper considers the global geometry of general low-rank minimization problems via the Burer-Monterio factorization approach.
(1) Compared to the existing state-of-the-art results, this work improves the RIP constant required for the general low-rank minimization problem to have a global benign landscape.
(2) The theoretical analyses are sound. This work unifies the theoretical framework for both symmetric and nonsymmetric cases. The sufficiency of the RIP bound is theoretically proved and the necessity is justified by a constructed example.

**Time Spent Reviewing:**

6

---

> ### Author Response · Authors · 2021-08-08
> **Response to Reviewer vdzf**
>
> We thank the reviewer for the detailed comments and suggestions. Please find our response to the comments below.
>
> 1. The work is motivated by the high computational complexity of the Singular Value Projection (SVP) Algorithm. However, without theoretical or empirical comparisons, it is not clear why the Burer-Monterio factorization approach achieves better overall computational complexities. I think the Burer-Monterio factorization approach has better computation complexity within per iteration, but might require high iteration complexities.
>
> Response: We thank the reviewer for the valuable comment. We admit that the Burer-Monterio factorization approach may have a higher iteration complexity than the SVP algorithm. However, in the case when the RIP constant is smaller than 1/3, we can guarantee the global linear convergence using both approaches. Therefore, the number of iterations is on the order of $O(\log(\epsilon))$ for both approaches, where $\epsilon$ is the desired optimality gap. On the other hand, the per-iteration complexity is $O(nr)$ and $O(n^2)$ for the Burer-Monterio approach and the SVP algorithm, respectively. In the case when $r \ll n$, the Burer-Monterio approach is $O(n/r)$ faster than the SVP algorithm. Therefore, we believe that the total computational complexity of the Burer-Monterio approach is better than the SVP algorithm and this intuition is widely adopted in literature. In addition, this intuition has been numerically verified in [1]. We will provide the discussion in the revised paper.
>
> 2. It is not clear why there is a gap between the RIP constants for the rank-1 case and the rank-r case.
>
> Response: We thank the reviewer for the valuable comment. In the rank-1 case, most variables in Theorem 4 are reduced to vectors. This reduction enables a simplification of the conditions in Theorem 4 and leads to a stronger bound (delta < 1/2). In the rank-r case, the simplification for the rank-1 case fails and we can only get a weaker bound (delta <= 1/3). To understand the fundamental difference between rank-1 and higher rank cases, consider a positive semidefinite matrix X where the goal is to decompose it as BB^T. In the rank-1 case, there are two isolated solutions for B. However, in the rank-r case with r>1, there is a continuum of solutions for B. So, as soon as the rank becomes greater than one, we move from the regime of isolated solutions to infinitely many solutions for which the second-order sufficient optimality conditions can never hold when finding B is formulated as an optimization problem due to the continuity of the solution trajectory. Hence, having a gap between the RIPs of the rank-1 and rank-r cases is expected.
>
>
> References:
>
> [1] Zheng, Q. and Lafferty, J., 2015. A convergent gradient descent algorithm for rank minimization and semidefinite programming from random linear measurements. arXiv preprint arXiv:1506.06081.

---

> > ### Comment · Reviewer_vdzf · 2021-08-22
> > **Response to author feedback**
> >
> > Thank you very much for your careful response and for addressing my major concerns. Overall, this work improves the RIP bound compared to the state-of-the-art results. It is an important contribution to the field of low-rank matrix recovery. I am happy to increase my score to 8.

---

### Decision · Program_Chairs · 2021-09-27

**Decision:**

Accept (Poster)

**Comment:**

The paper provides improved analysis for both symmetric and asymmetric low-rank optimization problems via the Burer-Monterio factorization approach. Specifically, based on the assumption that the problem satisfies restricted isometry property (RIP), this paper provides much tighter RIP constants for ensuring the following three cases: (1) non-existence of spurious second-order critical point, (2) strict saddle property, and (3) the existence of a counterexample that has spurious second-order critical points. On the other hand, reviewers pointed out that the current presentation of the main results (like Theorem 4) appears too complicated and could be simplified and improved. Also, the analysis cannot be applied to other low-rank optimization problems (e.g., phase retrieval) that have no RIP.

Overall, all reviewers appreciate the technical contribution of this paper and agree that the merits outweigh the pitfalls, so I recommend accept. Nevertheless, please modify the paper accordingly to improve the presentation and highlight the limitation of RIP-based analysis in the introduction while mentioning some practical problems (e.g., quantum state tomography) that obey RIP.